**Anatomy of the magmatic plumbing system of Los Humeros Caldera (Mexico): implications for geothermal systems**

Federico Lucci[1, *], Gerardo Carrasco-Núñez[2], Federico Rossetti[1], Thomas Theye[3], John C. White[4], Stefano Urbani[1], Hossein Azizi[5], Yoshihiro Asahara[6], and Guido Giordano[1, 7]

[1]Dipartimento di Scienze, Sez. Scienze Geologiche, Università Roma Tre, Largo S. L. Murialdo 1, 00146 Roma, Italy

[2]Centro de Geociencias, Universidad Nacional Autónoma de México, Campus UNAM Juriquilla, 76100, Queretaro, Mexico

[3]Institut für Anorganische Chemie, Universität Stuttgart, Stuttgart, Germany

[4]Department of Geosciences, Eastern Kentucky University, Richmond, KY 40475, USA

[5]Mining Department, Faculty of Engineering, University of Kurdistan, Sanandaj, Iran

[6]Department of Earth and Environmental Sciences, Graduate School of Environmental Studies, Nagoya University, Nagoya 464-8601, Japan

[7]CNR - IDPA, Via Luigi Mangiagalli 34, 20133 Milano

*Corresponding Author e-mail: federico.lucci@uniroma3.it

**ABSTRACT**

Understanding the anatomy of magma plumbing systems of active volcanoes is essential not only for unraveling magma dynamics and eruptive behaviors, but also to define the geometry, depth and temperature of the heat sources for geothermal exploration. The Pleistocene-Holocene Los Humeros volcanic complex is part of the Eastern Trans-Mexican Volcanic Belt (Central Mexico) and it represents one of the most important exploited geothermal fields in Mexico with *ca*. 90 MW of produced electricity.

A field-based petrologic and thermobarometric study of lavas erupted during the Holocene (post-Caldera stage) has been performed with the aim to decipher the anatomy of the magmatic plumbing system existing beneath the caldera. New petrographic, whole rock major element data and mineral chemistry were integrated with a suite of mineral-liquid thermobarometric models. Compared with previous studies, where a single voluminous melt-controlled magma chamber (or "Standard Model") at shallow depths was proposed, our results support a scenario characterized by a heterogeneous multilayered system comprising a deep (*ca*. 30 km) basaltic reservoir feeding progressively shallower and smaller discrete stagnation layers, and batches up to shallow-crust conditions (1kbar, *ca*. 3km). Evolution of melts in the feeding system is mainly controlled by differentiation processes through fractional crystallization (plagioclase + clinopyroxene + olivine + spinel). We demonstrate the inadequacy of conceptual models based on the classical, melt-dominated, single, voluminous, long-lived magma chamber for the magmatic plumbing systems at LHVC. We instead propose a magmatic plumbing system made of multiple, more or less interconnected magma transport and storage layers within the crust, feeding small (ephemeral) magma chambers at shallow-crustal conditions. This revised scenario provides a new configuration of the heat source feeding the geothermal reservoir that should be taken into account to drive future exploration and exploitation strategies.

**Keywords**

Magmatic plumbing system, Thermobarometry, heat source, geothermal exploration, Trans Mexican Volcanic Belt, Los Humeros, Mexico

## 1.INTRODUCTION

Recent views on the structure of volcanic plumbing systems have moved from the "Standard Model" (*sensu* Gualda and Ghiorso, 2013) of a single, bowl-shaped magma chamber inside which all petrologic processes of differentiation and assimilation occur (e.g. Hildreth, 1979, 1981; Hildreth and Wilson, 2007) to more complex arrays of stratified and variably interconnected of transient magma accumulation zones, set in largely crystallized and vertically extensive mush zones (e.g., Bachman and Bergantz, 2004; 2008; Cashman and Giordano, 2014; Cashman et al., 2017). Furthermore, the time required for the assembly of large magma chambers is now believed to be very short, within the span of decades to a few thousands of years, for tens to hundreds of km$^3$ of eruptible magma (e.g. Glazner, 2004; Charlier et al., 2007), which are then rapidly evacuated during eruptions of caldera-forming ignimbrites (e.g., Begué et al., 2014; Rivera et al., 2014; Wotzlaw et al, 2014; Matthews et al., 2015; Carrasco-Núñez et al., 2018). Key factors in determining the internal architecture of the magmatic systems is the magma intrusion rate, which controls whether successive pulses of magma will coalesce to form progressively larger chambers, as well as the formation of ductile shells surrounding the magma chamber that prevent country rock failure, favoring the inflation of the reservoir (Jellinek and de Paolo, 1981; Annen, 2009). Numerical simulations suggest that caldera systems smaller than 100 km$^2$ are fed by plumbing systems encapsulated by country rock that remains sufficiently brittle, while larger systems are more ductile, which favors an increase in size (Gregg et al., 2012).

The implications of such innovative conceptual models on the modeling of the heat source in magmatic-bearing geothermal systems are significant. Nonetheless, common numerical modeling of conductive-convective heat transfer in caldera-related geothermal systems have commonly envisaged the classic magma chamber as a single body, chemically stratified, entirely at magmatic temperatures, whose dimensions and depths have been usually constrained by volcanological and petrological data (e.g. Verma, 1985; Wohletz et al., 1999). More complex modeling requires the "unpacking" of the stratigraphy of a volcano by the identification of the various "magma chambers" or magma storage layers that fed the different eruptions in space and time (e.g., Solano et al., 2014; Di Renzo et al., 2016; Cashman et al., 2017; Jackson et al., 2018).

A key to decipher where magmas are stored and, therefore, the anatomy of a magmatic plumbing system, is the understanding of pre-eruptive processes such as mineral crystallization and the migration and stagnation of melts prior to their eruption (Feng and Zhu, 2018, Putirka, 2008; Keiding and Sigmarsson, 2012; Scott et al., 2012, Barker et al., 2015; Jeffery et al., 2013; Cashman and Giordano, 2014; Pamukcu et al., 2015; Lucci et al., 2018). Early segregated minerals reflect the magmatic environment (i.e., pressure – temperature – magma/fluid composition, oxidation state) and thus their growth, texture and chemistry provide an important archive of information (Ginibre et al., 2002; Feng and Zhu, 2018; Ginibre et al., 2007; Streck, 2008; Giuffrida and Viccaro, 2017; Viccaro et al., 2016; Putirka et al., 2008; Lucci et al., 2018).

Accordingly, petrographic observations and mineral chemistry of primary minerals, integrated with opportunely selected thermobarometry models (e.g., Putirka, 2008; Masotta et al., 2013) could lead to the comprehension and reconstruction of the magmatic storage/feeding systems of the erupted products (Feng and Zhu, 2018; Giuffrida and Viccaro, 2017; Elardo and Shearer, 2014; Petrone et al., 2016; Zheng et al., 2016; Eskandari et al., 2018; Shane and Coote, 2018; Scott et al., 2012; Stroncik et al., 2009; Barker et al., 2015; Jeffery et al., 2013; Keiding and Sigmarsson, 2012).

In this paper we present a geothermobarometric study of the post-caldera Pleistocene-Holocene products of the Los Humeros volcanic complex (LHVC), located at the eastern terminus of the Neogene-Quaternary Trans-Mexican Volcanic Belt (TMVB) (Fig. 1), with the goal of reconstruting the present-day geometry and structure of the magmatic plumbing system. These data are used to develop a conceptual model for the magmatic heat source of the active and currently exploited geothermal system. The magmatic heat source for LHVC has been constrained by the geometry of the caldera, the volume and mass balance calculations of the associated ignimbrites (Ferriz and Mahood, 1984, 1987; Verma, 1984, 1985a, 1985b, Verma et al., 1990; Verma and Andaverde, 1995; Verma et al., 2011; Verma et al. 2013), all related to a single magma body.

We propose a new and more realistic vision of the magmatic plumbing systems, made of multiple interconnected magma stagnation layers within the crust. These new findings must be considered into the new developing conceptual geothermal models to improve strategies for exploration and exploitation of the geothermal system within the LHVC. The results and approach presented in this work have also a general value and could represent an efficient strategy to explore and reconstruct, through petrological investigation, the pre-eruptive geometry and the anatomy of active magmatic feeding systems.

## 2.GEOLOGICAL SETTING

### 2.1 Regional Geology

LHVC is the largest and easternmost Quaternary caldera (Fig. 1) of the 1200 km-long active continental arc of the Trans-Mexican Volcanic Belt (TMVB), generated since *ca.* 20 Ma by the subduction of Cocos plate beneath central Mexico (e.g. Demant, 1978, Ferrari et al. 1999, 2012; Gomez-Tuena et al., 2003, 2007a, 2007b, 2018; Norini et al., 2015). LHVC is located in the eastern sector of the TMVB, which is characterized by monogenetic volcanism, scattered basaltic cinder and scoria cones, maar volcanoes of basaltic and rhyolitic composition, large rhyolitic domes and major stratovolcanoes such as Pico de Orizaba (or Citlaltépetl) and Cofre de Perote (e.g., Yáñez and García, 1982; Negendank et al., 1987; Carrasco-Núñez et al., 2010, 2012a;).

The Paleozoic to Mesozoic crystalline basement of eastern TMVB is exposed along the Teziutlán Massif (Viniegra, 1965; Ferriz and Mahood, 1984) made of metamorphic (greenschists K-Ar dated at 207 ± 7 Ma, in Yáñez and García, 1982) and intrusive (granodiorites and granites with whole-rock K-Ar ages of 246 ±7 Ma

and 181 ±5 Ma, Yáñez and García, 1982) rocks. The crystalline basement is partially covered by a thick, highly deformed Mesozoic sedimentary succession part of the Sierra Madre Oriental NW-SE fold and thrust belt formed by the Late Cretaceous to Paleocene compressional Laramide Orogeny (e.g. Campos-Enriquez and Garduño-Monroy, 1987; Suter, 1987; Fitz-Díaz et al., 2018). Oligocene to Miocene granodiorite and syenite intrusions are randomly exposed within the area (whole-rock K-Ar ages of 31-15 Ma, Yáñez and García, 1982). Miocene volcanism in the area  is represented by andesites of the Cerro Grande volcanic complex (Gómez-Tuena and Carrasco-Núñez, 2000), dated at 8.9 to 11 Ma (K/Ar method on whole-rock, Carrasco-Núñez et al. 1997), and the Cuyoaco Andesite dated at 10.5 Ma (K/Ar method, Yáñez and García, 1982) to the west of LHVC, which may correlate with the Alseseca Andesite (Yáñez and García, 1982) exposed to the north. Neogene andesitic volcanism (Ferriz and Mahood, 1984; Yáñez and García, 1982) represented by the Teziutlán Andesite, K-Ar dated (whole-rock) between 3.5 and 5 Ma by Yáñez and García (1982) and at 1.55 Ma by Ferriz and Mahood (1984). This andesitic activity was recently dated by the $^{40}$Ar/$^{39}$Ar method at 2.61-1.46 Ma (Carrasco-Núñez et al., 2017a) and correlates with most of the thick andesitic successions of the subsurface geology of LHVC.

## 2.2 Los Humeros Volcanic Complex

The volcanic evolution of the LHVC consists of three main stages (Carrasco-Núñez et al., 2018): (i) pre-caldera stage; (ii) caldera stage; and (iii) post-caldera stage. The pre-caldera stage is represented by relatively abundant rhyolitic domes, which erupted mainly on the western side of Los Humeros caldera, with an isolated spot to the south, and in some buried lavas identified in the geothermal well-logs (Carrasco-Núñez et al., 2017a). This volcanism has been recently dated by both U-Th and $^{40}$Ar/$^{39}$Ar methods (Carrasco-Núñez et al., 2018), providing ages spanning from 693.0±1.9 ($^{40}$Ar/$^{39}$Ar, plagioclase) to 270±17 ka (U/Th, zircon), which overlap with the age range obtained from other domes of the western sector outside the caldera, where K-Ar ages (sanidine) of 360±100 ka and 220±40 ka were obtained (Ferriz and Mahood, 1984). The caldera stage consists of two major caldera-forming events, separated by a large Plinian eruptive episode. The first and largest caldera-forming eruption produced Los Humeros caldera (18 km in diameter) during the emplacement of the Xaltipan ignimbrite, a rhyolitic, welded to non-welded, ash-rich deposit, radially distributed around the caldera. The dense rock equivalent (DRE) volume of this event was estimated at 115 km$^3$ by Ferriz and Mahood (1984). The age of the Xaltipan ignimbrite was established by whole-rock K-Ar dating at 460±20 ka (plagioclase) and 460±130 ka (biotite) (Ferriz and Mahood, 1984), however Carrasco-Núñez et al. (2018) based on coupled zircon U-Th dating and $^{40}$Ar/$^{39}$Ar method (plagioclase) geochronology provided a younger age of 164.0 ± 4.2 ka.

Following this catastrophic event an eruptive pause occurred, resuming with a sequence of intermittent Plinian episodes at 70±23 ka ($^{40}$Ar/$^{39}$Ar method on plagioclase, Carrasco-Núñez et al., 2018), separated by short gaps marked by thin paleosoils. The deposits consist of thick (1-6 m) coarse pumice-rich, well-sorted,

massive and diffuse-stratified layers,rhyodacitic in composition, which are grouped as the Faby Tuff (Ferriz and Mahood 1984; Willcox, 2011). The second caldera-forming episode produced the 9-10 km large Los Potreros caldera, which is associated with the emplacement of the compositionally-zoned andesitic-rhyodacitic-rhyolitic Zaragoza ignimbrite (Carrasco-Núñez et al., 2012b). This is an intraplinian pyroclastic flow deposit, with an estimated volume of ca. 15 km$^3$ DRE (Carrasco-Núñez and Branney, 2005). Previous ages of this unit were reported at 100 ka (K-Ar dating, plagioclase: Ferriz and Mahhod, 1984) and at 140±24 ka ($^{40}$Ar/$^{39}$Ar method, plagioclase: Willcox, 2011). However, a new $^{40}$Ar/$^{39}$Ar (on plagioclase) younger age of 69±16 ka for the Zaragoza ignimbrite was recently obtained (Carrasco-Núñez et al., 2018), confirmed by the fact the Zaragoza ignimbrite overlies a rhyodacitic lava flow dated at 74.2±4.5 ka (zircon U-Th dating).

According to Carrasco-Núñez et al. (2018) during the post-caldera stage (Fig. 1) two different eruptive phases occurred. The first one was a late Pleistocene resurgent phase characterized by the emplacement of felsic domes in the central area at about 44.8±1.7 ka (zircon U-Th ages; Carrasco-Núñez et al., 2018), which is slightly younger than the previously reported whole-rock K-Ar date (60±20 ka, glass: Ferriz and Mahood, 1984). Outside of the caldera, to the north, a rhyolitic dome erupted at 50.7±4.4 ka ($^{40}$Ar/$^{39}$Ar, plagioclase; Carrasco-Núñez et al., 2018). This was followed by a sequence of explosive eruptions, producing dacitic pumice fall units (Xoxoctic Tuff, 0.6 km$^3$) and interbedded breccia and pyroclastic flows deposits of the Llano Tuff (Ferriz and Mahood 1984; Willcox, 2011), with a maximum age of 28.3±1.1 ka (C-14, Cal BP 30630, Rojas-Ortega, 2016). The second eruptive phase of the post-caldera stage is a Holocene ring-fracture and bimodal activity that occurred towards the south, north and central part of Los Humeros caldera (Carrasco- Núñez et al., 2017). It is characterized by alternating episodes of effusive and explosive volcanism with a wide range of compositions. The volcanic products span from basaltic-andesitic, basaltic trachytic, trachyandesitic lava flows and dacitic, trachydacitic, andesitic and basaltic pumice and scoria fall deposits erupted by tens of monogenetic eruptive centers located in the LHVC (Ferriz and Mahood, 1984; Dávila-Harris and Carrasco-Núñez, 2014; Norini et al., 2015; Carrasco-Núñez et al., 2017b). Most of the effusive activity was initially thought to be 40-20 ka (whole-rock K-Ar dating, Ferriz and Mahood, 1984), however recent dating reveals that most of this activity is Holocene (Carrasco et al., 2017b). Trachyandesitic and andesitic basalts lavas erupted to the north of the LHVC at about 8.9±0.03 ka (C-14 age, Carrasco-Núñez et al, 2017b). A rhythmic alternation of contemporaneous bimodal explosive activity produced trachyandesitic and basaltic fall layers grouped as the Cuicuiltic Member erupted at 7.3±0.1 ka (C-14 age, Dávila-Harris and Carrasco-Núñez, 2014). This activity migrated towards the southern caldera rim to forms a well-defined lava field. This ring-fracture episode erupted trachyandesite and olivine-bearing basaltic lava flows, at 3.9±0.13 ka (C-14 age, Carrasco-Núñez et al, 2017b), and the most recent eruptions erupted trachytic lava flows near the SW caldera rim, at 2.8± 0.03 ka (C-14 age, Carrasco-Núñez et al, 2017b).

**2.3 Los Humeros geothermal system**

The LHVC hosts one of the three most important geothermal fields in Mexico, with an installed 93 MW of electric power produced from 20 geothermal wells (Romo-Jones et al., 2017). The existing conceptual models for the Los Humeros geothermal field (LHGF) (see Norini et al., 2015 for a review) stem from the hypothesis of a unique, large and voluminous cooling magma chamber of 1000-1500 Km$^3$ in volume, at depth of 5 to 10 km from the surface (Verma, 1984, 1985a, 1985b, 2000; Verma et al., 1990; Verma and Andaverde, 1995; Verma et al., 2011; Verma et al. 2013; Carrasco-Núñez et al. 2018), representing the heat source of the geothermal field (Martínez et al., 1983; Verma, 1983, 2000; Campos-Enríquez and Garduño-Monroy, 1987). However, the LHGF is characterized by a low number of productive geothermal wells (ca. 20 out of 50; Norini et al., 2015; Carrasco-Núñez et al., 2017a). The confined distribution of these productive wells along the NNW-SSE trending "Maxtaloya-Los Humeros-Loma Blanca" fault system (MHBfs in Fig. 1) cutting across the Los Potreros caldera (e.g., Norini et al., 2015; Carrasco-Núñez et al., 2017a) also corresponds to the almost unique, narrow and sharp surface thermal anomaly recognized within the caldera (Norini et al., 2015). These observations raise doubts on the existence of a voluminous superficial heat source feeding the LHGF. That makes it important, for a better comprehension and exploitation of the geothermal resource, a revised assessment of the structure of the magmatic plumbing system beneath LHVC.

**3.MATERIALS AND METHODS**

In this work we focus on petrographic investigations including a textural and chemical (mineral chemistry and major-elements bulk-rock) characterization of the Los Humeros post-caldera stage (LHPCS) (Carrasco-Núñez et al. 2017b). Following the recently published geological map (Carrasco-Núñez et al. 2017b) and geochronology (Carrasco-Núñez et al. 2018) of the LHVC, more than fifty samples of the LHPCS lavas were collected in the field with the goal to describe all the compositional variability of erupted products during the LHPCS activity (Figs. 1, 2a-d). In the description of the volcanic units, abbreviations follow Carrasco et al. (2017b). Petrographic characterization through polarized-light microscopy (PLM) was produced for all collected samples. The most preserved and representative samples of every major LHPCS volcanic unit were then selected (Fig. 1) for bulk and mineral chemistry investigations.

With respect to the intra-caldera domain (Fig. 2a), we selected lava samples belonging to: (i) LH27-1 from the mafic lavas inside the Xalapasco Crater (Qb1), (ii) LH27-2 from the Maxtaloya trachyandesites (Qta4) constituting the rim-walls of Xalapasco craters, (iii) LH4 from San Antonio-Las Chapas lavas (Qta3) outcropping in the Los Humeros town, (iv) LH5-2 from mafic lavas (Qb1) outcropping west to Los Humeros town, (v) LH5-1 from Chicomiapa-Los Parajes felsic lavas (Qt2) outcropping in the north-western part of Los Potreros caldera, (vi) LH6-1 from El Pajaro unit (Qt1) outcropping in the north-western part of Los Potreros caldera. In addition to these units, we also selected three more samples (LH13, LH26-1 and LH26-2) from

lavas and domes of intermediate compositions, outcropping (Fig. 2b-c) in the center of Los Potreros caldera
between Xalapasco crater and Los Humeros town, and not reported on the published geological map.
Concerning the extra-caldera products (Fig. 2d), we selected one sample for each of the four major lava
flows: (i) LH15 from El Limón lava flow (Qab), (ii) LH21-2 from Sarabia lava flow (Qta1), (iii) LH17 from
Tepeyahualco lava flow (Qtab), and (iv) LH18 from Texcal lava flow (Qb1).
The samples were investigated first by optical microscopy and then through back scattered electron (BSE)
imaging for the definition of magmatic fabrics, textures and constituent mineral assemblages. Mineral
chemistry was then defined through electron microprobe analyses (EMPA). Whole rocks (major element)
composition of selected samples was obtained through ion coupled plasma – optical emission (ICP-OE) and
X-ray fluorescence (XRF) analyses. Analytical protocols are described in the Appendix A.
In the following, mineral abbreviations follow Whitney and Evans (2010), whereas types of zoning and
textures are after Ginibre et al. (2002), Streck (2008) and Renjith (2014).

**4.MAJOR ELEMENT BULK COMPOSITION**
Studied samples show a continuous series from mafic to felsic compositions, with $SiO_2$ ranging 46.5-67.6
wt%, and $Na_2O + K_2O$ ranging 3.4-9.2 wt% (with $K_2O/Na_2O < 1$) (Fig. 3a; Table 1). LHPCS mafic rocks ($SiO_2<$
50 wt%; 3 samples) show composition with $SiO_2$ 46.5-49.4 wt%, $Al_2O_3$ 16.2-17.1 wt%, CaO 9.8-10.7 wt%,
MgO 8.0-8.4 wt% with Mg# (molar $MgO/[MgO+FeO_{tot}]$) = 60-61 and $Na_2O+K_2O$ ranging 3.4-3.5 wt%. LHPCS
intermediate rocks ($50<SiO_2<63$ wt%; 8 samples) contain 54.4-62.1 wt% $SiO_2$, with $Al_2O_3$ 15.7-20.7 wt%,
$Na_2O+K_2O$ 5.3-7.1 wt%, MgO 2.2-3.6 wt% (Mg# 43-51), and low CaO 4.6-8.5 wt%. LHPCS felsic rocks
($SiO_2>63$ wt%; 2 samples) show $SiO_2$ ranging 64.9-67.6 wt%, associated with $Al_2O_3$ 15.5-15.8 wt%, MgO 0.7-
1.2 wt% (Mg#: 26-34), CaO 2.1-2.8 wt%, and $Na_2O+K_2O$ 8.2-9.2 wt%.
On the total alkali versus silica (TAS) diagram (Le Maitre et al., 2002) LHPCS lavas span from basalt to
trachyte (Fig. 3b). Los Humeros mafic rocks fall in the "Basalt" field and, following the existing literature
(e.g., Barberi et al., 1975, Bellieni et al., 1983; Le Maitre et al., 2002; White et al., 2009; Giordano et al.,
2012), can be classified as mildly-alkaline (or transitional) basalts and alkali-basalts. The high $TiO_2$ contents
(1.34-1.5 wt%), together with MgO <12 wt% and low $Al_2O_3/TiO_2$ values (average value 11.5) exclude the LH
mafic rocks as high-Mg melts (picrites) or komatiites (e.g., Redman and Keays, 1985; Arndt and Jenner,
1986; Le Maitre et al., 2002; Gao and Zhou, 2013; Azizi et al., 2018a, 2018b).
Intermediate products fall in the "Basaltic trachyandesites" and "Trachyandesites" fields; these rocks will
be referred hereafter as "trachyandesites". The Los Humeros felsic (i.e., $SiO_2$ >63 wt%) lava samples fall in
the "Trachyte" field. Selected Harker diagrams for major elements are presented in Figure 3, using $SiO_2$ wt%
as differentiation index. Negative correlations are observed for CaO (Fig. 3c) and Mg# (Fig. 3d), whereas
positive correlations are observed for $Na_2O$ (Fig. 3c).

**5. PETROGRAPHY**

**5.1 Basalts**

LHPCS basalts show vesicle-rich (up to 35 vol%) highly-porphyritic (phenocrysts up to 50 vol%) textures (Fig. 4a-d). Studied basalts do not show presence of fragments from host-rocks or from previous magmatic rocks, therefore can be defined as lithic-free (e.g., Geshi and Oikawa, 2014). The magmatic fabric is fluidal as defined by the alignment of plagioclase laths in the groundmass. Based on the presence of clinopyroxene (Cpx) in the mineral assemblage, basalts can be further subdivided into: (i) Cpx-free basalt of the extra-caldera Texcal lava flow (LH18); and (ii) Cpx-bearing basalts of the intra-caldera lavas at western Los Potreros and at Xalapasco crater (LH5-2 and LH27-1, respectively).

Cpx-free basalt (LH18) contains euhedral to subhedral olivine (ca. 20 vol%) and euhedral to anhedral plagioclase (ca. 25-30 vol%) phenocrysts in a holocrystalline groundmass. The latter consistis of plagioclase with swallow-tail morphology, dendritic to spinifex olivines and opaque oxides (Fig. 4a, b). Olivine and plagioclase phenocrysts are generally slightly chemically zoned (see below), showing homogeneous cores with normal concordant monotonous zoning texture at outer rim (Fig. 4a, b). Major phenocrysts of olivine (up to 2.5 mm in size) with Cr-spinel inclusions are observed (Fig. 4a). It is worth notice that no pyroxenes are observed in all samples collected from Texcal basalt.

Cpx-bearing intra-caldera basalts (LH5-2, LH27-1) show euhedral to subhedral plagioclase (ca. 25 vol%), euhedral olivine (ca. 10-15 vol%), subhedral to anhedral yellow to colorless clinopyroxene (ca. 10-15 vol%) and rare subhedral anorthoclase (< 2 vol%) phenocrysts (Fig. 4c) in a holocrystalline groundmass (Fig. 4d). The latter is made up of (in order of abundance) elongated platy plagioclase, olivine, colorless to green clinopyroxene, opaque oxides and rare alkali-feldspar. All phenocrysts show core-rim zoning textures (Fig. 4c, d): (i) olivine, plagioclase and clinopyroxene with homogeneous cores and normal concordant monotonous zoning at outer rims, (ii) plagioclase and clinopyroxene with homogeneous cores and low-amplitude euhedral oscillatory zoning at rims, (iii) rare plagioclase and clinopyroxene with homogeneous cores and normal concordant step zoning at rims, and (iv) very rare plagioclase with patchy cores and normal convolute monotonous zoning at rims. Large phenocrysts of olivine (up to 1.5 mm in size) and plagioclase (up to 3 mm in length) are commonly observed. Vesicle size is up to 5mm in diameter.

**5.2 Trachyandesites**

LHPCS intermediate volcanic products are lithic-free and show low- to medium-porphyritic textures (phenocrysts ranging 10-40 vol%), with a general fluidal fabric as indicated by orientation of plagioclase and clinopyroxene laths in the groundmass (Fig. 4e, f). Intermediate products vary from poorly vesicular (< 10 vol% in LH13) to vesicle-rich (ca. 30 vol% in LH4) lavas. In the highest vesiculated sample (LH4), size of vesicles (3-5 mm in diameter) is comparable to those of intra-caldera basalts (LH5-2, LH27-1). Based on the presence of orthopyroxene (Opx) in the mineral assemblage, trachyandesites can be further subdivided

into: (i) Opx-free (LH21, LH15); and (ii) Opx-bearing (LH4, LH13, LH17, LH26-1, LH26-2, LH27-2) trachyandesites.

Opx-free trachyandesites shows euhedral to subhedral plagioclase (ca. 15-20 vol%), euhedral to subhedral olivine (ca. 10 vol%), euhedral to anhedral yellow-to-colorless clinopyroxene (ca. 10-20 vol%), subhedral alkali-feldspars (ca. 10-15 vol%) phenocrysts, in a holocrystalline microcrystalline groundmass composed of elongated platy plagioclase, colorless clinopyroxene, olvine, alkali-feldspar and opaque oxides, in order of microlites abundance.

Opx-bearing trachyandesites are generally characterized by euhedral to subhedral plagioclase (ca. 15-20 vol%), euhedral to subhedral clinopyroxene (ca. 10-20 vol%), euhedral colorless orthopyroxene (ca. 10-20 vol%), euhedral to subhedral alkali-feldspars (ca. 10-15 vol%) and euhedral to subhedral olivine (< 10 vol%) phenocrysts in a holocrystalline to hypohyaline microcrystalline groundmass made of feldspar (plagioclase and alkali-feldspars) microlites, pyroxene (clinopyroxene and orthopyroxene) microlites, olivine microlites, opaque minerals and glass (Fig. 4f). To note it is the absence of olivine in the Maxtaloya trachyandesite sample LH27-2.

Most of phenocrysts observed in LHPCS trachyandesites show zoning textures characterized by homogeneous cores surrounded by (i) monotonous zoning at outer rims, (ii) low-amplitude euhedral oscillatory zoning at rims and (iii) normal concordant step zoning at rims. Homogeneous unzoned clinopyroxene phenocrysts are commonly observed. Major phenocrysts of clinopyroxene (up to 2 mm in size) and plagioclase (up to 2 mm in length) characterized by homogeneous cores and normal concordant monotonous zoning are reported in all studied trachyandesites. It is also reported the presence of (i) rare clinopyroxene phenocrystals with growth mantle textures, (ii) rare plagioclase phenocrysts with patchy rounded zone corners cores, (iii) very rare clinopyroxenes with homogeneous cores and growth mantle texture at rims are observed. Very rare large phenocrysts of olivine (1.5-2.0 mm in size) presenting resorption patterns at rim and characterized by spinel inclusion are reported in the LH26-1 sample.

## 5.3 Trachytes

LHPCS trachytes show lithic-free phyric textures, with low porphyritic index (phenocrysts ranging 10-25 vol%), and fluidal fabrics as shown by iso-orientation of plagioclase, alkali-feldspars and clinopyroxene laths in the groundmass (Fig. 4g, h). They range from vesicle-poor (< 5 vol%) to vesicle-free textures, with size of vesicles never exceeding 0.05 mm in diameter. The two analyzed trachytic samples (LH5-1 and LH6) are both characterized by the presence of orthopyroxene; however, the two mineral assemblages differ substantially.

The low-$SiO_2$ (64.93 wt%) LH5-1 trachyte is characterized by euhedral to subhedral phenocrysts of plagioclase (ca. 10-15 vol%), clinopyroxene (ca. 10 vol%), orthopyroxene (ca. 10 vol%), olivine (ca. 5-10 vol%) and sanidine (< 10 vol%), in a hypohyaline microcrystalline groundmass made of (in order of

abundance) sanidine, orthopyroxene, clinopyroxene, rare plagioclase, rare olivine, rare opaque minerals and very rare glass. All phenocrysts are generally unzoned. Mafic phenocrysts with homogeneous cores surrounded by normal concordant monotonous zoning are also observed. Rare major plagioclase (up to 1.5 mm in length) phenocrysts present patchy cores and normal concordant step zoning textures at rims. Rare major clinopyroxene (up to 1.0 mm in length) unzoned homogeneous phenocrysts show inclusions of olivine + magnetite.

The high $SiO_2$ (67.58 wt%) LH6 trachyte is made up of sanidine (ca. 10 vol%), plagioclase (ca. 5-10 vol%) and orthopyroxene (ca. 10 vol%) phenocrysts in a fine grained trachytic mesostasis. Only major plagioclase and orthopyroxene phenocrysts show core-rim zoning textures with homogeneous cores associated either with normal monotonous zoning or normal low-amplitude oscillatory zoning at rims. Dimension of phenocrysts are comparable to those of LH5-1 trachyte.

**6.MINERAL CHEMISTRY**

Mineral compositions as obtained from electron microprobe analyses and mineral formulae for mineral assemblages of LHPCS lavas are presented in Supplementary Tables 1, 2, 3, 4 and 5 (for feldspar, clinopyroxene, olivine, orthopyroxene and spinel and opaque minerals, respectively).

**6.1 Feldspar**

In basaltic rocks (Fig. 5a), feldspars are predominantly plagioclase. Plagioclase phenocrysts show anorthitic ($X_{An}$ = 59-81%, average 67%) cores and normally zoned ($X_{An}$ = 42-59%, average 53%) rims. Orthoclase component ($X_{Or}$) is always less than 2%. Plagioclase microlites in the groundmass show andesine ($X_{An}$ = 19-60%) composition, with $X_{Or}$ ranging 1-7%. Alkali-feldspars occur as both rare anorthoclase phenocrysts ($Ab_{60}Or_{37}$), and microlites in groundmass ($Ab_{62-79}Or_{9-35}An_{0-13}$).

Plagioclase from trachyandesites (Fig. 5b) have anorthite-rich ($X_{An}$ = 45-87%, average 67%) cores and normally zoned ($X_{An}$ = 27-69%, average 48%) rims. Cores with $X_{An}$ in the range 72-87% are observed in all major phenocrysts. Plagioclase core compositions are comparable to those of basalts. The $X_{Or}$ ranges 1-8%. Plagioclase microlites in groundmass show andesine ($X_{An}$ = 29-63%) composition with $X_{Or}$ always less than 10%. Alkali-feldspars occur as (i) anorthoclase ($Ab_{59-68}Or_{11-30}$) and sanidine ($Ab_{49-50}Or_{43-48}$) phenocrysts, and (ii) anorthoclase ($Ab_{49-70}Or_{15-38}$) and sanidine ($Ab_{38-48}Or_{47-61}$) microlites in groundmass.

Trachytes (Fig. 5c) show generally unzoned plagioclase phenocrysts with oligoclase-andesine ($X_{An}$ = 26-45%) composition. Rare An-rich ($X_{An}$= 52-70%) cores are reported from major plagioclase phenocrysts in the low-silica trachyte LH5-1. The $X_{Or}$ is always less than 8%. Plagioclase microlites in groundmass are rare, with Ab-rich ($An_{21-30}Ab_{66-69}Or_{4-10}$) composition. Alkali-feldspar is represented by anorthoclase as phenocrysts ($Ab_{65-66}Or_{20-21}$) and groundmass microlites ($Ab_{64-66}Or_{21-24}$).

**6.2 Clinopyroxene**

Apart from the LH18 basalt and LH6 trachyte, clinopyroxene, is the most common mafic mineral recognized in studied samples. It occurs generally as single crystal (Fig. 4e). However rare crystals showing growth mantle textures are locally reported in trachyandesites. Very rare phenocrysts in trachyandesites show patchy cores. Major clinopyroxene phenocrysts in trachyandesites and trachytes contain inclusions (Fig. 4e, g) of olivine, magnetite and plagioclase.

Polarized light microscopy coupled with BSE images and chemical investigations highlighted the presence of unzoned (Fig. 4g) and zoned (homogeneous cores surrounded by low-amplitude oscillatory zoning or normal monotonous zoning or normal step zoning textures at rims; e.g., Fig. 4e) clinopyroxene phenocrysts. Very rare phenocrysts showing growth mantle textures at rim are reported. No evidence of resorption/dissolution textures are observed in studied samples.

The Cpx population, based on textural observations and mineral chemistry, (Fig. 6 a-f) can be classified in five major categories: (i) Cpx1 cluster is represented by homogeneous cores of all zoned phenocrysts in basalts; (ii) Cpx2 population is represented by homogeneous cores of all zoned phenocrysts from trachyandesites and trachytes; (iii) Cpx3 group represent both the unzoned phenocrysts in all studied samples and the rims (low-amplitude oscillatory, normal monotonous and normal step zoning) of cpx1 and cpx2 phenocrysts from all studied samples; (iv) Cpx4 population is constituted by microlites and microphenocrystals in groundmass from all analyzed samples; and (v) Cpx5 cluster collects together the emerald-green euhedral to subhedral microlites in groundmass of intra-caldera basalts (LH5-2, LH27-1) and rare normal monotonous zoning rims of major clinopyroxene (Cpx2) phenocrysts from few trachyandesites (LH15, LH17, LH26-2).

The Cpx1 shows Mg# of 45-75, Ca 0.78-0.90 apfu, Q+J 1.84-1.94 and J/(J+Q) 0.03-0.06, and it can be classified as Ti-rich augite ($Wo_{41-48}En_{25-42}Fs_{14-32}$).

The Cpx2 shows a Mg# of 59-84, Ca 0.20-0.92 apfu, Q+J 1.77-1.95 and J/(J+Q) 0.01-0.06, and it can be classified as diopside-rich augite ($Wo_{11-48}En_{36-64}Fs_{9-32}$).

The Cpx3 shows Mg#: 20-86, Ca 0.27-0.97 apfu, Q+J 1.57-1.98 and J/(J+Q) 0.01-0.07, and it can be classified as diopside-rich augite ($Wo_{12-49}En_{14-57}Fs_{8-62}$). The composition of Cpx3 partially overlaps those of Cpx1 and Cpx2 groups, as it would be expected for phenocrysts with homogeneous cores (i.e. Cpx1 and Cpx2) and the respective low-amplitude oscillatory zoning or normal monotonous zoning rims (Cpx3) (e.g., Streck, 2008).

The Cpx4 shows Mg# of 31-81, Ca 0.24-0.87 apfu, Q+J 1.87-1.97 and J/(J+Q) 0.01-0.06, corresponding to diopside-rich augite ($Wo_{12-46}En_{18-60}Fs_{11-38}$). The composition of Cpx4 partly overlaps that of Cpx3, however their textural characteristics are completely different.

The Cpx5 differs from previous pyroxenes, with a large spread in Mg# ranging 5-73, Ca 0.03-0.83 apfu, Q+J 1.51-2.07 and J/(J+Q) 0.07-0.89. The Cpx5 can be classified as Aegirine-Augite (Na< 0.3 apfu, XAeg< 0.30; with XAeg= Na apfu if Na < $Fe^{3+\ Tot}$, XAeg= $Fe^{3+\ Tot}$ apfu if Na > $Fe^{3+Tot}$) to Aegirines (Na= 0.68-0.88 apfu,

XAeg= 0.40-0.88). Cpx5 clinopyroxenes are generally Ti-enriched (TiO$_2$ up to 2.8 wt%, Ti up to 0.08 apfu)
and straddle the Q+J=2 line defining the boundary for "normal" pyroxenes (Morimoto, 1989), thus
indicating the presence of a NaR$^{2+}_{0.5}$Ti$^{4+}_{0.5}$Si$_2$O$_6$ component (Morimoto, 1988, 1989; Huraiova et al., 2017)
(Fig. 6c).
The compositional variation of clinopyroxenes can be summarized in the Na vs. Ti diagram (Fig. 6e-f).
Interestingly, Augite-rich (Cpx1, Cpx2, Cpx3 and Cpx4) clinopyroxenes generally show positive correlation
and linear distribution characterized by a progressive Ti- and Na-depletion, from Ti-Augite cores (Cpx1) in
basalts to DiHd-rich Augite (Cpx3, Cpx4) specimens in trachytes. The Cpx5, with Aegirine-Augite and
Aegirine, diverges from this trend. It shows a negative correlation characterized by a progressive
enrichment of Na content, with respect to a general Ti-depletion. Aegirine enrichment could be diagnostic
of ferric iron (Fe3+) content increasing during the magmatic differentiation, whereas the diopside-
hedenbergite enrichment could testify an increase of ferrous iron (Fe2+) in magma (e.g. Huraiova et al.,

408  2017).


**6.3 Olivine**
Olivine is found in all analyzed samples, except for LH27-2 trachyandesite and LH6 trachyte. It consists of
idiomorphic (Fig. 4 a, c) to skeletal (e.g., Donaldson, 1974; Fowler et al., 2002; Faure et al., 2003; Welsch et
al., 2013) (Fig. 4b) phenocrysts, and microlites in the groundmass (Fig. 4h). Olivine crystals, both
phenocrysts and microcrystals, show homogeneous cores with concordant normal monotonous zoning
outer rims. In basalts, olivine shows a continuous compositional range (Fig. 7a) from Fo$_{86}$Fa$_{14}$Mtc$_0$Tep$_0$
(phenocryst in LH5-2 basalt) to Fo$_{05}$Fa$_{91}$Mtc$_1$Tep$_3$ (groundmass microlites in LH27-1 basalt). The highest
MnO (up to 1.7 wt%) values are systematically found in Fe-rich olivine microlites in basalts. Low
monticellite concentration (CaO always < 1.0 wt%) in LHPCS samples is typical for magmatic olivine (i.e.,
Melluso et al., 2014). CaO content positively correlates with the fayalite (FeO) compound (Fig. 5a). Together
with the Mg#, the CaO content allows to discriminate olivine phenocrysts in three coherent compositional
clusters: i) olivine from basalts, with Mg#= 79-87 and CaO= 0.21-0.73 wt%, ii) olivine from trachyandesites,
with Mg#= 67-80 and CaO= 0.08-0.43 wt%, and iii) olivine from trachytes with Mg#= 58-63 and CaO= 0.16-
0.42 wt%. A minor number of analyzed phenocrysts in basalts show Cr$_2$O$_3$ content in the range ca. 0.05-0.07
wt%. It is, instead, below detection limit for almost all analyzed olivine crystals in LHPCS lavas. A minor
cluster of peridote Mg-olivine (Fo$_{99}$Fa$_1$) xenocrysts, characterized by disequilibrium textures (resorption
patterns) at rim, have been identified in LH26-1 trachyandesite lava.

**6.4 Orthopyroxene**
Orthopyroxene occurs in most of the LHPCS trachyandesite (Fig. 4f) and trachyte samples (Fig. 4 g-h).
Orthopyroxene phenocrysts are generally unzoned with homogeneous textures. In trachyandesites, they

show intermediate ($En_{41-83}Fs_{14-55}Wo_{2-10}$) compositions (Fig. 7b), with Mg# of 43-86, $Al_2O_3$ up to 2.12 wt%, $TiO_2$ 0.08-1.33 wt%, and CaO 1.20-4.72 wt%. Similar compositions ($En_{62-79}Fs_{18-33}Wo_{3-7}$) have been obtained for microlites in groundmass (Fig. 7b) with Mg# of 65-81, $Al_2O_3$ 0.48-1.53 wt%, $TiO_2$ 0.21-0.60 wt% and CaO 1.35-3.49 wt%.

In trachytes, orthopyroxene phenocrysts present Mg# ranging 59-65, with low $Al_2O_3$ (0.18-0.73 wt%), low $TiO_2$ (0.11-0.32 wt%) and CaO (0.81-1.88 wt%), corresponding to Fe-rich composition ($En_{56-63}Fs_{34-39}Wo_{2-4}$) with a minor Ca-Cpx substitution (Fig. 7b). Orthopyroxene microlites in groundmass (Fig. 7b) show comparable hypersthene ($En_{46-60}Fs_{35-45}Wo_{3-7}$) composition with Mg# of 50-63, $Al_2O_3$ 0.25-0.82 wt%, $TiO_2$ 0.19-0.31 wt% and CaO 1.32-3.27 wt%.

The compositional variation of orthopyroxyene is summarized in $Al^{Tot}$ vs. Mg# diagram (Fig. 7c). Orthopyroxene crystals from trachyandesites are characterized by higher content of Al (apfu) and higher Mg#, whereas those from trachytes are richer in ferrous iron (lower Mg# values) and in manganese (Mn up to 0.04 apfu).

**6.5 Spinel and Opaque Minerals**

Basalts show a diversified set of opaque minerals. Phenocrysts are (in order of abundance): i) Al-spinel ($TiO_2$ 0.58-1.00 wt%; Mg# 58-71; Cr# 21-30, with [Cr#= 100 Cr/(Cr+Al)]), ii) Ti-magnetite ($TiO_2$ 1.83-21.58 wt%; MgO 0.06-2.19 wt%; MnO 0.44-0.63 wt%) and (iii) ilmenite (MgO up to 2.18 wt%). Groundmass is characterized by Fe-Ti oxides (ca. 20-30 µm in diameter; Fig. 4f) as ilmenite (MgO 0.27-1.50 wt%) and Ti-magnetite (MgO 0.18-1.89 wt%). Cr-spinels ($TiO_2$ 3.37-8.55 wt%; Mg# 14-28; Cr# 62-72) are found just as inclusions, up to 200 µm in diameter (Fig. 4a), in larger Mg-rich olivine phenocrysts.

Trachyandesites are characterized by phenocrysts of Ti-magnetite (MgO 0.07-3.84 wt%), ilmenite (MgO 1.11-4.79 wt%) and rare rutile (MgO 0.47 wt%). Groundmass microcrystals (ca. 20-30 µm in diameter) show a comparable composition with Ti-magnetite (MgO 0.33-3.77 wt%), ilmenite (MgO 0.33-4.79 wt%) and rare rutile (MgO < 0.05 wt%). Similar to basalts, Cr-spinels ($TiO_2$ 6.09-6.47 wt%; Mg# 19-21; Cr# 65-68) are found only as inclusions (100-200 µm in diameter) in major Mg-rich olivine phenocrysts.

In trachytes, Fe-Ti oxides show euhedral to subhedral habit and, based on chemistry, they are ilmenite (MgO 2.06-3.31 wt%) and Ti-magnetite (MgO 1.41-5.47 wt%). Phenocrysts (up to 50-100 µm in diameter; Fig. 4g) and groundmass microcrystals (ca. 15-20 µm in diameter) show the same compositions.

**7. MINERAL-LIQUID THERMOBAROMETRY**

In order to define the thermobaric (T-P) environmental conditions of the magmatic feeding system of the LHPCS, we integrate thermobarometry models based on olivine (Beattie, 1993; Putirka et al., 2007; Putirka, 2008), orthopyroxene (Putirka, 2008), plagioclase (Putirka, 2005b; Putirka, 2008), alkali-feldspar (Putirka, 2008) and clinopyroxene (Putirka et al., 1996, 2003; Putirka, 2008; Masotta et al., 2013) chemistry. Due to

the paucity/absence of glasses, we assume the whole rock composition as representative of the original liquid (or nominal melt) in equilibrium with phenocrysts (Putirka, 1997, 2008; Mordick and Glazner, 2006; Aulinas et al., 2010; Dahren et al., 2012; Barker et al., 2015). We are aware that such a procedure put the focus on early steps of the crystallization history, characterized by high melt/crystal ratios. Relatively late melt compositions, related to the solidification of the groundmass, are not present or can simply not be analyzed. Thermobarometric calculations were developed after the application of mineral-melt equilibrium filters and considering pre-eruptive $H_2O^{liq}$ values obtained through the plagioclase-liquid hygrometer model (eq. 25b in Putirka, 2008). Plagioclase-liquid thermometry and barometry were calculated using eq. (24a) and eq. (25a), respectively, of Putirka (2008), mainly based on the Ca/Na distribution between melt and Pl. Alkali-feldspar-liquid thermometry was calculated considering the K-Na exchange, applying eq. (24b) in Putirka (2008). Olivine-liquid equilibrium thermometry was calculated integrating the models of Beattie (1993) and Herzberg and O'Hara (2002) with the thermometric eq. (2) in Putirka et al. (2007). Orthopyroxene-liquid thermometry was calculated by Fe-Mg partitioning following the model of Beattie (1993; in the revised form [eq. 28a] in Putirka, 2008). Barometry model of Wood (1974) based on the Na and Al content in Opx, in the revised form [eq. 29a] in Putirka (2008), was applied.

Clinopyroxene-liquid thermometry and barometry, for diopside-augite pyroxenes in basalts and trachyandesites (groups Cpx1, Cpx2, Cpx3, Cpx4), were calculated by the application of the Jd-DiHd exchange thermometer (Putirka et al, 1996, 2003) using [eq. 33] in Putirka (2008) and the Al-partitioning barometric model [eq. 32c] in Putirka (2008). Clinopyroxene-liquid thermometry and barometry, for diopside-augite pyroxenes in trachytes (groups Cpx3 and Cpx4), were calculated by the application of the Jd-DiHd exchange thermometer (Putirka et al, 1996, 2003; Putirka, 2008) recalibrated for alkaline differentiated magmas using [eqn. Talk33] and [Eqn. Palk 2012], respectively, in Masotta et al. (2013). Clinopyroxene-liquid thermometry and barometry, for augite-aegirine pyroxenes (Cpx5), were calculated integrating [eq. 33] and [eq. 32c] in Putirka (2008) with equations [Eqn. Talk2012] and [Eqn. Palk 2012] in Masotta et al. (2013). Results of mineral-melt equilibrium tests (Figs. 8, 9, 10), hygrometry calculations (Fig. 10) and geothermometric estimates are presented contextually in supplementary mineral chemistry tables. Summary of the thermobarometry estimates are reported in a Pressure-Temperature diagram (Fig. 11).

**7.1 Test for Mineral-Melt Equilibrium**

Prerequisite for the application of mineral-liquid thermobarometry models based on mineral-melt equilibrium conditions is to test and verify that the mineral and the chosen liquid compositions represent chemical equilibrium pairs (e.g., Putirka, 2008; Keiding and Sigmarsson, 2012). Petrographic investigations (i.e., polarized light and BSE imaging) and calculation of mineral-liquid partition coefficients were integrated with the aim to select only mineral specimens at equilibrium with the hosting melt (e.g., Putirka, 2008; Keiding and Sigmarsson, 2012).

The predominant euhedral to subhedral habit of crystals is generally considered as an evidence of equilibrium with the surrounding melt (e.g., Keiding and Sigmarsson, 2012). However, crucial for mineral-liquid thermo-barometric modeling, it is the use of phenocrysts with strongly zoned textures (patchy-, sector-, reverse-, coarse banding oscillatory-zoning), or with disequilibrium textures (resorption patterns, dissolution surfaces, reaction rims and mineral mantles/clots) (e.g. Ginibre et al., 2002; Streck, 2008). These textures imply that core(s) and rim(s), or different portions of the same grain, crystallized and reacted in an evolving liquid with progressively different compositions (e.g., Mordick and Glazner, 2006; Putirka, 2008; Keiding and Sigmarsson, 2012). As defined by Streck (2008), when crystals are complexly zoned, it can be difficult to find criteria to be used for evaluation of crystal populations and their equilibrium with respective hosting melt. However, it is not the case of the LHPCS studied samples, where phenocryst assemblages generally do not show disequilibrium patterns or complexly zoned textures (e.g., Ginibre et al., 2002; Streck, 2008). All microprobe analyses related to those rare crystals presenting morphological evidence of disequilibrium texture, such as patchy zoning, were discarded.

Then, the mineral-liquid equilibria between liquid and previous selected minerals, were investigated using: (i) the Fe-Mg exchange coefficient, (ii) the An-Ab partitioning coefficient, and (iii) the comparison between observed and predicted normative components of minerals.

The partitioning of Fe-Mg between mineral and liquid is known as Fe-Mg exchange coefficient, or $K_D^{Min-Liq}$(Fe-Mg) (defined as $K_D^{Min-Liq}$(Fe-Mg) = [MgO$^{Liq}$FeO$^{Min}$]/[MgO$^{Min}$FeO$^{Liq}$], where *Liq* is the liquid composition, *Min* is the mineral composition and MgO and FeO are molar fractions; Roeder and Emslie, 1970; Langmuir and Hanson, 1981; Putirka, 2005a; Putirka, 2008). It is used here to test the equilibrium between mafic minerals (olivine, orthopyroxene and clinopyroxene) and liquid (e.g., Maclennan et al., 2001; Putirka, 2008; Stroncik et al., 2009; Aulinas et al., 2010; Keiding and Sigmarsson, 2012; Melluso et al., 2014; Feng and Zhu, 2018).

We calculated $K_D^{Min-Liq}$(Fe-Mg) values using (i) equation (17) in Putirka (2008) for Ol and Opx; (ii) temperature-dependent equation (35) in Putirka (2008) for diopsidic-augitic Cpx in basalts and trachyandesites; and iii) the Na-corrected equation (35a) in Masotta et al. (2013) for Na-rich Cpx5 group and for all Cpx from LH5-1 and LH6-1 trachytes. The calculated $K_D^{Min-Liq}$(Fe-Mg) values for olivine and orthopyroxene are plotted in a Rhodes's diagram (Dungan et al., 1978; Rhodes et al., 1979; Putirka, 2005; Putirka, 2008) to graphically test the equilibrium between Ol (Fig. 8a) or Opx (Fig. 8b) and the respective hosting melts (Liq). Furthermore, the Rhodes's diagram is useful to recognize: (i) presence of xenocrystals and/or antecrystals; (ii) late or groundmass crystallization; (iii) crystal removal (decrease of Mg#$^{Liq}$ only); and (iv) closed system crystallization (decrease of Mg#$^{Min}$ only) by deviations of the measured compositions from the expected ones (Rhodes et al., 1979; Putirka, 2008; Melluso et al., 2014).

The calculation of $K_D^{Cpx-Liq}$(Fe-Mg) does not consider variations of Ca and Al contents in Cpx (Rhodes et al., 1979; Putirka, 1999, 2005b, 2008). Therefore, a further equilibrium test was achieved through the

comparison of analysed Cpx compositions (as expressed by the components EnFs, DiHd and CaTs; where CaTs is Ca-Tschermak) with component contents predicted from melt composition (e.g., Putirka, 2008; Mollo et al., 2010; Jeffery et al., 2013; Barker et al., 2015; Ellis et al., 2017). Normative components of Cpx were calculated following the scheme proposed in Putirka et al., (1996) and Putirka (2008). Calculation of Cpx components based on melt composition was performed using equations (eq 3.1a) for DiHd, (eq 3.2) for EnFs and (eq 3.4) for CaTs in Putirka (1999). A graphical presentation (e.g., Jeffery et al., 2013; Barker et al., 2015) of this test is shown in Figure 9.

The partitioning of An-Ab between mineral and liquid is known as An-Ab exchange coefficient, or $K_D^{Pl-Liq}$(An-Ab) (defined as $K_D^{Pl-Liq}$(An-Ab) = $[XAb^{Pl}XAlO_{1.5}^{Liq}XCaO^{Liq}]/[XAn^{Pl}XNaO_{0.5}^{Liq}XSiO_2^{Liq}]$, where *Liq* is the liquid composition, *Pl* is the plagioclase composition and all components are in molar fractions; Carmichael et al., 1977; Holland and Powell, 1992; Putirka et al., 2007; Putirka, 2008; Lange et al., 2009; Keiding and Sigmarsson, 2012; Jeffery et al., 2013; Barker et al., 2015; Waters and Lange, 2015). Figure 10 presents a comparison of measured composition of plagioclase with that calculated from the melt composition, using the thermodynamic model eq (31) in Namur et al. (2012). A similar test can be applied for alkali-feldspars (Putirka, 2008).

In summary, we accept: (i) Ol with $K_D^{Ol-Liq}$(Fe-Mg)= 0.30 ± 0.06 (Roeder and Emslie, 1970; Putirka, 2005a; Putirka, 2008 and references therein) (Fig. 8a); (ii) Opx with $K_D^{Opx-Liq}$(Fe-Mg)= 0.29 ± 0.06 (Putirka, 2008 and references therein) (Fig. 8b); (iii) Cpx with $K_D^{Cpx-Liq}$(Fe-Mg)= 0.28 ± 0.08 (Putirka, 2008) and verifying the one-to-one (± 0.1) relationship between predicted vs. observed normative components (EnFs, DiHd and CaTs) for at least two of the monitored components (Fig. 9); (iv) Pl with $K_D^{Pl-Liq}$(An-Ab)= 0.27 ± 0.11 for T > 1050°C and $K_D^{Pl-Liq}$(An-Ab)= 0.10 ± 0.05 for T < 1050°C (Putirka, 2008) or falling within ± 0.1 of the one-to-one relationship between predicted vs observed An components (Fig. 10); and (v) Afs with $K_D^{Afs-Liq}$(An-Ab)= 0.27 ± 0.18 (Putirka, 2008). All mineral-liquid pairs exceeding the accepted exchange coefficient values for Ol, Cpx, Opx and Fsp were discarded for thermobarometric analyses.

### 7.2 Pre-eruptive $H_2O^{Liq}$ content estimates

Thermobarometric models for volcanic systems require an initial estimate of the pre-eruptive water concentration (wt%) in melt ($H_2O^{Liq}$). It was determined in this work by using the plagioclase-liquid hygrometer model [eq. 25b] in Putirka (2008). Hygrometry calculations were produced after the application of plagioclase-liquid equilibrium filters. The calculated pre-eruptive $H_2O^{Liq}$ wt% values (±1σ standard deviation of the weighted mean) are plotted as isolines in Fig. 10. The hygrometer of Putirka (2008) indicates (Fig. 10): (i) $H_2O^{Liq}$ negative values in basalts, from -0.20 to -0.40 wt%, with a weighted mean of -0.37 ± 0.20 wt% (MSWD= 0.0026; n= 95); (ii) trachyandesites pre-eruptive water content in the range $H_2O^{Liq}$: 0 – 1.40 wt% (weighted mean of 0.57 ± 0.13 wt%, MSWD= 0.13, n= 245); and (iii) trachytes with the highest water concentration ($H_2O^{Liq}$: 1.40 – 1.90 wt%; weighted mean of 1.46 ± 0.32 wt%, MSWD= 0.059, n=

37). Following the approach of Keiding and Sigmarsson (2012), negative values in basalts are interpreted as
anhydrous melt compositions. Coherently with the existing literature (e.g., Webster et al., 1999), the
anhydrous character is then assumed as a $H_2O^{Liq}$ < 1 wt% content.
Application of plagioclase-liquid hygrometer model (Putirka, 2008) defines anhydrous environment for
pressure-temperature calculations in LHPCS basalts. Whereas hydrous conditions are required for evolved
LHPCS melts and in particular for trachytic lavas, where the effect of 1 wt% $H_2O$ is expected to generate a
temperature decrease of ca. -40 °C and a pressure increase of ca. + 1.0 kbar in geothermometers and
geobarometers, respectively (Putirka, 2008; Keiding and Sigmarsson, 2012).
On contrary, existing studies (e.g., Kushiro, 1969; Sisson and Grove; 1993; Yang et al., 1996; Putirka, 2005a,
2005b, 2008; Kelley and Barton, 2008; Keiding and Sigmarsson, 2012) demonstrated a negligible effect of
water for basaltic and intermediate melts showing $H_2O^{Liq}$ ranging 0 – 1 wt%.

**7.3 Thermobarometry Results**
***7.3.1 Basalts***
When applied to phenocryst cores, the Pl-Liq thermobarometry (Fig. 11a-c) show that all LHPCS basaltic
materials have magmatic anhydrous T in the range 1230-1266 °C (weighted mean of 1250 ± 5 °C, ±1$\sigma$
standard deviation of the weighted mean, MSWD= 0.112, n= 95). Pressure estimates are in the range 6.5-
8.7 kbar (weighted mean of 7.9 ± 1.1 kbar, ±1$\sigma$ standard deviation of the weighted mean, MSWD= 0.024,
n= 28) for LH18 Ol-basalts, and 7.2-10.3 kbar (weighted mean of 9.2 ± 0.7 kbar (±1$\sigma$), MSWD= 0.064, n= 67)
for LH5.2 and LH27.1 Ol-Cpx-basalt. Olivine-melt equilibrium (Fig. 11a-c), for the olivine compositional
range of Fo 80-85%, yields T window of 1240-1297 ± 27 °C (±1$\sigma$), consistent with the results obtained with
Pl-liq thermometry. The Cpx-thermobarometry (Fig. 11a, c), for both Cpx1 (phenocryst cores) and Cpx3
(phenocryst rims and unzoned phenocrysts), provides temperature of 1006-1209 °C (weighted mean of
1124 ± 12 °C (±1$\sigma$), MSWD= 3.4, n= 82). Pressure ranges 3.1-11.5 kbar (weighted mean of 7.6 ± 0.8 kbar
(±1$\sigma$), MSWD= 2.7, n= 36) for Cpx1, and 2.5-7.7 kbar (weighted mean of 4.0 ± 0.8 kbar (±1$\sigma$), MSWD= 0.63,
n= 14) for Cpx3. Thermobaric estimates for Cpx4 (microlites in groundmass) indicate shallow conditions (0.3
– 3.0; weighted mean of 1.6 ± 1.2 kbar (±1$\sigma$), MSWD= 0.38, n= 6) for temperatures (1006-1123 °C;
weighted mean of 1060 ± 54 °C (±1$\sigma$), MSWD= 2.9, n= 6) comparable to those obtained for Cpx1 and Cpx3.
Higher temperature estimates (1067-1221 °C; weighted mean of 1157 ± 53 °C (±1$\sigma$), MSWD= 2.4, n= 7) at
low-pressure (0.4-4.7; weighted mean of 2.9 ± 1.1 kbar (±1$\sigma$), MSWD= 0.83, n= 7) are instead obtained for
a limited number of Cpx5 (aegirine-rich) compositions (Fig. 11a, c).

***7.3.2 Trachyandesites***
Based on the Opx- presence/absence criterion, two populations of trachyandesites have been
discriminated in this study.

Opx-free trachyandesites LH15 and LH21-2 (El Limón and Sarabia lava flows, respectively) are characterized by i) plagioclase phenocryst cores crystallized at T of 1190-1263 °C (weighted mean of 1248 ± 7 °C (±1$\sigma$), MSWD= 1.09, n= 39) and P of 4.8-9.4 kbar (weighted mean of 7.7 ± 0.9 kbar (±1$\sigma$), MSWD= 0.14, n= 39); ii) comparable temperature (1193-1263 °C; weighted mean of 1227 ± 37 °C (±1$\sigma$), MSWD= 2.3, n= 6) and pressure (6.7-9.6 kbar, mean value of 7.8 ± 2.4 kbar (±1$\sigma$), MSWD= 0.101, n= 6) obtained for rare phenocryst rims and microlites at equilibrium; iii) olivine-melt equilibrium (with Fo: 75-80%) showing a T window of 1030-1055 ± 27 °C (±1$\sigma$); iv) rare Cpx2 (clinopyroxene phenocryst cores) showing equilibrium with melt and yielding T 1061-1239 °C (weighted mean of 1116 ± 29 °C (±1$\sigma$), MSWD= 2.3, n= 12) and P ca. 2.9 -8.3 kbar (weighted mean of 5.2 ± 1.2 kbar (±1$\sigma$), MSWD= 1.5, n= 12); v) Cpx3 (rims of and unzoned phenocrysts) showing equilibrium with melt and yielding thermobarometric results (T 938-1139 °C, with weighted mean of 1074 ± 15 °C (±1$\sigma$), MSWD= 1.9, n= 32; and P 1.0-4.4 kbar with weighted mean of 2.8 ± 0.5 kbar (±1$\sigma$), MSWD= 0.22, n= 32); vi) Cpx4 (groundmass microcrystals) compositions indicating, with respect to Cpx3, comparable temperatures (1026-1127 °C, with weighted mean of 1059 ± 16 °C (±1$\sigma$), MSWD= 0.71, n= 14) at lower pressure conditions (0.3 – 3.6 kbar with weighted mean of 1.4 ± 0.8 kbar (±1$\sigma$), MSWD= 0.35, n= 14). The unique Cpx5-liquid pair at equilibrium yielded P-T conditions of 5.6 ± 1.5 kbar and 1122 ± 30 °C.

Thermobarometric estimates (Fig. 11a, d) for Opx-bearing trachyandesites (LH4, LH13, LH17; LH26-1; LH26-2; LH27-2) show overlapping P-T conditions for plagioclase populations with: i) phenocryst cores crystallizing at T: 1145-1228 °C (weighted mean of 1187 ± 4 °C (±1$\sigma$), MSWD= 1.17, n= 166) and P: 4.1-7.7 kbar (weighted mean of 5.8 ± 0.5 kbar (±1$\sigma$), MSWD= 0.059, n= 166), and ii) phenocryst rims and microcrystals forming at T: 1140-1224 °C (weighted mean of 1168 ± 8 °C (±1$\sigma$), MSWD= 0.92, n= 34) and P: 4.4-8.5 kbar (weighted mean of 6.4 ± 1.0 kbar (±1$\sigma$), MSWD= 0.14, n= 34). Lower temperatures (1050-1090 ± 27 °C (±1$\sigma$)) are obtained using olivine (Fo 70-80%) – liquid equilibrium model.

Thermobarometers applied to pyroxenes indicate: i) Cpx2 (phenocryst cores) crystallizing at T: 979-1204 °C (weighted mean of 1060 ± 8 °C (±1$\sigma$), MSWD= 1.8, n= 101) and P: 3.4 -11.5 kbar (weighted mean of 7.0 ± 0.3 kbar (±1$\sigma$), MSWD= 0.94, n= 101), ii) Cpx3 crystallizing at T: 959-1106 °C (weighted mean of 1026 ± 6 °C (±1$\sigma$), MSWD= 1.3, n= 145) and P: 1.2-6.9 kbar (weighted mean of 4.3 ± 0.2 kbar (±1$\sigma$), MSWD= 0.72, n= 145), iii) rare Cpx4 showing general equilibrium with melt and forming at P-T conditions of : 920-1123 °C (weighted mean of 1020 ± 21 °C (±1$\sigma$), MSWD= 2.7, n= 24) and P: 0.1-3.4 kbar (weighted mean of 1.8 ± 0.6 kbar (±1$\sigma$), MSWD= 0.56, n= 24), and iv) Opx yielding crystallization conditions, for both phenocrysts and microlites, of T 1048-1123 °C (weighted mean of 1078 ± 5 °C (±1$\sigma$), MSWD= 0.24, n= 129) and P: 0 -2.8 kbar (weighted mean of 1.1 ± 0.6 kbar (±1$\sigma$), MSWD= 0.057, n= 84). In all trachyandesites samples, temperatures obtained through Ol-Liq model and Cpx-Liq model are comparable (Fig. 11a, d), whereas the Pl-Liq model shows higher T values. These results can be interpreted as an earlier plagioclase crystallization

with respect to olivine and clinopyroxene. Orthopyroxene (Opx) can be considered a tracer of
trachyandesitic magma stagnations at shallow depths, since the invariably lower pressure values obtained
by Opx-liquid barometer.

### 7.3.3 Trachytes

Magmatic P-T conditions (Fig. 11a, e) of trachytic (LH5.1 and LH6) melts are defined by: i) plagioclase
crystallization at T: 1050-1094 °C (weighted mean of 1069 ± 6 °C (±1$\sigma$), MSWD= 0.39, n= 37) and P: 4.7-9.0
kbar (weighted mean of 6.5 ± 1.0 kbar (±1$\sigma$), MSWD= 0.20, n= 37), ii) olivine (Fo55-65%) – liquid regression
indicating olivine crystallization at 900-920 ± 27 °C (±1$\sigma$), iii) clinopyroxene crystallization, both phenocrysts
(Cpx3) and groundmass (Cpx4), at temperature of ca. 955 °C (weighted mean of 956 ± 14 °C (±1$\sigma$), MSWD=
0.00056, n= 17) and very shallow-depth conditions (P weighted means of 2.3 ± 0.9 kbar (±1$\sigma$), MSWD=
0.047, n= 10 and 1.6 ± 1.1 kbar (±1$\sigma$), MSWD= 0.04, n= 7; for Cpx3 and Cpx4, respectively). Shallow-depth
conditions are also obtained for orthopyroxene crystallization with T: 960-1006 °C (weighted mean of 990 ±
7 °C (±1$\sigma$), MSWD= 0.28, n= 49) and P: 0.2-3.6 kbar (weighted mean of 1.6 ± 0.9 kbar (±1$\sigma$), MSWD= 0.101,
n= 35). The alkali-feldspar-liquid thermometer provided temperature estimates always <500°C, here
interpreted as feldspar re-equilibration in subsolvus/subsolidus post-eruptive conditions (Nekvasil, 1992;
Brown and Parsons, 1994; Plumper and Putnis, 2009; Kontonicas-Charos et al., 2017; Latutrie et al., 2017).
Interestingly, temperatures obtained through Pl-Liq model are higher than those obtained with Ol-Liq, Cpx-
Liq and Opx-Liq, suggesting an earlier crystallization of plagioclase with respect to mafic minerals.
Moreover, the Pl-Liq models indicate thermobaric estimates comparable to those obtained for
trachyandesitic rocks.


**8. DISCUSSION**

**8.1 Major-elements mass balance modeling**

Based on the textural evidence documenting: (i) Cpx-bearing basalts being mainly characterized by euhedral olivine and plagioclase and subhedral-anhedral clinopyroxene, indicating crystallization of olivine and plagioclase prior to clinopyroxene (e.g., Bindeman and Bailey, 1999); and (ii) all LHPCS volcanic rocks not showing disequilibrium textures (such as fine-sieve textures, resorption surface, crystal clots, disequilibrium growth-mantel, reverse zoning, reaction-rims, breakdown mantle and dissolution; e.g., Streck, 2008) typical of AFC-mixing processes, we suggest that the studied LHPCS volcanic rocks represent cogenetic melts, belonging to the same line of descent, excluding major mass-change due to assimilation and mixing (AFC-mixing) processes. In order to test this hypothesis, we applied fractional crystallization (FC) modeling (e.g., White et al., 2009; Moghadam et al., 2016; Lucci et al., 2016). The FC-modeling is focused on these hypotheses: (i) direct cogenetic relationship between all LHPCS basalts, and (ii) common genesis for all LHPCS trachyandesites and trachytes through differentiation via fractional crystallization starting from the same basaltic parental melt.

Major-element mass balance models (e.g., Bryan et al., 1969) can be used to test and define relative proportion of phases involved in Rayleigh fractional crystallization (RFC, Daughter = Parent - fractionating assemblage) and crystal accumulation (Cumulate = Melt + accumulated assemblage) hypotheses (e.g., White et al., 2009; Moghadam et al., 2016; Lucci et al., 2016).

If Parent melt (for RFC) or Cumulate (for crystal accumulation) compositions are assumes as matrix **b**, and the FC-model is solved for **b**, then **b** = Liquid (Daughter or Melt) + Minerals (fractionating or accumulated assemblage). If compositions of Liquid and Minerals are known (matrix **A**), it is possible to estimate, by least squares approximation, their proportion (in matrix **c**). The similarity of **b'** (matrix c multiplied with matrix **A**) to **b** (real value) is quantified with the sum of the square of the residuals ($\Sigma r^2$) as:

$$\sum r^2 = \sum_{i-1}^{n} (b_i' - b_i)^2 \qquad \text{(Eq. 1)}$$

RFC and Cumulate model results are considered acceptable when $\Sigma r^2 < 1.0$. Proportion of Liquid (Daughter or Melt) is expressed with **F** in matrix **c**.

Major-element mass balance models are calculated in the system $SiO_2$-$TiO_2$-$Al_2O_3$-$FeO^*$-$MnO$-$MgO$-$CaO$-$Na_2O$-$K_2O$. The LH5-2 cpx-bearing basalt, with the lowest SiO2 and the highest MgO contents, was selected as possible source for all pyroxene-bearing trachyandesites and trachytes. The fractional crystallization hypothesis is then tested for all the LHPCS studied rocks and considering the magmatic mineralogy made of An-rich plagioclase, Ti-rich clinopyroxene, Mg-rich olivine, spinel. The same mineral assemblage was used then to verify the cogenetic relationship between studied LHPCS basalts through progressive crystal accumulation. All calculations were managed with Microsoft Office Excel 2019. Results of FC-models are presented in Supplementary Table 6.

The RFC modeling has been applied to all studied trachyandesites and trachytes. It was verified that a fractionation of the Pl+Cpx+Ol+Sp assemblage in the range of: (i) 45-63 wt% ($\Sigma r^2$ 0.37-0.92) is necessary to produce Opx-free trachyandesites, (ii) 59-69 wt% ($\Sigma r^2$ 0.38-0.92) is capable to produce Opx-bearing trachyandesites, and (iii) 73-74 wt% ($\Sigma r^2$ 0.88-0.91) is requested to produce trachytes. The crystal accumulation has been tested to verify the genetic linkage between Cpx-free basalt (LH18-1) and Cpx-bearing basalts (LH5-2, LH27-1). It was verified that a crystal accumulation of the Pl+Cpx+Ol+Sp assemblage in the range of 16-17 wt% ($\Sigma r^2$ 0.05-0.15), with Cpx ranging 5-7 wt%, can produce the LHPCS Cpx-bearing basalts.

The results obtained from FC-models thus indicate that the LHPCS volcanic rocks are genetically linked melts, due to crystal accumulation (basalts) and fractional crystallization (intermediate and felsic rocks) of a Pl+Cpx+Ol+Sp mineral assemblage. Trachyandesites and trachytes represent different degrees of fractionation (RFC values in the range 45-74%) starting from a Cpx-bearing basaltic source. Cpx-bearing basalts are interpreted as the result of crystallization and accumulation of Cpx, together with Pl+Ol+Sp, in a pristine Cpx-free basaltic melt. Results from FC-models also confirm the possibility to produce hydrous felsic melts starting from a nominal anhydrous ($H_2O$ < 1 wt%; e.g., Webster et al., 1999) mafic parental melt. Integrating FC-model and hygrometer (Putirka, 2008) results, LHPCS trachytes show $H_2O$ ca. 1.4-2.0 wt% and represent the ca. 25 wt% fractionated residual melt from a parental basaltic source characterized by $H_2O$ in the range 0.3-0.5 wt%.

## 8.2 Magma evolution beneath Los Humeros

The conceptual model of the present-day LHPCS magmatic plumbing system beneath the Los Humeros caldera is presented in Fig. 12. Based on textural observations, mineral chemistry and thermobaric estimates the early HT (1230-1270 °C) stage of LHPCS magma evolution is represented by high-anorthite plagioclase phenocrysts and Mg-rich olivine ($X_{Fo}$= 80-85%) crystallizing in the deep (ca. 8 kbar) basaltic reservoir. Where these magmas erupted directly, they formed Cpx-free Ol-basalt lava flows such as the Texcal Lava flow (LH18). This scenario, for LH18 basalt sample, is confirmed by (i) olivine and plagioclase with homogeneous cores and normal monotonous zoning textures at rims, indicating a fast growth during ascent of magma (e.g., Streck, 2008); (ii) olivine with spinifex, dendritic and skeletal textures, interpreted as supercooling mineral texture largely resulting from rapid olivine-supersaturated magma rise from deeper level during the eruption (e.g., Donaldson, 1974; Nakagawa et al., 1998; Fowler et al., 2002; Dahren et al., 2012; Welsch et al., 2013), and (iii) plagioclase specimens with swallow-tailed crystal morphology, interpreted as rapid plagioclase growth due to undercooling related to eruption process (e.g., Renjith, 2014).

A permanence of these basaltic melts in the deep reservoir, together with a temperature decrease of ca. 100 °C can lead to clinopyroxene appearance/crystallization in the system (e.g., Groove, 2000) and its

progressive accumulation in the phenocryst assemblage. This hypothesis is supported by Cpx-Liq thermometry models for Cpx1 (Ti-rich augites in basalts) indicating Cpx appearance at ca. 7-8 Kbar and 1150 °C (mean values), and by FC-models indicating a Pl+Cpx+Ol+Sp crystal accumulation up to 15-17 wt% in the pristine basaltic melt to produce the Cpx-bearing basalts.

Where these magmas erupted as intra-caldera basalts (LH5-2, LH27-1), they are characterized by the further crystallization of (i) progressively Fe-rich olivine (up to $X_{Fo}$= 17-20%), (ii) Ab-rich plagioclase ($X_{An}$= 25-30%), (iii) Cpx3 unzoned homogeneous phenocrysts and overgrowth (normal monotonous and normal low-amplitude oscillatory zoning) on Cpx1-cores, (iv) Cpx4 (Di-rich) microcrystals and microlites and (v) Cpx5 (Aeg-Aug) Na-clinopyroxenes. This mineral assemblage (mineral chemistry and textures) together with the obtained thermobarometric results, describes a near-isothermal magma uprising within a narrow temperature window of ca. 1070-1150 °C. Such crystal-bearing magmas ascend from the deeper reservoir to intermediates and shallower stagnation levels, where different phases would crystallize, before the eruption (e.g., Feng and Zhu, 2018). In particular, (i) the homogeneous unzoned cores of phenocrysts represent the early crystallization at equilibrium with the melt, (ii) the normal low-amplitude oscillatory zoning from Pl and Cpx phenocrystals indicates a kinetically driven crystallization (e.g., Ginibre et al., 2002; Streck, 2008; Renjith, 2014), whereas the normal monotonous zoning observed in many Pl, Cpx and Ol phenocrysts indicates a fast growth during ascent of the magma (e.g., Streck, 2008); (iii) microlites formation indicates water exsolution driven crystallization (e.g., Rutherford, 2008; Renjith, 2014) during a relative rapid ascent or eruption processes (e.g. Renjith, 2014); and (iv) the similarity of compositions between Pl and Cpx phenocrysts rims and microlites confirms that there were essentially no major changes in the temperature of any of these basaltic magmas during the ascent (e.g., Rutherford, 2008). This scenario of rapid ascent of LHPCS basaltic magmas is also supported by the observed high-vesicularity textures, interpreted as bubble-growth processes during a relative fast magma rise precluding exsolved volatile to escape (e.g. Sparks, 1978; Sparks et al., 1998; Rutherford and Gardner, 2000; Rutherford, 2008; Costa et al., 2013; Feng and Zhu, 2018).

Fractional crystallization of An-rich plagioclase, Fo-rich olivine, Ti-rich augite and spinel (Pl+Ol+Cpx+Sp in RFC-models) in the primary cpx-bearing basaltic magmas produces residual melts (ca. 30-55 wt%) of trachyandesitic compositions. These evolved buoyant melts will be prone to leave the basaltic reservoir to produce shallower intrusions in a vertically extensive magmatic system (e.g., Jackson et al., 2018), carrying early-formed phenocrysts (i.e., anorthitic plagioclase antecrysts) to the intermediate reservoir and stall. Within this intermediate vertically-distributed layered storage system in the middle crust, Cpx2 clinopyroxene and all the rest of plagioclase phenocrysts start to crystallize producing progressively evolved felsic residual melts able to migrate upward in the feeding system or erupt (e.g., Freundt and Schminke, 1995; Patanè et al., 2003; Klugel et al., 2005; Stroncik et al., 2009; Aulinas et al., 2010; Dahren et al., 2012; Keiding and Sigmarsson, 2012; Scott et al., 2012; Jeffery et al., 2013; Coombs and Gardner, 2001; Barker et

al., 2015; Feng and Zhu, 2018). Similarly to LHPCS basalts, the phenocryst morphologies and textures, together with the microlites compositions and the vesicle-rich textures decrived in trachyandesitic melts, suggest a nearly isothermal rapid ascent precluding exsolved volatiles to escape and producing water exsolution driven crystallization (e.g., Rutherford, 2008; Renjith, 2014).

The shallowest magma stagnation level (< 3kbar; mean 1.5 kbar) has been here interpreted as a complex magma plexus constituted by a system of small magma volumes, distributed in locally interconnected pockets and batches. In this plexus mafic and intermediate magmas shortly stall prior to erupt. Whereas more evolved melts reside for a relatively longer time, enough to crystallize orthopyroxene and to enabling the escape of part of the exsolved volatiles (e.g., Sparks et al., 1998; Feng and Zhu, 2018; Clarke et al., 2007), as suggested by  phenocryst textures and compositions, and by poor-vesicle textures observed in Opx-trachyte samples (LH5-1, LH6-1).

Compositional reverse zoning associated with disequilibrium textures and dissolution/resorption patterns in phenocrysts, are widely considered indicators of both magma-replenishment or assimilation processes (e.g., Wright and Fiske, 1972; Duda and Schminkcke, 1985; Clague et al., 1995; Yang et al., 1999; Klugel et al., 2000; Zhu and Ogasawara, 2004; Stroncik et al., 2009; Ubide et al., 2014; Viccaro et al., 2015; Gernon et al., 2016; Feng and Zhu, 2018). In the case of LHVC, almost all investigated LHPCS samples, from basalts to trachytes, contain mainly phenocrysts with homogeneous cores and low-amplitude oscillatory or normal monotonous zoned rims (Pl+Ol+Cpx) or unzoned homogeneous phenocrysts (as in case of Cpx3 and Opx). Rare specimens not suitable for mineral-liquid thermobarometry, such as plagioclase and clinopyroxene with patchy cores or olivine xenocrysts, are reported. The general absence of disequilibrium textures and patterns in LHPCS studied samples, is therefore interpreted as a lack of evidence of major mixing/recharge and/or assimilation processes acting in the plumbing system (e.g., Cashman et al., 2017 and references therein). This hypothesis is in line with the results obtained from tests for mineral-melt equilibria. Rhodes's diagram (Rhodes et al., 1979; Putirka, 2008) for olivine compositions (Fig. 8a) highlights a progressive decrease in Mg#$^{liq}$ from basalts to trachytes coupled with general absence of xenocrystals/antecrystals cargo. This behavior is compatible with a complete removal from the melt of previously crystallized Mg-olivine (Roeder and Emslie, 1970; Dungan et al., 1978; Rhodes et al., 1979; Putirka, 2008; Melluso et al., 2014). All LHPCS melts (from basalts to trachytes) invariably show suites of olivines with maximum forsterite (Fo) contents in equilibrium with the respective whole rocks, and vertical trends consistent with closed-system melt differentiation (Roeder and Emslie, 1970; Rhodes et al., 1979; Putirka, 2008; Melluso et al., 2014). Similar behavior is obtained for orthopyroxene (Fig. 8b), where again Rhodes's test highlights (i) absence of antecrystals, and (ii) Opx-suites progressively and normally Fe-enriched from trachyandesites to trachytes. The absence of clinopyroxene clots and overgrowth mantle textures on orthopyroxene crystals, again excludes the occurrence of magma mixing/recharge processes (Laumonier et al., 2014; Neave et al., 2014; Zhang et al., 2015; Feng and Zhu, 2018). Such interpretation is supported also by field observations,

where the interbedded basaltic andesite and trachydacite fall deposits of the ca. 7 ka Cuicuiltic Member
show no evidence of magma-mixing (Dávila-Harris and Carrasco-Núñez, 2014).
An-Ab partition coefficients (e.g., Putirka, 2008; Jeffery et al., 2013) show a comparable scenario (Fig. 10) in
which: (i) the LHPCS basalts are characterized by suites of plagioclases with maximum anorthite (An)
contents in equilibrium with the respective whole rocks, and progressive $An^{Pl}$ decrease consistent with
closed-system differentiation; and (ii) the progressive decrease in predicted $An^{Liq}$ from basalt to trachyte is
compatible with evolved melts differentiation via fractional crystallization. The LHPCS intermediate and
evolved products show plagioclase phenocrysts characterized by An-rich homogeneous cores (An 70-85%),
with compositions comparable to those of basalts. These An-rich cores can be crystallized in two possible
scenarios. The first one is related to the $H_2O$ content in magma. Increasing the water content in melt
strongly favors crystallization of An-richer plagioclase. A water content rise from 0.5 to 2.0 wt% could lead
to an increase of the An component up to 6-8 mol% (Bindeman and Bailey, 1999; Sano e Yamashita, 2004;
Ushioda et al., 2014). In this view, the An-rich plagioclase in intermediate and felsic rocks can be
interpreted as the response to the increasing water-content in the fractionated melt. The second scenario
implies that An-rich plagioclase taps a more primitive stage of basalt segregations. Since plagioclase
phenocrystals with An in the range 65-81% are commonly found in LHPCS basalts, the An-rich plagioclase
cores in trachyndesites and trachytes could represent either antecrysts derived from crystallization of early
sills in the magmatic reservoir system (sensu Jackson et al., 2018) or crystallization products in an earlier
stage of the trachyandesite and trachyte segregation from the basaltic reservoir (e.g., Bindeman and Bailey,
1999; Kinman and Neal, 2006). We suggest that both scenarios concurred to the genesis of An-rich
phenocrysts in trachyandesites and trachytes. Noteworthy, when An-rich plagioclase crystals are found (in
mafic and intermediate rocks with Pl+Ol+Cpx assemblages), it implies that no significant clinopyroxene
crystallization has occurred prior to the anorthitic plagioclase (Bindman and Bailey, 1999).
With respect to plagioclase, a similar behavior is observed also for clinopyroxene and in particular for Cpx1
and Cpx2 (clinopyroxene cores in basalts and in trachyandesites+trachytes, respectively) populations. Since
these mineral cores (Pl, Cpx1 and Cpx2) generally present normal growth rims (i.e., Ab-rich Pl and Cpx3), we
suggest that stagnation levels at both intermediate and shallower depths underwent crystallization in a
closed system. Otherwise, features such as: i) diffused reverse zoning, ii) high-temperature crystal-clots,
mantling and overgrowth, iii) disequilibrium and dissolution textures (e.g., Stroncik et al., 2009; Cashman et
al., 2017; Feng and Zhu, 2018 and references therein), should be widely observed, but this is not the case in
the LHPCS studied lavas.

**8.3 The magma plumbing system**
The petrological archive constituted by the LHPCS lavas, spanning from transitional- and alkali-basalts to
trachytes, describes the Holocene activity of the LHVC. Harker diagrams for major element bulk

compositions of the LHPCS lavas are characterized by linear trends (Fig. 3 b-d) comparable to those expected for cogenetic melts (e.g., Giordano et al., 2012). Major-element FC-modeling confirms the hypothesis of a common genesis for the LHPCS volcanic rocks through crystal fractionation/accumulation processes of the same mineral assemblage (Pl+Cpx+Ol+Sp). Furthermore, textural observations and results from FC-models permit to exclude mass-change or mass-addition processes driven by AFC-Mixing processes.

Results obtained from the application of different and independent thermobarometry models (Fig. 11) confirm the working hypothesis of a complex magmatic plumbing system rather than a single "standard" magma chamber (e.g., Keiding and Sigmarsson, 2012; Cashman and Giordano, 2014; Cashman et al., 2017; Feng and Zhu, 2018) developed beneath the active Los Humeros caldera and feeding the LHPCS volcanism.

With the aim to propose an updated and realistic conceptual model of the present-day main storage zones and magma plumbing system within the crust below Los Humeros caldera, we integrate pressure-temperature estimates acquired in this study with the existing data related to the crustal structure and corresponding physical parameters of the study area. The resulted model is shown in Figure 12.

The density of TMVB crust shows a large range between 1800 kg/m$^3$ for unconsolidated sediments to about 3000 kg/m$^3$ for the lower crust and 3300 kg/m$^3$ for the upper mantle (Dziewonski and Anderson, 1981; Campos-Enríquez and Sánchez-Zamora, 2000; Davies, 2013). The available compilation of crustal data for LHVC is recovered by the measure N°10 of the Crust 1.0 global model (Dziewonski and Anderson, 1981; Davies, 2013). The measure N°10 (yellow star in Fig. 1) is located within the study area at the southern termination of the Tepeyahualco Lava Flow and describes a crust made of five main seismic layers(Fig. 12): (i) upper sediments (thickness: 1 km, density 2110 kg/m$^3$); (ii) middle sediments (thickness: 0.5 km, density 2370 kg/m$^3$); (iii) upper crust (thickness: 13.6 km, density 2740 kg/m$^3$); (iv) middle crust (thickness: 15.3 km, density 2830 kg/m$^3$); and (v) lower crust (thickness: 13.6 km, density 2920 kg/m$^3$). Inferred (seismic) Moho depth is reported at -41.7 km with an upper mantle density of 3310 kg/m$^3$ (Dziewonski and Anderson, 1981; Davies, 2013). Here we use a five-tiered density model as derived from the Crust 1.0 global model to convert obtained pressure estimates to crustal depths below LHVC.

The thermobarometry models applied to the LHPCS lavas define a broad region of crystallization between ca. 0 and 30 km in depth that can be described with a quadrimodal distribution of pressure values (Fig. 12). This allow us to propose a complex polybaric continuous heterogenous multilayered transport and storage magmatic system.

A deep-seated anhydrous Ol-basalt reservoir at depths of ca. 28-33 km (7.6-9.2 kbar), at the boundary between lower and middle crust, below the caldera is recorded by (i) An-rich Pl cores (XAn = 50-70 %), and (ii) Ti-rich augitic Cpx1 cores (Mg# up to 75, TiO$_2$ up to 4.57 wt%). For this mafic reservoir, the overlapping of the calculated anhydrous temperature estimates as derived from Pl-Liq, Cpx1-Liq and Ol-Liq pairs spans ca. 1000 – 1300 °C. The highest anhydrous temperature values are derived from the Cpx-free Ol-basalt

Texcal lava flow (LH18), where the convergence of Pl-Liq thermobarometry and Ol-Liq thermometry models indicate conditions of ca. 1230-1270 °C at ca. 8 kbar. Lower anhydrous temperatures of ca. 1000-1210 °C are obtained at a comparable average pressure values for Cpx-bearing intra caldera Ol-basalts (LH5-2; LH 27-1). These results are in agreement with existing literature on the near-liquidus melting behavior of high-Al basaltic magmas (Mg# ca. 60-70 and $Al_2O_3$: 17-19 wt%) under dry conditions (e.g., Thompson, 1974; Grove et al., 1982; Crawford et al., 1987; Bartels et al., 1991; Grove, 2000). At 1250-1300 °C and ca. 10 kbar (Point *A* in Fig. 11) the basaltic melt is in equilibrium with a mantle peridotite mineral assemblage of olivine + clinopyroxene (Kushiro and Yoder, 1966; Presnall et al., 1978; Grove et al., 1982; Fuji and Scarfe, 1985; Takahashi, 1986; Fallon and Green, 1987; Bartels et al., 1991; Sisson and Layne, 1993; Wagner et al., 1995; Grove et al., 1997; Grove, 2000; Kinzler et al., 2000). Following the models proposed by Thompson (1974), Bartels et al. (1991) and Grove (2000), a temperature decrease would lead primary melts to pass the "dry basaltic liquidus" and start the crystallization of Ol+Pl (higher temperatures) or Ol+Cpx+Pl (lower temperatures) assemblages (see stability fields in Fig. 11). Given the ubiquitous presence in all LHPCS basalts of well-developed euhedral to subhedral olivine crystals (both as phenocrysts and microlites) at equilibrium with anorthitic plagioclase, it is possible to exclude that crystallization history started at depth > ca. 10-12 kbar where olivine is not a stable phase and the primary assemblage would be characterized only by Cpx+Pl+Sp in equilibrium with melt (Kushiro and Yoder, 1966; Thompson, 1974; Presnall et al., 1978; Bartels et al., 1991; Grove, 2000).

A second magma transport and storage systems can be recognized at depths of 15-30 km (ca. 4.5 − 7.8 kbar), in continuity with the deeper basaltic reservoir and distributed along the whole middle crust thickness, as recorded by the wide range of pressure estimates obtained from plagioclase ($X_{An}$= 40-70%) and Cpx2 clinopyroxene cores (Mg#: 59-84; $TiO_2$ mean value 0.99 wt%). Thermometry models based on plagioclase, Cpx2 clinopyroxene and olivine show convergence for hydrous temperature values in the range of 979 − 1263 °C. Thermobarometry models, together with textures and petrographic relations in all analyzed trachyandesite and trachyte samples suggest that all plagioclase, all Cpx2 clinopyroxene phenocrysts, and part of microlites grew in this second storage system. In particular, it is possible to observe two main crystallization temperature conditions: (i) at ca. 1190 °C (weighted mean value, MSWD= 2.2, n= 205) plagioclase phenocrysts crystallization in trachyandesite melts is observed, whereas (i) at the lower temperature of ca. 1070 °C (weighted mean value, MSWD= 1.7, n= 155) is reported the crystallization of all olivine, all Cpx2 phenocrysts, and plagioclase phenocryst in trachytes. We interpret the common Pl+Cpx2 phenocryst-forming barometric conditions as the evidence of a growth-dominated regime within this second magma storage zone (e.g., Barclay et al., 1998; Humphreys et al., 2006; Scott et al., 2012). whereas the smaller crystals (microcrystals and microlites) represent the nucleation-dominated regime (Scott et al., 2012) that can be associated with ascent-related decompression of melts at shallower levels (e.g., Cashman, 1992; Cashman and Blundy, 2000; Humphreys et al., 2009).

The third melt storage zone occurs at shallower depths of ca. 10-15 km, possibly corresponding to the transition between middle- and upper-crust, as indicated by convergence of barometric estimates (weighted mean value of 3.9± 0.2 kbar (±1σ), MSWD= 0.80, n= 203; P ranging ca. 1-7 kbar) obtained from Cpx3 clinopyroxene (i.e., unzoned phenocrysts and overgrowth/rims around earlier formed Cpx1- and Cpx2-cores) population. For this third storage zone, the Cpx3-Liq thermometry model indicates a mean temperature of 1040 °C (weighted mean value, MSWD= 2.6, n= 203; T ranging ca. 940-1210 °C), comparable to those calculated for Ol+Cpx2 assemblages in the previous described second and deeper stagnation system. The obtained pressure estimates for the second and the third storage systems are compatible with multiple magma storage pockets in which melts of comparable compositions ascend slowly enough for phenocrysts to form (e.g. Scott et al., 2012), and start cooling before the final ascent to shallower conditions (e.g., Dahren et al., 2012; Chadwick et al., 2013; Gardner et al., 2013; Jeffery et al., 2013; Preece et al., 2013; Troll et al., 2013). Taking into account the textures and the chemistry of Cpx3 clinopyroxene phenocrysts, the obtained thermobarometric estimates could be interpreted as the pressure-temperature environment of last major levels of magma stagnation and fractionation (Putirka, 1997; Klugel et al., 2005; Galipp et al., 2006; Stroncik et al., 2009).

The fourth shallowest storage zone located at depths of ca. 3-7 km (weighted mean value of 1.5± 0.2 kbar (±1σ), MSWD= 0.24, n= 177; P ranging ca. 0.1-4.5 kbar), is required to explain the presence of (i) Cpx4 clinopyroxene (microcrystals and microlites) in all LHPCS lavas, and (ii) Aeg-rich Cpx5 clinopyroxene in basalts, and iii) Fe-olivine (Fo = 55-65%) and orthopyroxene in Opx-bearing evolved LHPCS lavas. Magmas in this shallow storage system show a wide range of temperature values calculated for hydrous melts: (i) ca. 1060 °C (weighted mean value, MSWD= 2.4, n= 7) for Aeg-rich Cpx5 crystallization in basalts; (ii) ca. 1070 °C (weighted mean value, MSWD= 1.09, n= 168) for Cpx4 and Opx crystallization in trachyandesites; and (iii) ca. 965 °C (weighted mean value, MSWD= 2.2, n= 78) for olivine, Cpx4 and Opx crystallization in trachytes. Thermobaric estimates obtained for Aeg-rich Cpx5 agree with those calculated for transitional basalts at Pantelleria (White et al., 2009 and references therein), whereas orthopyroxene crystallization conditions overlap with the existing literature for intermediate rocks (e.g., Rutherford et al., 1985; Wallace and Anderson, 2000; Reubi and Nicholls, 2004; Allan et al., 2013; Jeffery et al., 2013). The broad distribution of melt chemistry from basalt to trachyte, together with the obtained thermobaric estimates, define a shallow magma storage environment characterized by progressive accumulation of small locally interconnected magma pockets and batches (e.g., Reubi and Nicholls, 2004; Jeffery et al., 2013) dispersed in the upper crust (<10 km) with a possible magma plexus at a depth of 2-4 km under the caldera (e.g., Armienti et al., 1989; Freundt and Schminke, 1995; Pietruszka and Garcia, 1999; Patanè et al., 2003; Klugel et al., 2005; Stroncik et al., 2009; Dahren et al., 2012; Jeffery et al., 2013; Coombs and Gardner,, 2001, 2004).

**8.4 "Standard" versus multilayered magmatic plumbing system**

Existing conceptual models for LHVC are based on the "Standard Model" (*sensu* Gualda and Ghiorso, 2013),
considering a single, bowl-shaped and long-lived, melt-dominated magma chamber of 1000-1500 $km^3$, at
depth of 5 to 10 km (Verma, 1983, 1984, 1985a, 1985b; Verma and Lopez, 1982; Verma et al., 1990; Verma
and Andaverde, 1995; Verma et al., 2011; Verma et al. 2013; Carrasco-Núñez et al., 2018). However, these
models mainly refer to the Los Humeros Caldera stage activity (Carrasco-Núñez et al., 2018 and references
therein), lasted ca. 130 ky, where the major caldera-forming events (Xaltipan and Zaragoza ignimbrites, 115
$km^3$ and 15 $km^3$ DRE, respectively) and the large Plinian eruptive episode (Faby Tuff, 10 $km^3$ DRE)
necessitated feeding from a huge, voluminous magma chamber (Carrasco-Núñez and Branney, 2005;
Carrasco-Núñez et al., 2018).
On the other hand, the Holocene eruptive phase of the LHPCS is a characterised by bimodal volcanism
(Carrasco- Núñez et al., 2017a; Carrasco-Núñez et al., 2017b; Carrasco-Núñez et al., 2018), typified by
alternating episodes of effusive and explosive volcanism with a wide range of compositions, spanning from
basaltic to trachytic lava flows and mafic to felsic pumice and scoria fall deposits, erupted by tens of
monogenetic eruptive centers located in the LHVC (e.g., Norini et al., 2015; Carrasco- Núñez et al., 2017a;
Carrasco-Núñez et al., 2017b; Carrasco-Núñez et al., 2018 ). This volcanic activity is characterized by
patially distributed, small volumes of erupted material (ca. 6 $km^3$ of mafic lavas, 10 $km^3$ of intermediate and
felsic lava, and 1 $km^3$ of mafic and felsic tephra; Carrasco-Núñez and Branney, 2005). Furthermore, key
elements, such as the lithic-free character of the LHPCS volcanic products, their overall textures and
chemistry of the constituent mineral assemblages, coupled with the results from RFC-models, suggest that
LHPCS magmatism is characterized by batches of magma evolving in a nearly closed system, unaffected by
magmatic assimilation and mixing/recharge processes. In particular, the almost complete lack of magma
mixing/recharge events (e.g., Lee et al., 2014) is confirmed by the absence of the typical expected minerals
textures (e.g., Streck, 2008; Renjith, 2014) such as: (i) fine-sieve textures and resorption surfaces due to
reaction with a more primitive magma; (ii) glomerocryst-forming due to the recrystallization/suturing at rim
of resorbed crystals; (iii) reverse zoning due to compositional inversion in open/recharged system; and (iv)
reaction rims, breakdown mantles and crystal clots due to the disequilibrium-triggered recrystallization into
a new set of minerals.
The existing literature focused on magma recharge processes (e.g., De Paolo, 1981; Hofmann, 2012; O'Neill
and Jenner, 2012; Lee et al., 2014) highlights that a high evacuation/eruption efficiency would shorten the
residence-time of magma in the storage chamber and would reduce the effect of crystallization in
modifying the magma composition (Lee et al., 2014). Moreover, in case of eruption/evacuation rates higher
than the recharge rates (e.g., Lee et al., 2014), it is possible to hypothesize a magmatic system dominated
by ephemeral, closed-system magma batches not affected by major mixing processes prior of their
evacuation/eruption (e.g., De Paolo, 1981; Hofmann, 2012; O'Neill and Jenner, 2012; Lee et al., 2014). This
scenario best approximates the characteristics observed for all the Holocene LHPCS magmatic products. In

addition, the lack of liquid-dominated zone(s) (e.g., Bachmann and Bergantz, 2008), where mixing could occur (e.g., Cashman and Giordano, 2014), suggests that the remnants of the huge magma chamber of the LH caldera stage are now completely solidified and crosscut by the uprising LHPCS mafic and felsic magmas. This scenario is also coherent with the post-caldera eruption behavior observed in other volcanic complexes, such Ischia (e.g., Casalini et al., 2017), and it is consistent with the recent literature proposing complex magma chamber reservoirs made up of multiple discrete melt pockets with no mass-exchange and reactivated shortly before eruption (e.g., Cashman and Giordano, 2014; Cashman et al., 2017; Casalini et al., 2017).

Thermobarometric estimates obtained in this study, combined with the existing literature and integrated with information from the crustal structure beneath Los Humeros caldera, therefore permit us to discard the "standard model" of the huge voluminous chamber in favor of a more feasible conceptual model characterized by a polybaric magmatic plumbing system of multiple, more or less interconnected magma transport and storage layers, i.e. transient batches and ponds of different magmas, localized beneath Los Humeros nested caldera and feeding the Holocene activity of the LHVC. In particular,  our results indicate that magma transport and storage levels beneath Los Humeros caldera are vertically distributed across the whole crust from ca. 30 to 3 km (from the lower- to the very upper-crust) with density contrasts between the different crustal layers acting as a controlling parameter for ascending or stalling magmas (e.g., Dahren et al., 2012), reflecting the buoyant magma compositions and the melt fractions (e.g., Cashman et al., 2017; Jackson et al., 2018). Moreover, it is possible to propose that each of these crust/density boundaries have determined lateral transport and grow of magma stagnation pockets (e.g., Dahren et al., 2012; Jackson et al., 2018).  At depths < 5 km, buoyant magmas and fractionated melts (from mafic to felsic) ascending from all the lower storage zones are stalled once more. The shallowest complex multi-storage system is interpreted as a plexus of scattered, more or less interconnected, ephemeral small-volume batches and pockets of melts, without any defined spatial distribution, as confirmed by field-locations of the LHPCS studied lavas eruptive centers.

A shallow storage zone presenting magmas with heterogenous compositions (from mafic to felsic) has been already proposed by Dávila-Harris and Carrasco-Núñez (2014) to explain the eruptive history of the intra-caldera Cuicuiltic Member that was produced by the coeval eruption of mafic and felsic unmixed magmas. However, a shallow ponding system characterized by heterogeneous composition of magmas involved beneath Los Humeros caldera is not an exceptional case. Examples of shallow heterogeneous reservoirs beneath active volcanic complexes are widely reported (e.g., Nairn et al., 1998; Kratzmann et al., 2009; Sigmarsson et al., 2011; Keiding and Sigmarsson, 2012).

Our results also agree with the work of Creon et al. (2018), where calculated fluid saturation depths derived for melt inclusions in post-caldera lavas indicate different magma-ponding levels within a range of depths

between 5 and 13 km, together with a possible deeper reservoir (26-32 km) and a final shallow stagnation
level at ca. 1.5-3.0 km.

**8.5 Implications for the active geothermal system**
The geothermal activity of a volcanic complex is expected to be the result of stagnation and cooling of
magmas in the shallower storage zone (e.g., Gunnarsson and Aradóttir, 2015), where classic conductive
models are adopted to model the heat source, mainly controlled by age and volume of the magmatic
system (Smith and Shaw, 1975; Cathles et al., 1997; Duffield and Sass, 2003; Gunnarsson and Aradóttir,
2015; Carrasco-Núñez et al., 2018). As widely demonstrated (e.g., Smith and Shaw, 1975; Cathles et al.,
1997), a very large intrusion would produce a long-lived hydrothermal/geothermal system. Many numerical
models (e.g., Cathles et al., 1997) suggest that, in the most favorable conditions, a voluminous (>2000 km$^3$)
intrusion/chamber of mafic melt could be able to sustain a convective geothermal system up to 800 Ky. On
the other hand, very small mafic sills and dike intrusions (<10 km$^3$) would produce very localized thermal
anomalies and could cool down to the solidus temperature in less than 0.1 ky (Nabelek et al., 2012), and
definitely cool in ca. 1 ky (e.g. Cathles et al., 1997). Convection due to hydrothermal fluid circulation,
increases the cooling rate of a magmatic intrusion (Cathles et al., 1997).
The present geothermal activity of LHVC is characterized by a limited NNW-SSE non-homogeneous areal
distribution within the Los Potreros nested caldera (e.g., Norini et al., 2015; Urbani et al., 2019). Based on
(i) the young age (Upper Pleistocene-Holocene) of most of the LHPCS volcanic activity; (ii) the relatively
small erupted volumes of the LHPCS lavas, in particular of those erupted within the Los Potreros caldera;
and (iii) the existence of a shallower magmatic plexus characterized by heterogeneous unmixed magmas
(this study), we therefore discard the hypothesis of a single, large and voluminous shallow magmatic
chamber homogenously distributed beneath the caldera, in favor of a more feasible scenario characterized
by an upper crustal plexus made of small, single-charge ephemeral pockets of different magmas localized
beneath Los Humeros nested caldera, very close or within the Los Humeros exploited geothermal field. In
this scenario, every LHPCS magma pocket and cryptodome within the Los Humeros caldera (see Urbani et
al., 2019) could be interpreted as a scattered and localized short-lived (ca. 0.1-1 ky; Cathles et al., 1997)
heat source, whereas the cooling and solidified remnants of the huge magma chamber of the caldera stage
could still represent a background positive thermal anomaly affecting the volcanic field.
Our reconstruction of the Los Humeros heat source therefore suggests the possible existence of a wide
background positive thermal anomaly associated to the cooling solidified remnants of the voluminous
magma chamber of the caldera stage, with juxtaposition of scattered high-frequency heat sources related
to the very shallow intrusive complex that make-up the surficial (upper crustal) plexus of the LHPCS
magmatic plumbing system.
In the light of our results, a revision/update of the heat source feeding the Los Humeros geothermal system
is needed to produce correct and up-to-date geothermal potential estimates of the geothermal field and to
develop efficient geothermal exploration and exploitation strategies.

## 1049     9. CONCLUSIONS

In this study we propose an integrated field-based petrographic-mineralogical approach to unravel the
evolution and configuration of the present-day magmatic plumbing system feeding the post-caldera stage
activity of LHVC. The main results of this study can be summarized as follows:
(i)    The Rayleigh fractional crystallization (RFC) models demonstrate that all LHPCS magmas, from

basalts to trachytes, belong to the same line of descent and evolve through a progressive

fractionation of the Pl+Cpx+Ol+Sp mineral assemblage.

(ii)   A complex polybaric magmatic transport and storage system, characterized by multiple magma

levels more or less interconnected in space and time, has been recognized based on application of

mineral-melt thermobarometry models.

(iii) A deep mafic reservoir (at ca. 30 km depth) is identified by the Pl+Ol assemblage in basalts.

Intermediate magma storage systems (in the whole middle crust) are described by the composition

of the Cpx phenocrysts , whereas  a shallow magmatic stagnation system (ca. 1.5 kbar; 3-5 km

depth) is defined by crystallization of Cpx microlites (aegirine clinopyroxenes in basalt) and, in

particular, by Opx growth in most evolved melts. All the Cpx-bearing lavas are produced by

progressive differentiation via polybaric fractional crystallization during magma ascent through the

plumbing system.

(iv) The chemical composition of the main phases (Ol, Pl, Cpx, Opx), together with results from FC-

modelling, do not support a magmatic feeding system dominated by magma mixing and magma

replenishment. They are instead compatible with a plumbing system dominated by discrete levels,

pockets and batches of melts.

(v)   The thermobarometric results indicate that, unlike previously believed, the configuration of the

magmatic plumbing system is vertically extensive across the entire crust. A deeper residence zone

for basalts is proposed at ca. 8 kbar (ca. 30-33 km depth), together with a complex zone, from

middle (6-4 kbar) to upper crust (0.5 kbar) depths, where basalts rapidly ascend and stall prior to

erupt. This zone also corresponds to depths where smaller batches of mafic magma differentiate to

trachyandesites and trachytes at times interconnected with the lower feeding zone.

(vi) The main outcome for the modeling of the magmatic heat source of the LHVC geothermal system is

the inadequacy of conservative conceptual models based on the classical melt-dominated, single,

long-lived and voluminous magma chamber (i.e., "Standard Model"), in favor of an innovative and

more realistic vision of the magmatic plumbing systems made of multiple, more or less

interconnected, magma transport and storage layers within the crust, feeding small (ephemeral)
magma pockets at shallow-crust conditions.
(vii) The proposed model for the magmatic plumbing system at LHVC provides a new configuration of
the heat source feeding the present geothermal reservoir that must be taken into account for
geothermal exploration and exploitation purposes.

**APPENDIX A: Analytical details**

**A.1 Petrography of volcanic samples**

Rock magmatic fabrics, textures and mineral assemblages were studied on polished thin sections, using a Nikon Eclipse 50iPol polarized light microscope (PLM) equipped with Nikon Ds-Fi2 CCD camera (Nikon, Tokyo, Japan) and Nikon Nis-Elements software (Ver4.30.01), at Laboratorio di Microtettonica, Dip. Science, Università Roma Tre (Roma, Italia). Mineral abbreviations follow Whitney and Evans (2010).

**A.2 Bulk major element geochemistry**

After washing in distilled water, samples were grounded in an agate mill, pre-contaminated with an aliquot of sample. Whole-rock major element concentrations (4 samples) were measured at the Activation Laboratories (Ontario, Canada), through ion coupled plasma (ICP)- optical emission (OE). For major elements the uncertainty ($1\sigma$) is estimated better than 2% for values higher than 5 wt %, and better than 5% in the range 0.1-5 wt %. Additional samples (9) were analyzed by X-ray fluorescence (XRF) using a ZSX Primus II (Rigaku Co., Japan) at Nagoya University, Japan. Loss on ignition (LOI) was measured from the sample powder weight in a quartz glass beaker in the oven at 950°C for five hours. XRF-analyses were carried out following the procedure presented in Azizi et al., (2015; 2018a; 2018b). For major elements the uncertainty ($1\sigma$) is estimated better than 1% for values higher than 10 wt %, and better than 5% in the range 0.1-10 wt %.

**A.3 Mineral chemistry**

Polished thin sections (13 samples) selected for petrography investigations, were then studied for mineral chemistry and ca. 2400 analyses of mineral phases were obtained with a Cameca SX100 electron microprobe (EMP) at the Institut für Anorganische Chemie, Universität Stuttgart.

Operating conditions were 15 kV and 10 to 15 nA, counting times of 20 s both for peak and background. Spot sizes were 1-10 μm depending on the phases analyzed. Compositions were determined relative to natural and synthetic standards.

A set of reference materials (i.e., natural and synthetic oxides, and minerals) was used for routine calibration and instrument stability monitoring. In particular we used: (i) Si, Ca: natural wollastonite (P&H Developments); (ii) Si, Fe: natural fayalite USNM 85276 (Jarosewich et al., 1980); (iii) K: natural orthoclase (P&H Developments); (iv) Na: natural pure albite from Crete (Greece); (v) Al: synthetic corundum (P&H Developments); (vi) Mg: synthetic periclase (P&H Developments); (vii) Mn: natural rhodonite (P&H Developments); (viii) Ti: synthetic rutile (P&H Developments); (ix) Cr: synthetic chromium oxide (P&H Developments). Repeated analyses of the standards (Supplementary Table 7) resulted in one-sigma ($1\sigma$) standard deviations close to the ones calculated from counting statistics. For the major minerals, calculated $1\sigma$ (%) precisions are (i) better than 1.5 % for Si; (ii) better than 2% for Al; (iii) 1 to 5% for Ca, Mg, Fe, Mn, Ti

and Cr, applying the above-mentioned applied conditions. For Na and K, calculated $1\sigma$ (%) precisions are below 5% for analyses of feldspars and Aeg-rich clinopyroxene. The $1\sigma$ accuracy is estimated to be up to three times larger than the precision because additional effects such as uncertainty of the mass absorption coefficients that are used for the matrix correction of the microprobe raw data or instability of the beam may play a role.

Validation of mineral chemistry results were also achieved through opportune comparisons with the existing literature for (i) Mg-olivine (e.g., Hirano et al., 2004; White et al., 2009; Giordano et al., 2012; Melluso et al., 2014); (ii) Fe-olivine (e.g., Aldanmaz, 2006; White et al., 2009; Melluso et al., 2010; Giordano et al., 2012); (iii) aegirine-augite clinopyroxene (Cpx5 group; e.g., Piilonen et al., 1998; White et al., 2009; Njonfang et al., 2013); (iv) augite-diopside clinopyroxene (Cpx1-4 groups; e.g., Dawson and Hill, 1998; Aldanmaz, 2006; Melluso et al., 2010, 2014); (v) orthopyroxene (e.g., Papike et al., 1995; Aldanmaz, 2006; Carvalho and de Assis Janasi, 2012; Hu et al., 2018); (vi) feldspar (e.g., Keil et al., 1972; Giordano et al., 2012; Innocenti et al., 2013; Njonfang et al., 2013); and (vii) spinel and opaque minerals (e.g., Melluso et la., 2014). Back Scattered Electron (BSE) imaging was obtained by using the same electron microprobe with operating conditions of 15 kV, 50 nA. Mineral structural formulae of feldspar, olivine and spinel were calculated through the software CalcMin_32 (Brandelik, 2009). Mineral structural formulae of orthopyroxene were calculated following Putirka et al. (1996) and Putirka (2008). Clinopyroxene formula has been calculated following procedures reported in Putirka et al. (1996), Putirka (2008), Masotta et al. (2013). Clinopyroxenes were then classified integrating the Wo-En-Fs scheme (Morimoto, 1989) and J vs. Q scheme (Morimoto, 1988, 1989) with J= 2Na apfu and Q= (Ca+Mg+$Fe^{2+}$) apfu. Aegirine (XAeg) component correction, for Na-rich Cpx (Aegirine-Augite series), followed the scheme (XAeg= Na apfu if Na < $Fe^{3+\ Tot}$, XAeg= $Fe^{3+\ Tot}$ apfu if Na > $Fe^{3+\ Tot}$) proposed by Putirka et al. (1996), Putirka (2008) and based on $Fe^{2+}$ - $Fe^{3+}$ correction of Lindsley (1983).

**ACKNOWLEDGMENTS**

The authors are grateful to the Editor (Dr. C.J. Lissenberg), to Dr. C.M. Petrone and to an anonymous reviewer for their helpful and constructive comments that deeply contributed to improve the manuscript. The authors wish to thank the Comisión Federal de Electricidad (CFE, Mexico) for their assistance and support. This paper presents results of the GEMex Project, funded by the European Union's Horizon 2020 programme for Research and Innovation under grant agreement No. 727550 (scientific responsibility Guido Giordano), and by the Mexican Energy Sustainability Fund CONACYT-SENER, Project 2015-04-268074 (WP 4.5, scientific responsibility Gerardo Carrasco-Núñez). More information can be found on the GEMex Website: http://www.gemex-h2020.eu.

Authors would like to thank G. Norini for usefull discussions in the field. Special thanks to Javier Hernández, Jaime Cavazos, Francisco Fernández and Alessandra Pensa for their support in the fieldwork and logistics.

The Grant to Department of Science, Roma Tre University (MIUR-Italy Dipartimenti di Eccellenza, ARTICOLO

1, COMMI 314-337 LEGGE 232/2016) is gratefully acknowledged.

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

**SUPPLEMENTARY MATERIALS**
*Supplementary Tables S1: Feldspar, EMPA complete dataset.*
*Supplementary Table S2: Clinopyroxene, EMPA complete dataset.*
*Supplementary Table S3: Olivine, EMPA complete dataset.*
*Supplementary Table S4: Orthopyroxene, EMPA complete dataset.*
*Supplementary Table S5: Opaque Minerals and Spinels, EMPA complete dataset.*
*Supplementary Table S6: Major-Elements Mass-Balance Models.*
*Supplementary Table S7: Repetead measurement of EMP standards*

**FIGURES**

**Figure 1**

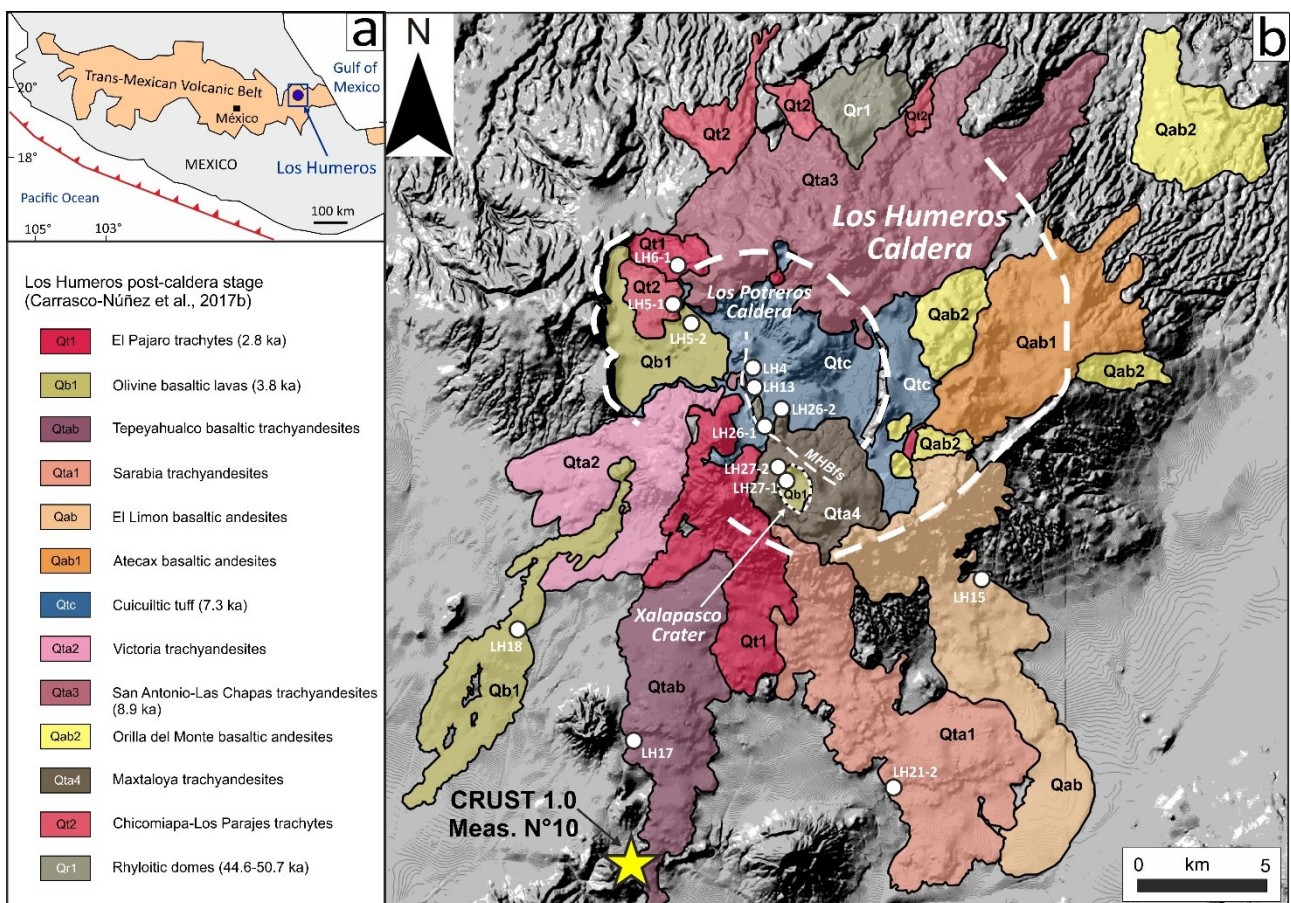

 **Figure-1.** Geological Context. (a) The Los Humeros volcanic complex (LHVC, blue dot) with respect to the

 Trans-Mexican Volcanic Belt (TMVB). (b) Shaded relief image obtained from 15 m resolution digital

 elevation model (DEM) of the LHVC. Volcanic products of the Los Humeros post-caldera stage are redrawn

 from Carrasco-Núñez et al. (2017b). The description of the volcanic units, their names and abbreviations

 follow Carrasco et al. (2017b). The map shows location (white dots) and volcanological significance of the

 samples used in this study.  The yellow star indicates the locality of the measure N°10 of the Crust 1.0

 global model (Dziewonski and Anderson, 1981; Davies, 2013).

**Figure 2**

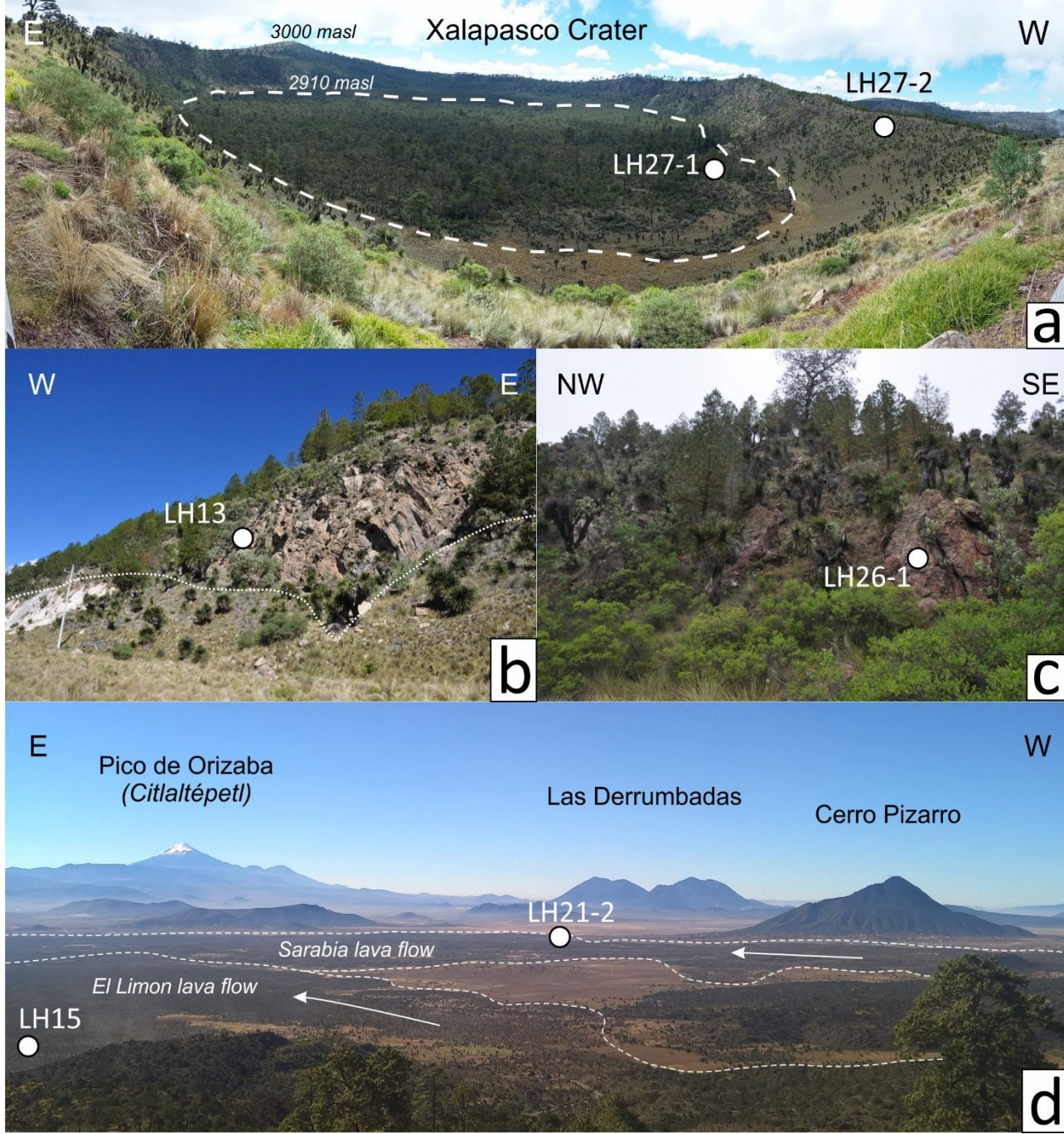


**Figure-2.** Field photographs of LHPCS volcanic products. (a) E-W panoramic view of Xalapasco crater; the
white dashed line indicates the limit of Cpx-bearing Ol-basalts lavas filling the crater. (b) Intra-caldera
trachyandesitic lavas outcropping at Los Potreros, south to Los Humeros town. (c) Trachyandesitic lava-
dome outcropping inside Los Potreros caldera, north to Xalapasco crater. (d) E-W panoramic view from the
SE Los Humeros caldera rim. With dashed lines are indicated the two major trachyandesitic lava flows of "El
Limón" and "Sarabia". Pico de Orizaba, Las Derrumbadas and Cerro Pizarro volcanoes are also indicated.
White dots indicate sampling localities.


**Figure 3**

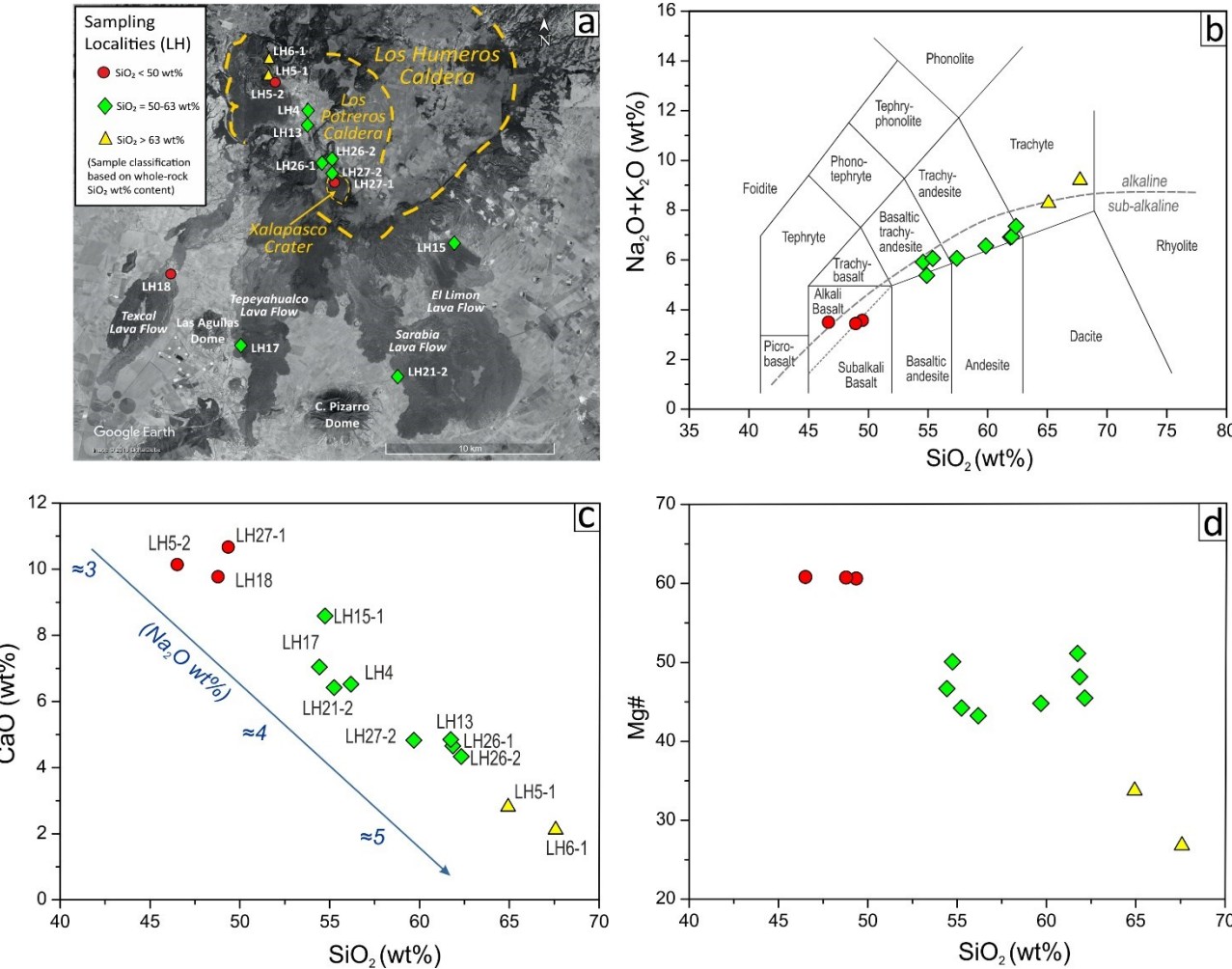


**Figure-3.** (a) Satellite image of the LHVC (Image Landsat from Google Earth Pro, 2018 Digital Globe;
courtesy of Google) with localization of samples selected for the application of Rayleigh Fractional
Crystallization model and for thermobarometry models. (b) Total alkali versus silica (TAS) diagram (Le
Maitre et al., 2002). (c-d) Major elements selected Harker diagrams for LHPCS studied lavas. The different
symbols (circle for basalt, diamond for trachyandesite and triangle for trachyte) represent the graphic code
that will be used coherently along the manuscript.


**Figure 4**

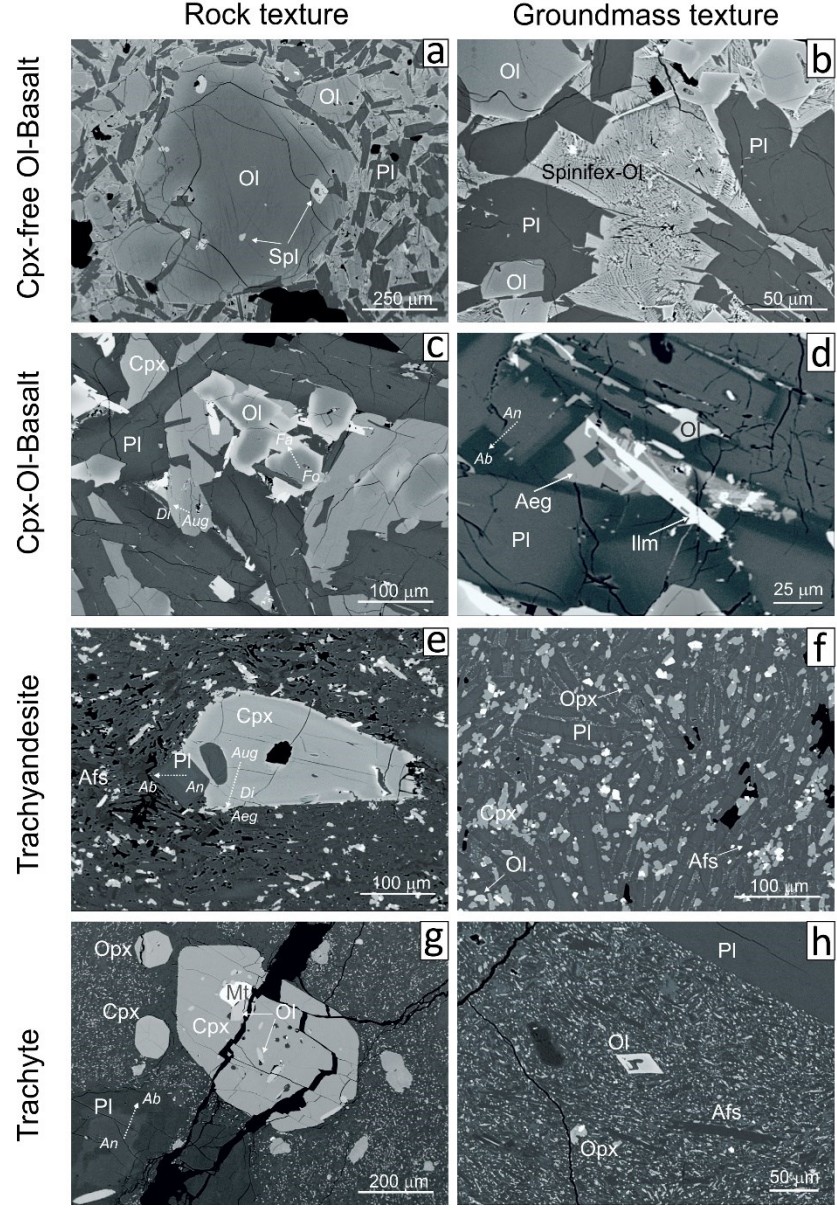


**Figure-4.** Microtextures and fabrics of the LHPCS lavas. (a) Back-scattered electrons (BSE) image of the Cpx-
free Ol-basalt fabric, dominated by euhedral unzoned homogeneous Pl+Ol, with major olivine phenocryst
characterized by Cr-Spl inclusions.  (b) BSE image of Cpx-free Ol-basalt groundmass highlighting the spinifex
to skeletal and dendritic crystallization of olivine, associated to the swallow-tailed morphology of
plagioclase. (c) BSE image of Cpx-bearing Ol-basalt. Normal monotonous zoning at rim is observed for all
the main mineral phases (Pl+Ol+Cpx). (d) BSE image of Cpx-bearing Ol-basalt groundmass characterized by
albitic plagioclase, aegirine-pyroxene, Fe-rich olivine and ilmenite. (e-f) BSE images of trachyandesites. It is
possible to observe a microcrystalline groundmass where major phenocrysts of Cpx and Pl are dispersed.
(g-h) BSE images of trachytes, characterized by a microcrystalline groundmass and Pl+Cpx+Opx phenocryst.
Plagioclase phenocrysts show normal monotonous to normal step zoning. Major Cpx phenocrysts present
inclusion of Ol+Mt.

 **Figure 5**

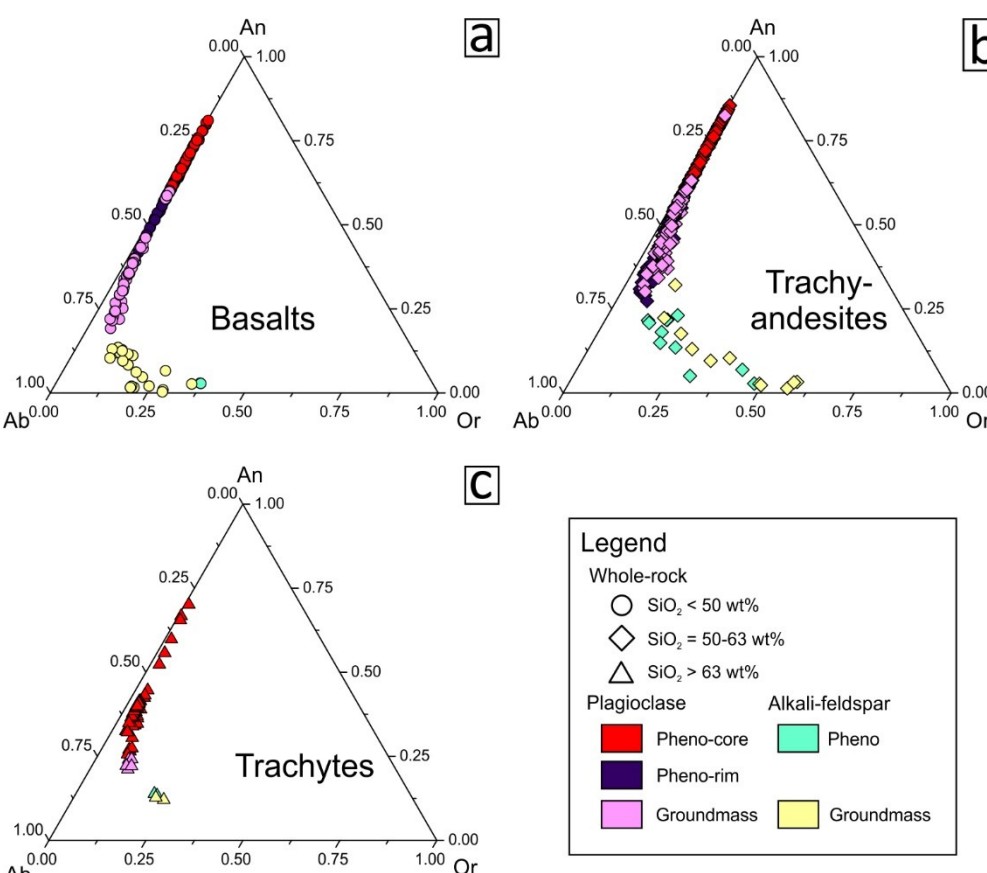

**Figure-5.** An-Ab-Or diagrams showing the composition of feldspar in (a) basalts (circles), (b) trachyandesites
(diamonds) and (c) trachytes (triangles) of LHPCS lavas.

**Figure 6**

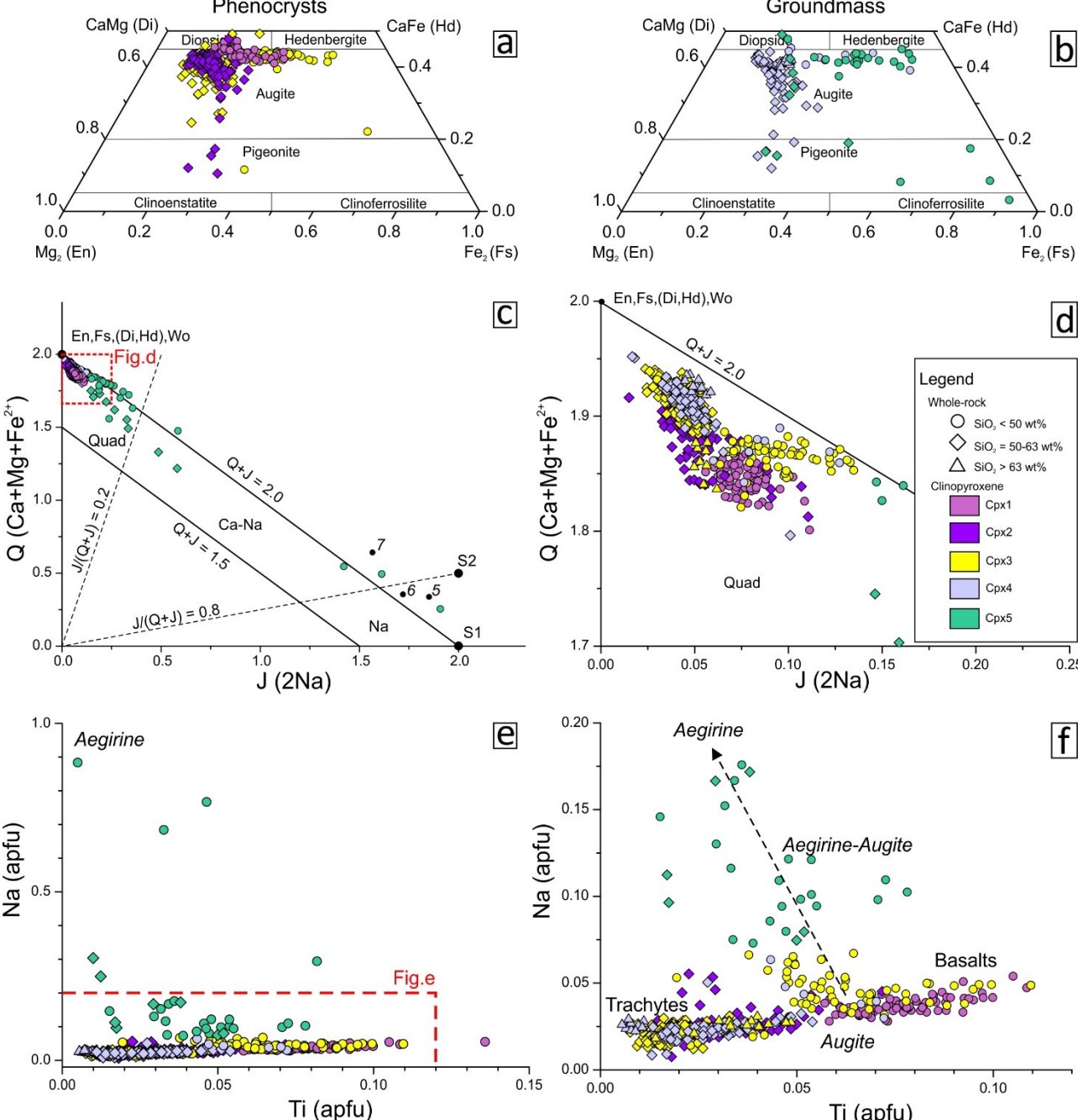


**Figure-6.** Di-Hd-En-Fs, Q-J and Ti vs. Na diagrams showing the composition of clinopyroxenes in LHPCS
lavas. Symbol shapes follow Fig. 3. (a) Di-Hd-En-Fs diagram for clinopyroxene phenocrysts (Cpx1, Cpx2,
Cpx3). (b) Di-Hd-En-Fs diagram for clinopyroxene microlites (Cpx4) and Na-clinopyroxenes (Cpx5). (c) Q-J
diagram for pyroxenes with indication of endmembers (Morimoto, 1989). (d) Enlargement of area indicated
in (c). (e) Ti vs. Na (apfu) diagram illustrating the compositional differences between clinopyroxenes. (f)
Enlargement of area indicated in (e), showing the main Augite trend characterizing the evolution from
basalts to trachytes and the divergent trend of Aegirine-Augite and Aegirine series.

**Figure 7**

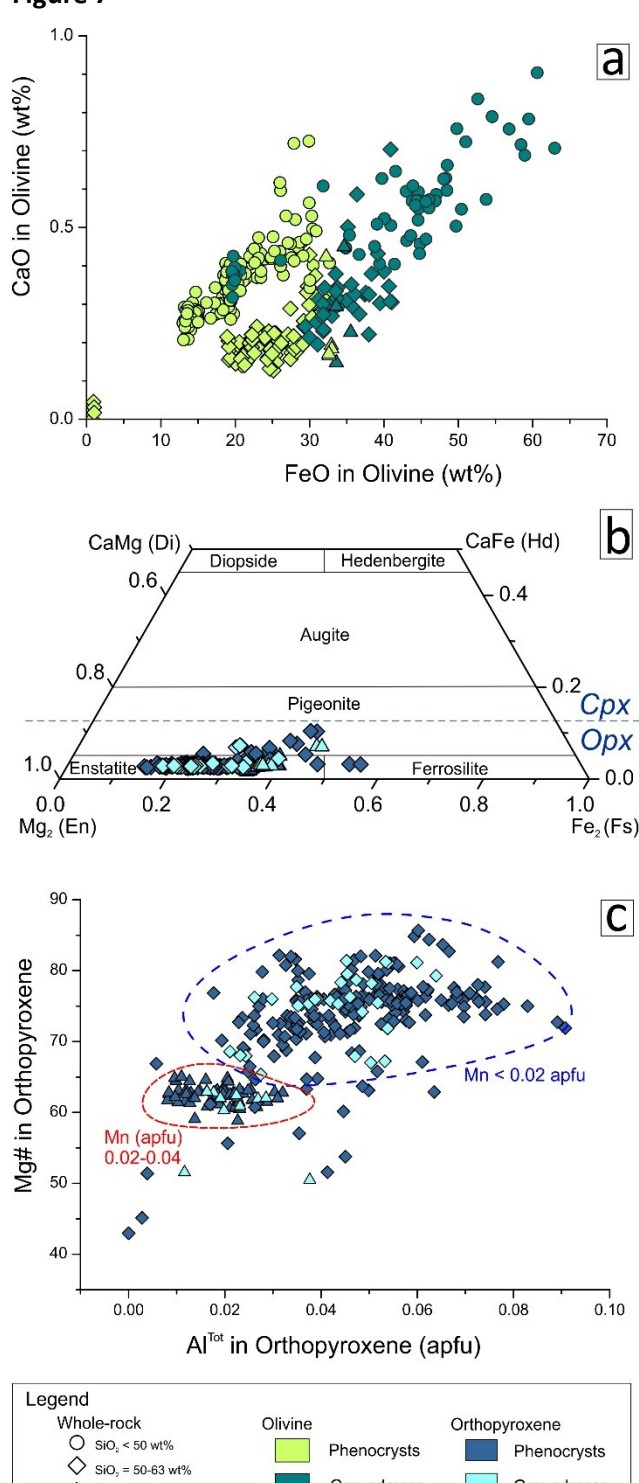


**Figure-7.** (a) CaO vs. FeO diagram showing the composition of olivine in LHPCS lavas. (b) Di-Hd-En-Fs
diagram showing the orthopyroxene chemistry in LHPCS studied lavas. (c) Al$^{Tot}$ vs Mg# diagram showing the
main compositional differences between orthopyroxene populations from trachytes and trachyandesites.
Mn (apfu) contents are also reported for the two populations.


**Figure 8**

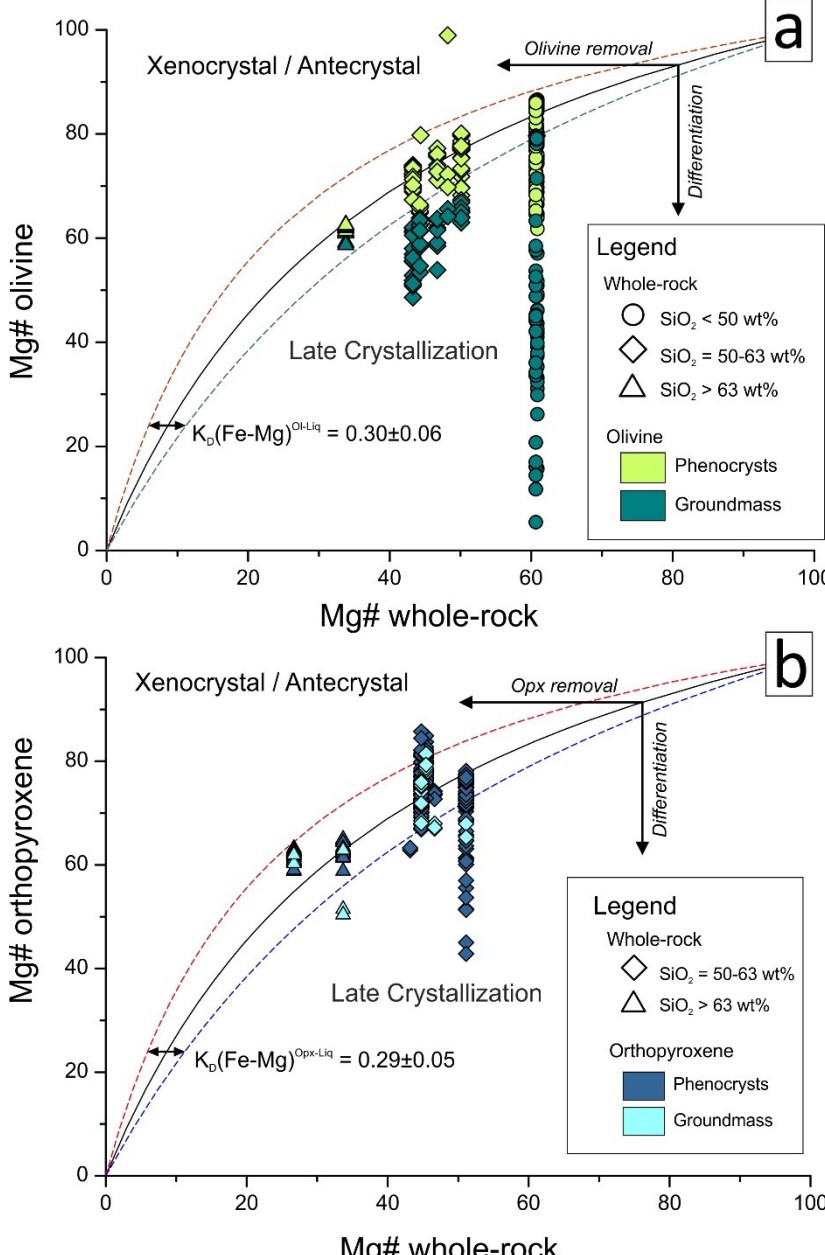


**Figure-8.** Rhodes diagrams showing the results of test of equilibrium liquid and olivine (a) and
orthopyroxene (b). The partitioning of Fe-Mg between mineral and liquid (Fe-Mg exchange coefficient) or
$K_D^{Min-Liq}$(Fe-Mg) is shown (black lines). The accepted range of equilibrium constant values for both figures (a)
and (b) is indicated by dashed lines. $K_D^{Min-Liq}$(Fe-Mg) values are from Putirka (2008). Nominal melt
compositions are selected from whole-rock analyses. Vectors of olivine and orthopyroxene removal from
melt and closed system differentiation are redrawn after Putirka (2008 and references therein). Fields of
"Xenocrystal/Antecrystals" and "Late Crystallization" are also indicated. Symbols and colors refer to Fig. 7.


**Figure 9**

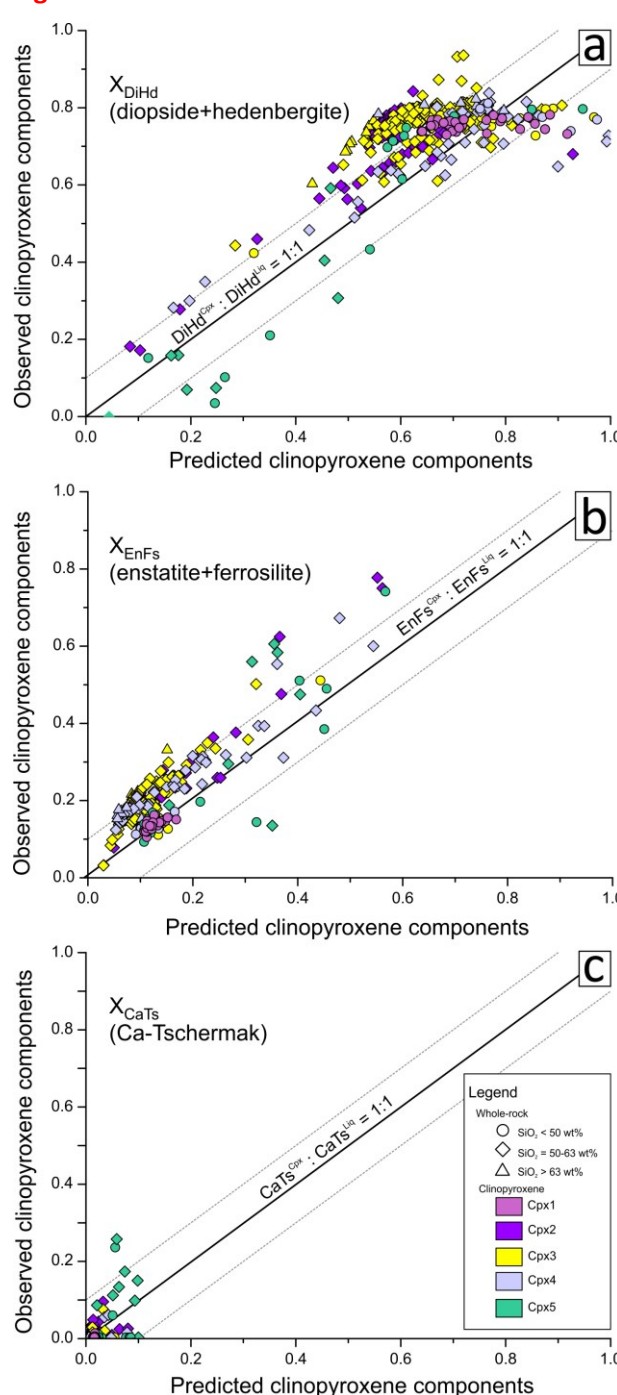



**Figures-9.** Clinopyroxene-melt equilibrium tests: (a) DiHd: diopside-hedenbergite, (b) EnFs: enstatite-
ferrosilite, and (c) CaTs: Ca-Tschermak components. Equilibrium associated with observed components in
pyroxenes are paired with predicted components in respective hosting-melts. The accepted range of
equilibrium is indicated in each figure by dashed lines. Nominal melt compositions for clinopyroxene are
selected from whole-rock analyses. Symbols and colors refer to Fig. 6.



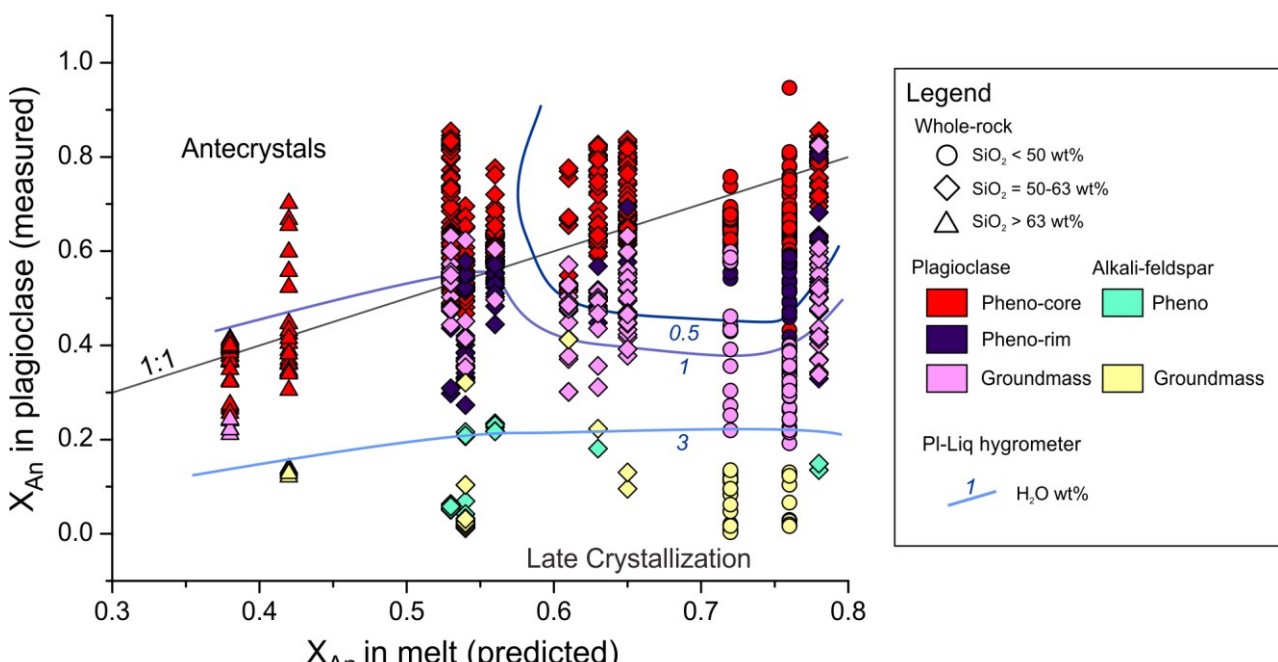


**Figure-10.** Plagioclase-melt equilibrium test. Equilibrium associated with anorthite ($X_{An}$) component in
plagioclase are paired with predicted anorthite in melt. Nominal melt compositions for plagioclase are
selected from whole-rock analyses. Calculated water concentrations using plagioclase-melt hygrometer
(Putirka, 2008) are reported in diagrams with isolines (graded blue lines). Symbols and colors refer to Fig. 5.

**Figure 11**

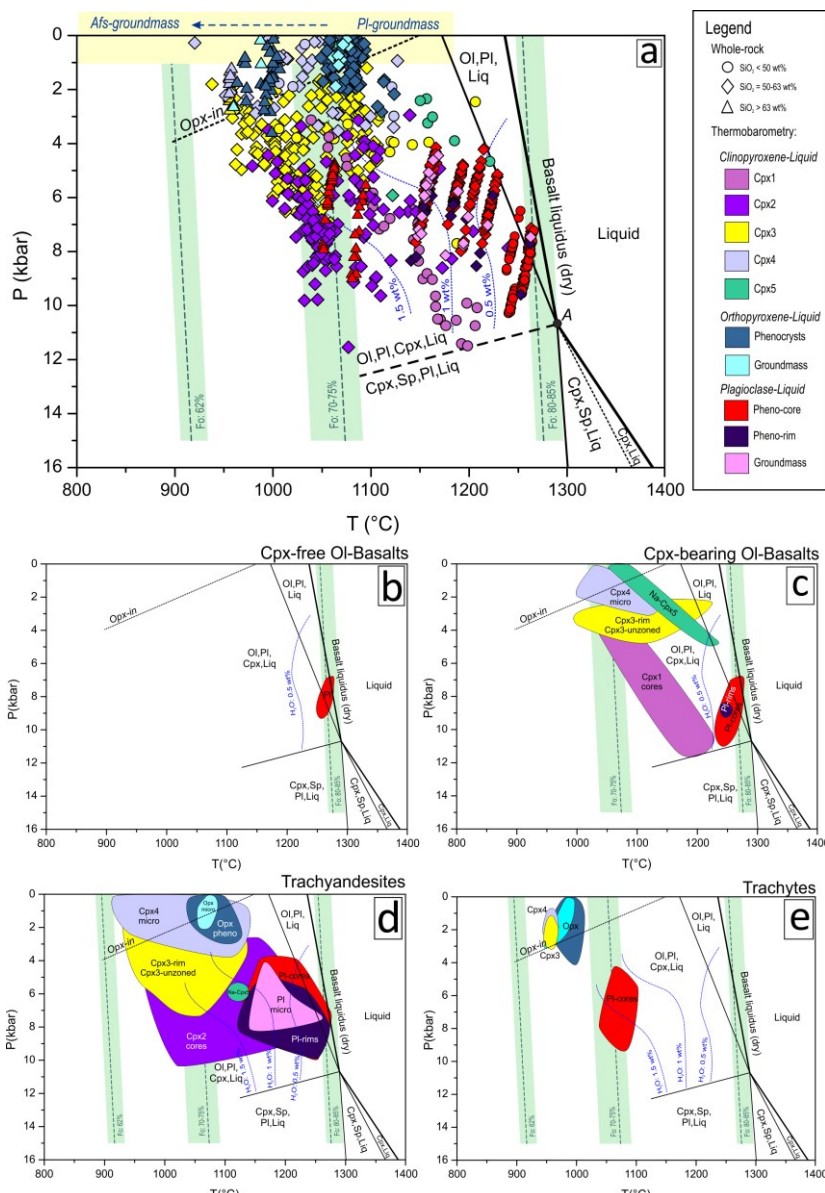


**Figure-11.** Thermobarometric estimates; a) A summary of the results obtained from thermobarometry
models applied to Los Humeros post-caldera stage lavas. Symbols refer to whole-rock chemistry
compositions, whereas colors of different phases refer to mineral chemistry diagrams. Green-shaded field
shows the results of olivine-liquid thermometry. Blue dashed isolines represent the results of plagioclase-
melt hygrometer. Yellow-shaded field indicates pressure-temperature domain of crystallization of feldspars
in groundmass. Basalt liquidus curve, Ol+Cpx+Pl+Sp+Liq stability fields and point "A" (basalt liquidus in
equilibrium with mantle peridotite mineral assemblage of Ol+Cpx) are redrawn after Grove (2000). Opx-in
stability curve is redrawn after Wallace and Anderson (2000). Schematized results are presented separately
for b) Cpx-free Ol-basalt; c) Cpx-bearing Ol-basalts; d) trachyandesites; and e) trachytes.

**Figure 12**

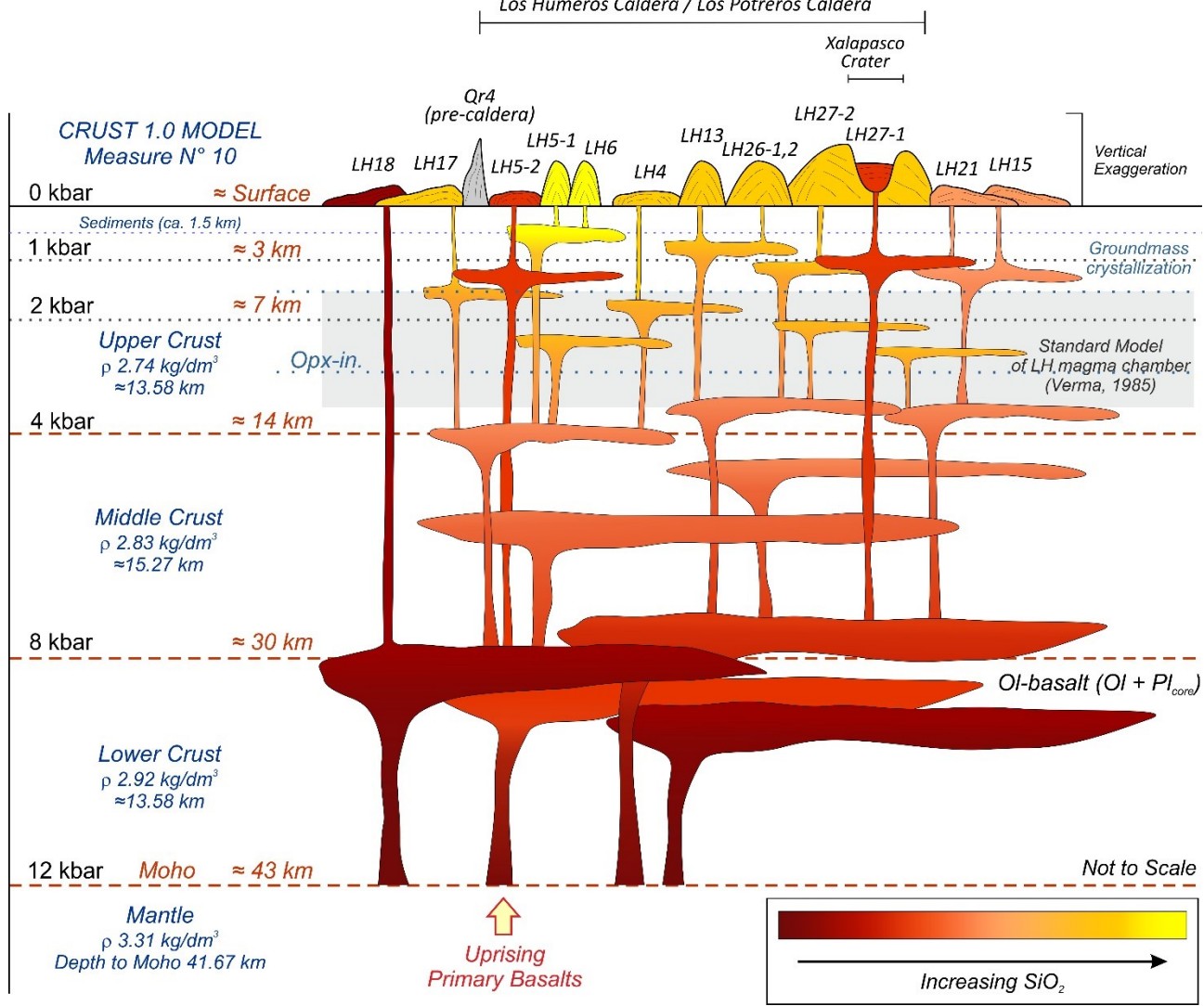


**Figure-12.** Schematic representation (not to scale) of the magmatic plumbing system feeding LHPCS
activity, beneath Los Humeros caldera as derived by pressure-temperature estimates obtained from
mineral-liquid thermobarometry models. The conceptual model is integrated with the crustal structure of
the study area as derived by the Measure N°10 of the Crust 1.0 global Model (Davies, 2013). Grey shaded
field indicates the depth and thickness of the existing conceptual model of a single, huge classical magma
chamber proposed by Verma (1985a, 1985b) and mainly related to the Los Humeros caldera-stage activity.

**Table 1**

**Table 1 - Major element bulk-rock compositions of LHPCS studied lava samples.**

| Rock type | Basalts | | | Trachyandesites | | | | | | | | Trachytes | |
|---|---|---|---|---|---|---|---|---|---|---|---|---|---|
| Sample | LH5-2 | LH18 | LH27-1 | LH17 | LH15-1 | LH21-2 | LH4 | LH27-2 | LH13 | LH26-1 | LH26-2 | LH5-1 | LH6-1 |
| $SiO_2$, wt% | 46.51 | 48.78 | 49.35 | 54.43 | 54.74 | 55.24 | 56.18 | 59.69 | 61.74 | 61.85 | 62.14 | 64.93 | 67.58 |
| $TiO_2$ | 1.471 | 1.490 | 1.372 | 1.394 | 1.075 | 1.561 | 1.375 | 1.016 | 0.882 | 0.889 | 0.933 | 0.738 | 0.605 |
| $Al_2O_3$ | 16.23 | 16.17 | 17.11 | 16.33 | 20.68 | 15.99 | 16.57 | 17.39 | 15.68 | 15.70 | 16.82 | 15.47 | 15.83 |
| $Fe_2O_3^{tot}$ | 10.78 | 10.62 | 10.26 | 8.08 | 6.49 | 8.62 | 7.88 | 5.76 | 5.15 | 5.22 | 5.32 | 4.58 | 3.73 |
| MnO | 0.161 | 0.160 | 0.155 | 0.123 | 0.092 | 0.133 | 0.114 | 0.087 | 0.085 | 0.085 | 0.095 | 0.077 | 0.074 |
| MgO | 8.44 | 8.29 | 7.97 | 3.57 | 3.28 | 3.45 | 2.90 | 2.36 | 2.72 | 2.45 | 2.24 | 1.18 | 0.69 |
| CaO | 10.14 | 9.77 | 10.67 | 7.04 | 8.59 | 6.42 | 6.52 | 4.83 | 4.85 | 4.66 | 4.52 | 2.81 | 2.12 |
| $Na_2O$ | 3.11 | 2.98 | 3.21 | 4.10 | 3.68 | 4.14 | 3.96 | 4.31 | 4.19 | 4.31 | 4.30 | 4.79 | 5.26 |
| $K_2O$ | 0.33 | 0.41 | 0.30 | 1.76 | 1.64 | 1.86 | 1.99 | 2.20 | 2.67 | 2.58 | 2.76 | 3.44 | 3.89 |
| $P_2O_5$ | 0.19 | 0.21 | 0.17 | 0.32 | 0.26 | 0.34 | 0.34 | 0.27 | 0.25 | 0.23 | 0.22 | 0.18 | 0.13 |
| LOI | 1.90 | 0.81 | -0.35 | 0.90 | 0.49 | 0.52 | 1.19 | 1.55 | 0.50 | 0.93 | 0.70 | 0.73 | 0.31 |
| Total (wt%) | 99.27 | 99.68 | 100.20 | 98.05 | 101.01 | 98.29 | 99.02 | 99.47 | 98.72 | 98.91 | 100.10 | 98.92 | 100.20 |
| Mg# | 61 | 61 | 61 | 47 | 50 | 44 | 43 | 45 | 51 | 48 | 45 | 34 | 27 |

Note: LOI - loss on ignition; Mg# - molar [Mg*100/(Mg + $Fe^{tot}$)].

