# Peer review of "Anatomy of the magmatic plumbing system of Los Humeros Caldera (Mexico): implications for geothermal systems"

_Solid Earth, 2019_

## Referee Comment (RC1) · Chiara Maria Petrone (Referee) · 16 Jun 2019

The authors present some new whole rock major elements and mineral chemistry data on a suite of rocks (from basalts to trachytes) of the post caldera activity of Los Humeros. Using a large amount of new mineral chemistry data, they calculate thermobarometric conditions for the Holocene plumbing system which allow them to put forward a new model in line with the recent idea of a trans-crustal plumbing system (e.g. Cashman et al., 2017).

I like the proposed model and how the thermobarometric constrains are put together with petrographic and mineralogical observations. However, there are aspects that

should be discussed in more details. In fact, the manuscript will benefit from a more in-depth discussion particularly regarding the implications on magma dynamics and the geothermal system, but also on other aspects such as the absence of magma mixing, the role of the crust and fractional crystallization. Some suggestions along these lines are: 1) A single liquid line of descent is inferred only from few major elements, namely $SiO_2$, Alk and CaO on a limited suite of rocks. This is a key aspect of magma evolution at Los Humeros, but it is not discussed thoroughly. Even when the connection with typical basalt-pantellerite is questioned, no further explanation is offered. For example, how do you explain the presence of aegirine augite given the lack of clear peralkaline composition? Magmatic diversity is really the results of simple fractional crystallisation? A more in-depth discussion on the origin an evolution of these suite of rocks is necessary. 2) The almost complete lack of magma mixing is remarkable, pointing to single batches of magma evolving in closed system. I assume these are completely evacuated during eruption and presumably not reactivated(?). This is a quite unusual dynamics for a caldera and different from previous activity with large ignimbrite. It can also have important implication on the geothermal system. It would be good to see a more in-depth discussion on these points and the implication on magma dynamics. 3) The area is characterised by $\sim$ 30 km of crust, but the role of the thick crust is not explored. 4) From the model it is seems that trachyandesites and trachytes form at 30-15 km in the second stagnation level (L774-790). Here it is claimed that pl+cpx2 + microlites form. If so it means that the liquid is not changing composition at shallower level. Therefore, the liquid must pond at the third stagnation level (15-10 km, cpx3) and at shallow level (7-3 km, cpx4, cpx5, Fe-olv and opx) for very short time and possibly shortly before the eruption. What are the implications for magma dynamics and for the geothermal system? 5) The paragraph on the implication for the Los Humeros geothermal system is a repetition of the proposed model of the plumbing system rather than a real discussion of the implication of the model. This part should be used to actually discuss how the geothermal system can be sustained in the light of the new model of small single-charge of magmas at different level (from 30 to 15 km upward)

and ephemeral pockets at shallow level. Is the current geothermal system different from the pre-caldera stage? If so, is there any inference that can be drawn?

The manuscript will also benefit from some reorganization. In particular, the thermbarometric section should be separate from the discussion. In fact, at the moment it is quite a long section, not extremely clear, with a long list of P-T conditions that are quite hard to see in Fig. 11. This part should be dedicated to present the new T-P data and discuss the discrepancy between different themobarometry (olv-liq gives always lower T, plg-liq higher T). The discussion should start at section 7.4.

The authors talk about "inverse thermobarometric modelling" (in the abstract and in various part of the text). However, the standard mineral-liquid approach is used throughout, which in literature is commonly referred to as mineral-liquid thermobarometry not as "inverse". A clarification on this point is necessary.

It is stated that the "this study attempts to emphasize the importance to integrate fieldpetrography, texture observations and mineral chemistry of primary minerals to unravel the pre-eruptive dynamics". However, the field evidence is quite sparse and not used to support the data and conclusions.

Overall, I recommend the manuscript for publication pending moderate reviews.

Kind regards, Chiara Maria Petrone

Specific comments along the text.

Major elements – the description of the major elements characteristics is quite redundant and so are the presented diagrams in Fig. 3. The TAS diagram with the subalkaine/alkaline division line (which is missing, why?) would be totally sufficient to inform the reader of the chemical composition of these rocks. Fig. 3c and d don't add any further information. The text is unnecessarily wordy accounting for the range of variability of SiO2, NaO2, K2O etc that we can clearly see in the diagram. It would be more useful to show the harker diagrams alongside the TAS. In addition, I find the

choice of symbols quite confusing. In fact, the information on the SiO2 contents is already shown in the x-axis, therefore the symbols are a simple repetition of the same information.

Petrography – it is not clear why the Texacal lava flow (Fig. 3) is called olivine basaltic lavas (Fig. 1) when olivine is always present as phenocrysts. Only 3 basalts and 2 trachytes are described which makes quite hard to attach too much meaning to the presence or absence of cpx (basalts) and the "substantial" difference in mineral assemblages of the trachytes. I suggest to be more careful in drawing conclusion from such a small number of samples.

Mineral chemistry. I find the section on cpx section very confusing and unnecessarily complicated with 5 different categories of cpx. The zoning pattern is completely lost. In fact, cpx 3 overlaps cpx1 & 2, therefore it is difficult to see if cpx in basalts and trachytes are actually zoned. If so how? The other interesting feature, the presence of microlite and outer rims of aegirine-augite composition, is hard to correlate to specific composition and/or location. Are the cpx microlites in basalts all cpx5? It is well expected that Fo and Mg# decrease from basalts to trachytes, so this is not really a good criterion for discriminating crystal population. How about the behaviour of trace elements such as Cr and Ni?

7.3.2 this part is very difficult to follow, it is a list of P-T condition very difficult to visualise. How do the reader will figure out the difference between opx-free and opx-bearing trachytes if they are not distinguishable in Fig. 11? Please rewrite this section.

Fig. 11 - The difference between basalts, trachyteandesites and trachytes is almost lost in this diagram, particularly for the trachytes. Maybe Fig. 11 can be redrawn and composed of 4 separates diagrams showing the inferred P-T condition for basalts, trachyandesites and trachytes. All the results can be shown together in the last diagram colour coded for composition. It would be easier to see where the different rock types crystallise and this should also be used in the discussion.

L64-68 - The recent work of Jackson et al Nature 2018 show a numerical modelling for the network of melt and mush.

L261-262 – It is not clear what do you mean with a lava flows being free of lithics.

L366 – It is quite interesting that plagioclase cores are more mafic in trachyandesites than in basalts. Why?

L432-441 – No need of this introduction

L472 - you haven't presented Fig2. 7-10 yet

L482-484 – please revise the English.

L485-487 – Strongly oscillatory zoning can be due to different causes mainly magma mixing and/or P-T-H2O fluctuation, it reflects the dynamics of magmatic processes and might be not an indication of equilibrium. On the contrary the low-amplitude oscillatory zoning is kinetically controlled (see Streck 2008 for further details). Therefore, it is important to stress which type of oscillatory zoning the authors refer to. In addition, the fact that the zoning is oscillatory precludes that core and rim formed in evolving liquids with progressively different compositions as argued.

L488-489. The use of WR composition instead of glass is problematic especially when dealing with microlite.

L502-503 – On which basis this additional criterion has been chosen? The test for equilibrium according to Putirka 2008 is satisfied when KD(Fe-Mg)opx-liq = 0.29±0.06.

L504-514 – This part is quite confusing. The Rhodes diagram (Fig. 8) are for olv and opx, eq. 17 of Putirka is for olivine, how the cpx-liq fits with this?

Appendix – what is the error on EPMA data? How Actlabs and XRF data compare?

---

## Referee Comment (RC2) · Anonymous Referee #2 · 25 Jul 2019

Comments on: Anatomy of the magmatic plumbing system of Los Humeros Caldera (Mexico), implications for geotermal systems Federico Lucci, et al., The authors did a great analytical job and their proposal about the multi-reservoir model is in accordance with recent investigatios. However, I think data could be used in more detail to explore other scenarios. The main idea will remain, the polybaric model seems unrefutable, but the processes involved during the evolution of magmas could be an important factor for the geothermal system. The model proposed by the authors rely in only one differentiation process; crystallization. Whereas mixing, and perhaps assimilation, seems to be important processes. Something should be explained about the origin of these magmas. Once explained, you can go further and propose how do you think they evolved.

[Figure]

I think mixing is evident, for example:

1) TAS diagram looks very linear. Find published liquid lines of descent for basalts evolving to trachytes just by crystallization. I think they do not look like the trend displayed by your samples.

2) Plagioclase microlites and phenocryst rims show a very wide compositional range An20-63. Is evident this are not in equilibrium.

3) The presence of Cr-rich titanomagnetites and Cr-poor titanomagnetites in the same sample. In fact, I suggest to remove the ulvospinel data, is of bad quality. Either, the crystals were very small and you excite the surrounding matrix or something failed with the standarization. Although data is not publishable, the relative abundance of Cr is evident and you should explain it.

Thermobarometric results should be used with caution. Its hard to match the results reported on the text with the supplementary data. But for example plagioclase data of almost the same composition found in different WR samples is used to calculate temperature and pressure. These models almost always will yield a number, the idea is to generate a good interpretation for the results, the best possible approximation. Even if is not evident on their tests, the authors should incorporate explanations on how mixing could affect their thermobarometry results. Protracted heating-mixing could be the driving force for convection-conduction in a geohermal reservoir. Figure captions should be more descriptive and informative.

The manuscript is overall well written. Below you will see more detailed comments.

L351-L353. Here you highlight the dissolution on pyroxene. You should do the same for plagioclase.

L368-L374. Aegirine is an index mineral on peralkaline rocks. The same for anorthoclase. You reported non peralkaline rocks occur in the studied area. Mixing processes should be explored. The wide compositional range on "microlites" reported above could

suggest the same mixing process.

L389. Fayalite is present, same comment as above. Fayalite occur in peralkaline rhyolites.

L446. It's correct to use the WR; however, you need first to define why this is valid if mixing could modified some magmas. The whole rock would simply be an integrated result of all magmatic processes (mixing+assimilation) that occured just before the eruption.

L484. This sentence is not clear. A crystal could have a patchy zone at the core and then be in equilibrium with the melt (rim). In your sentence you should clarify or have a reference to one of your figures.

L485-L487. Not neccesarily; just changes in temperature would record extreme compositional changes in crystals, without any mixing involved (mass exchange).

L488-489. First you need to clarify and yield some confidence to the reader about the origin of the melts and the magmatic processes that modified each magma. If mixing occurred, then the WR is a mixture of xenocrysts and phenocrysts. How does this affect the equilibrium between the phenocrysts and the WR? You should clarify the phenocrysts you used, is not clear and not proven the criteria to choose them. ". . .. . .. pristine liquids in equilibrium with early crystallized phenocrysts and microlites"? This is confusing; What dou you mean as microlites??, decompression induced crystals grown during decompression-cooling? Or these are microphenocrysts?? Do you mean all melts where tapped as they were formed?, I mean, how is posible that a liquid is in equilibrium with an early crystallized phenocryst and a microlite??

L502-503. You need to explain why this value was chosen.

L550. You should delete the negative values. Have no petrological meaning, no matter other authors have interpreted them as anhydrous. All models will yield data, the job of the petrologist is to evaluate which are usable.

L551-563. Does these water contents match the pressure calculated with other methods used in this investigation? 1.40 wt.% is very shallow. If these are subduction related magmas how is possible they are anhydrous? I know is dictated by the model, but what do you think, the model approach well the problem?? Moreover, if basalt are required to be anhydrous, then what is the origin of the water required for the evolved LHPCS?? Have you try other hygrometer, different to Lhur and Housh 1991,?

L567-569. You have a great amount of xenocrysts with felsic compositions. Many of the cores where temperature was calculated have the same composition as other cores measured in TA. The temperature is different because the WR composition in which are supposely in equilibrium varies. So, which one is the system in equilibrium??

L634-637. Are these megacrysts or resorbed crystals?, fragmented crystals??. Is there any possibility these are intrusive xenocrystals??. Give a reference for subsolidus equilibration of groundmass after eruption or explore an alternate possibility.

L650-653. This is confusing, or at least, with not enough information for the reader in order to understand the author's point of view. Are these basalts the product of differentiation of the former basalts mentioned just before? These evolution is recorded along the crystallization of the Fe-rich olivine, albites and aegirines? Report the textura of the borders, are they in equilibrium. Is this alkaline-low oxygen fugacity mineral assemblage the result of mixing with more alkaline melts? Mixing could occurred at depth or at shallow pressures??

L657-658. Then , what happened to the proposal that basalts should be anhydrous?

L670-673. I suggest a brief proposal about the origin of these magmas should be explained. If mafic and intermediate are tapped almost straight from deep reservoirs, then what is the origin for felsic magmas?, do they arrived already evolved to shallow levels? Moreover, the intermediate melts do not have any traces from mixing? Mixing has been reported as one of the mechanisms to produce andesite-dacite melts.

L674-683. I disagree. You need to explain why some TA and AB have a mixure of restitic phases as ulvospinel (titanomagnetite)? What I mean is that some samples contain evolved titanomagnetites and Cr-rich titanomagnetites. This is a strong evidence for mixing.

L682-L683. I do not understand what you try to explain. Rephrase.

L694-695. These are not the only evidence for mixing. L715. If I remember well this is the first time alkali-basalt is mentioned. Should be pointed before, this would explain the origino f aegirine anf fayalite???

L717-718. Harker diagrams are not the best option to explain fractional crystallization. In fact, it seems that the trends are very linearn typical of binary mixing-assimilation. Lets think about the mixture of Fe-Ti oxides you have, mixing occured. A very least you should show trace element evidence;. Trace elemnts are very sinsitive to mixing and assimilation, so you could adjust your model.

L721-724. Then, how does this support crystalization acting alone to form the trachytes??

L750. So what is the origino f these anhydrous basalts? If these evolved to more felsic melts what is the origin of water in those?

L806-813 This paragraph is very confusing. Do you mean this alkaline basalt will arrive almost at its liquidus at shallow pressure, and only then aegirine and fayalite would crystallize?? What is the origino f this alkaline basalt? He basalt has ilmenite-ulvospinel?, calculate the oxygen fugacity and discover is is enough reduced to crystallize fayalite-aegirine.

Figure 3. There are multiple attempts to recreate liquid line of descents for melts evolving by fractional crystallization from basalts with these compositions. Search and try to fit your data. To me look very linear and probably related to mass-addition processes.

Figure 5. Basalts, trachyandesites and trachytes contain anorthoclase as phenocrysts

and in the groundmass. I do not remember an explanation about this in the text. Could be mixing with alkaline melts??

Please also note the supplement to this comment:
https://www.solid-earth-discuss.net/se-2019-86/se-2019-86-RC2-supplement.pdf

---

## Author Comment (AC1) · 9 Oct 2019

REVIEW by Referee#1 (Dr. Chiara Maria Petrone)

We thank Dr. C.M. Petrone for the critical and constructive comments that deeply contribute to improve our manuscript. In the new version of the manuscript we have taken into accounts all the received comments and advice. We detail below how each comment was addressed and considered in the revised version (the Referee's comments typed in italics and our responses typed in red).

**Major comments**

*In fact, the manuscript will benefit from a more in-depth discussion particularly regarding the implications on magma dynamics and the geothermal system, but also on other aspects such as the absence of magma mixing, the role of the crust and fractional crystallization. Some suggestions along these lines are: 1) A single liquid line of descent is inferred only from few major elements, namely SiO2, Alk and CaO on a limited suite of rocks. This is a key aspect of magma evolution at Los Humeros, but it is not discussed thoroughly.*

We agree with Dr. C.M. Petrone that defining the line of descent only with Harker major elements is not enough. Therefore, we use major element mass balance models (Bryan et al., 1969) to test the hypothesis of differentiation via Fractional Crystallization (FC) following the scheme proposed in White et al., (2009), Moghadam et al. (2016) and Lucci et al. (2016).

In particular, using the Rayleigh Fractional Crystallization (RFC) model (e.g., White et al., 2009) we demonstrate that a progressive fractionation of Pl+Cpx+Ol+Spl assemblage is able to produce all trachyandesites and trachytes, starting from the most mafic Cpx-bearing ol-basalt (LH5-2).

Mass balance modeling and in particular crystal accumulation ("Cumulate" model in White et al., 2009; Lucci et al., 2016) demonstrate the genetic linkage existing between all LHPCS basalts. A progressive appearance of Cpx and accumulation of Pl+Cpx+Ol+Spl is able to produce Cpx-bearing basalts (LH5-2 and LH27-1) from the LH18 Cpx-free basalt.

The obtained results are in agreement with the improved textural observations and describe LHPCS magmas as melts belonging to a single line of descent and differentiated via fractional crystallization.

The results obtained through FC-Models are presented in the new chapter "8.1 Major-elements mass balance modeling" and in Supplemetary Table 6.

*Even when the connection with typical basalt-pantellerite is questioned, no further explanation is offered. For example, how do you explain the presence of aegirine augite given the lack of clear peralkaline composition? Magmatic diversity is really the results of simple fractional crystallization? A more in-depth discussion on the origin an evolution of these suite of rocks is necessary.*

We agree with the Reviewer's comment. The understanding of the origin and the evolution of the Los Humeros post caldera stage magmatism, would requests a deeper petrological approach, based on the integration of mineral chemistry, whole rock geochemistry (major+trace+REE elements) and Sr-Nd isotope ratios. However, unravelling the petrogenetic evolution of this active caldera is far beyond the scope of this manuscript, that is focused, instead, on the reconstruction of the anatomy of its magmatic plumbing system.

Therefore, we deleted Lines 718-724 (of the original manuscript): "Intra-caldera basalts show assemblages containing Aegirine-rich clinopyroxenes (Fig. 4 c, d), widely considered as one of the most reliable indicator of magma transition to peralkaline conditions (i.e., White et al., 2009; Melluso et al., 2014). However, absence of olivine in the most evolved LHPCS trachyte (LH6) excludes, at this stage, a connection with typical basalt-pantellerite suites where tephroitic-fayalite is commonly found in high-silica rocks (i.e., White et al., 2005; Ronga et al., 2010; Macdonald et al., 2012, Melluso et al., 2014 and references therein)".

We deleted also Lines 253-256 (of the original manuscript): "The Los Humeros felsic (i.e., SiO2 >63 wt%) lava samples belonging to the post-caldera stage (i) fall in the "Trachyte" fields, (ii) show potassic signature and (iii) are characterized by Agpaitic Index (molecular ratio

[(Na2O+K2O)/Al2O3]) values < 1 (range: 0.7-0.8) thus excluding a peralkaline character of these evolved melts."

Coherently, we modified the Chapter 4 (Lines 232-251 of the revised manuscript).

Concerning the finding of aegirine-augite cpx and fayalite olivine in LH5-2 and LH27-1 basalts, their presence is in agreement with the transitional- to alkali-basalt character identified in TAS following the criteria proposed by the existing literature (reported in manuscript at Lines 241-243 of the revised manuscript).

The new FC-models (new paragraph 8.1, Lines 654-703 of the revised manuscript), together with mineral chemistry and the improved textural observations, indicate that the magmatic diversity is the result of a magmatic evolution dominated by fractional crystallization processes.

Since the aim of the manuscript is to demonstrate the possibility to rebuilt the anatomy of the feeding system using the crystal-melt pairs (as in Stroncick et al., 2009; Aulinas et al., 2010; Dahren et al., 2012; Keiding and Sigmarsson, 2012; Coombs and Garner, 2014; Barker et al., 2015), we think that a deeper exploration of the magmatic diversity (i.e, petrology and petrogenesis of LHPCS magmas) is far beyond the scope of this work, while it clearly represents the core for a future paper.

*2) The almost complete lack of magma mixing is remarkable, pointing to single batches of magma evolving in closed system.  I assume these are completely evacuated during eruption and presumably not reactivated(?). This is a quite unusual dynamics for a caldera and different from previous activity with large ignimbrite. It can also have important implication on the geothermal system.  It would be good to see more in-depth discussion on these points and the implication on magma dynamics.*

We are glad that Reviewer Dr. Petrone completely agrees with our findings and interpretations, because this comment is totally in line with our evidence and conclusions.

Concerning the Los Humeros dynamics, this scenario is coherent with other postcaldera eruption behaviors, as it happened in Ischia (see Casalini et al., 2017). We integrated this important observation in the revised manuscript at Lines 969-973 (of the revised manuscript)

We agree with the Reviewer on the necessity to stress this point in the discussion. We better explored this point in the new Chapters 8.4 and 8.5.

We will add these considerations about magma dynamics in discussion, together with answer to comments 4 and 5.

*3) The area is characterised by ~30 km of crust, but the role of the thick crust is not explored.*

Regarding this comment, we must say that the Trans Mexican Volcanic Belt in the study area shows a crustal thickness of ca. 40-42 km (not 30 km). The few (only) reasonable crustal data come from Crust 1.0 Global Model (Dziewonski and Anderson, 1981; Davies, 2013), which is a list of thicknesses, densities and Moho depths based on seismic information (Lines 735-747 of the original manuscript).

We integrate the thermobaric estimates from our work to the crustal properties as derived from the Crust 1.0 Global Model to convert the obtained pressure estimates into depth values.

We also proposed in the original manuscript (Lines 840-844) that density contrasts between different crustal layers act as a controlling parameter for ascending or stalling magmas. Moreover, it is possible to propose that each of these crust/density boundaries would determine lateral transport and grow of magma stagnation pockets.

*4) From the model it is seems that trachyandesites and trachytes form at 30-15 km in the second stagnation level (L774-790). Here it is claimed that pl+cpx2 +microlites form.  If so it means that the liquid is not changing composition at shallower level.  Therefore, the liquid must pond at the third stagnation level (15-10 km, cpx3) and at shallow level (7-3 km, cpx4, cpx5, Fe-olv and opx) for*

*very short time and possibly shortly before the eruption. What are the implications for magma dynamics and for the geothermal system?*

The new major elements mass-balance modeling (see new section 8.1) demonstrates that Los Humeros post-caldera magmas are cogenetic and all belong to a single line of descent. Thermobarometry models indicate that all intermediate and felsic melts are generated in a polybaric magmatic system, vertically distributed in the whole crust beneath the caldera. The improved description of the phenocryst zoning textures and magmatic fabric (see new sections 5 to 8) demonstrate that LHPCS mafic and intermediate erupted magmas are characterized by kinetically dominated, fast-growth in a closed system during a rapid ascent up to the shallowest plexus prior to eruptions.

A full comprehension of magma dynamics in the LHPCS plumbing system would request a different research approach (magma-density, magma-viscosity and P-T relationship, degassing rate, ascent rate) that, at this stage, is out-of-the scope of the manuscript.

However, we agree that preliminary implications on magma dynamics could be recovered by textural (phenocrysts morphology, zoning textures, microlites, presence of vesicles) observations. Accordingly, we modified and readapted the new sections 8.2, 8.4 and 8.5.

*5) The paragraph on the implication for the Los Humeros geothermal system is a repetition of the proposed model of the plumbing system rather than a real discussion of the implication of the model. This part should be used to actually discuss how the geothermal system can be sustained in the light of the newmodel of small single-charge of magmas at different level (from 30 to 15 km upward) and ephemeral pockets at shallow level. Is the current geothermal system different from the pre-caldera stage? If so, is there any inference that can be drawn?*

Concerning the LH geothermal system, the main outcome of our work is that the hypothesis of the single large and voluminous, shallow magmatic chamber, which was proposed for the caldera stage (lasted ca. 130 ka, consisting of two major caldera-forming events: Xaltipan and Zaragoza ignimbrites, 115Km3 and 15Km$^3$ DRE, respectively, Carrasco-Núñez and Branney, 2005; Carrasco-Núñez et al., 2018), is no longer active in the Holocene. We, in contrast, are in favor of a more reliable scenario characterized by a polybaric magmatic system feeding shallow magmatic transient batches of different magmas localized beneath Los Humeros nested caldera, having a high output rate of spatially-distributed small-volumes of mafic to felsic products.

This result is presented at lines 860-875 of the original manuscript. However, we agree with reviewer's comment about the necessity to stress up the main implications for the geothermal system. We improve the Discussion and we explore the resonance of changing scenario in the new sections 8.4 and 8.5.

*The manuscript will also benefit from some reorganization. In particular, the thermobarometric section should be separate from the discussion. In fact, at the moment it is quite a long section, not extremely clear, with a long list of P-T conditions that are quite hardtop see in Fig. 11. This part should be dedicated to present the new T-P data and dis-cuss the discrepancy between different themobarometry (olv-liq gives always lower T, plg-liq higher T). The discussion should start at section 7.4.*

We agree with the reviewer's suggestion.

Section 7 (up to section 7.3) now are entitled as "Thermobarometric Estimates". The Discussion section (new section 8) starts now from the original section 7.4. Furthermore, section 8 is now divided in the following sub-sections: (i) 8.1 Major-elements mass balance modelling; (ii) 8.2 Magma evolution beneath Los Humeros; (iii) 8.3 The magma plumbing system; (iv) 8.4 "Standard" vs. multilayered magmatic plumbing system; and (v) 8.5 Implications for the active geothermal systems.

We concur that Fig.11 is too rich of information. We modified it and we propose a new Fig. 11 as follow: (i) Fig 11a is the original diagram with all the thermobarometric results plotted; and (ii) we

add Figs 11b, 11c, 11d, and 11e, for cpx-free basalts, cpx-bearing basalt, trachyandesites, and trachytes, respectively.

Concerning the discrepancy between thermometers, we must clarify that Ol-Liq gives not always temperatures lower than Pl-Liq models. At lines 572-576 of the original manuscript, Ol-Liq and Pl-Liq models applied to basalt produce comparable results.

Concerning trachyandesites and trachytes we improved thermo-baric data presentation and discussion (see lines 627-631 and 643-649 of the revised typescript), respectively.

*The authors talk about "inverse thermobarometric modelling" (in the abstract and in various part of the text). However, the standard mineral-liquid approach is used through-out, which in literature is commonly referred to as mineral-liquid thermobarometry not as "inverse". A clarification on this point is necessary.*

We agree, the Reviewer is right. We now use the correct term "mineral-liquid thermobarometry".

*It is stated that the "this study attempts to emphasize the importance to integrate field-petrography, texture observations and mineral chemistry of primary minerals to unravel the pre-eruptive dynamics". However, the field evidence is quite sparse and not used to support the data and conclusions.*

We agree, we stressed too much the sentence. We modified the text accordingly.

**Specific comments along the text**

*Major elements – the description of the major elements characteristics is quite redundant and so are the presented diagrams in Fig. 3. The TAS diagram with the subalkaine/alkaline division line (which is missing, why?) would be totally sufficient to inform the reader of the chemical composition of these rocks. Fig. 3c and d don't add any further information. The text is unnecessarily wordy accounting for the range of variability of SiO2, NaO2, K2O etc that we can clearly see in the diagram. It would be more useful to show the harker diagrams alongside the TAS. In addition, I find the choice of symbols quite confusing. In fact, the information on the SiO2 contents is already shown in the x-axis, therefore the symbols are a simple repetition of the same information.*

We do not completely agree with this comment.

Major elements geochemistry is a very short chapter of ca. 20 lines (Lines 232-251 in the revised manuscript). We are just describing the whole rock chemistry (of the studied rocks) that is used as nominal liquid in the mineral-liquid thermobarometry models, and now (in the revised manuscript) in the new FC-models.

Diagram 3c and 3d are simplified representative Harker diagrams. X axis is $SiO_2$ content as differentiation index. Diagram 3c is CaO vs $SiO_2$. The blu vector in diagram 3c represent the trend of Na2O with respect to SiO2. Diagram 3d is Mg# vs $SiO_2$, it represents the evolution of Mg# i.e. the evolution of the existing relationship between MgO and FeO*.

Concerning the subalkaline/alkaline division line, it was a mistake, but we correct this. It is missing due a wrong export from CorelDraw to JPG.

Concerning the choice of symbols, there was a precise strategy that can be followed along all the manuscript: (i) we chose the three different colors to help the reader to visualise the different magmas (red= basalts, green= trachyandesites and yellow= trachytes) as depicted Fig. 3a of the submitted typescript; (ii) we chose three different symbols (circle for basalts, diamond for trachyandesite, triangle for trachytes) as a code for all the following diagrams. In mineral chemistry diagrams (Fig. 5, 6, 7), in all the P-T model diagrams (Fig. 8, 9, 10), and in the final P-T model (Fig. 11), the reader can always recognize basalts (circle), trachyandesites (diamonds) and trachytes (triangle).

Modifying Fig. 3 would imply to modify the graphic code of all the other diagrams.

We agree to the reorganization of the description of the major element geochemistry, but we therefore prefer to maintain Fig. 3 as it is.

*Petrography – it is not clear why the Texacal lava flow (Fig. 3) is called olivine basaltic lavas (Fig. 1) when olivine is always present as phenocrysts. Only 3 basalts and 2trachytes are described which makes quite hard to attach too much meaning to the presence or absence of cpx (basalts) and the "substantial" difference in mineral assemblages of the trachytes. I suggest to be more careful in drawing conclusion from such a small number of samples.*

Concerning the Texcal lava flow, "olivine-basalt" is from the existing literature, and in particular in the already published geological map (Carrasco-Nunez et al., 2017b). However, we agree that the name is imprecise, therefore, we modified along the whole text in Cpx-free Ol-basalt (Texcal) and Cpx-bearing Ol-basalt (intracaldera basalts).
With respect to Fig. 1, the names and abbreviations of volcanic unit follow the geological map of Carrasco et al. (2017b). We specified that in the original caption.
Concerning the only 2 trachytes, we collected many different samples from the two trachytic lavas (LH5-1 and Lh6-1), all samples from the first trachyte show cpx+ol-bearing assemblages, all samples from the second trachyte are cpx+ol-free. We selected the two most representative samples, one for each trachyte.
Concerning the number of samples, we collected in the field more than 60 samples (as indicated by sample names such as LH27-2…). For every sample we collected a fresh block of 3-5 kg. Petrographic characterization was produced for near all collected samples. We concentrated then our efforts on the most fresh, preserved and representative samples, one for every major magmatic products of the post caldera stage.
Concerning basalts there are only three basaltic lava flows (Texcal, Xalapasco and Los Potreros). We agree with reviewer these samples could be a few for a good petrogenetic study based on whole rock geochemistry and isotopes. However, our aim is the thermobarometry modeling, and we think that selected representative samples for mineral investigation on primary assemblages are quite enough to build up a working model.
However, reading the text in the original form, we agree with reviewer: it seems we worked on just 13 samples. We rewrote the first part of Materials and Methods (Lines 201-209 of the revised manuscript), and better presented our work, and the number of samples we worked over.

*Mineral chemistry.*
*I find the section on cpx section very confusing and unnecessarily complicated with 5 different categories of cpx. The zoning pattern is completely lost. Infact, cpx 3 overlaps cpx1 & 2, therefore it is difficult to see if cpx in basalts and trachytes are actually zoned. If so how?*
We improved the textural description of clinopyroxene, following the nomenclature proposed by Streck (2008) as suggested by the reviewer in another comment. We agree that is necessary to improve the text to better present the five cpx-clusters discriminated through texture observations and mineral chemistry. However, we consider it is not possible to reduce the clusters for the following reasons:
(i) cpx1 and cpx2 are cores of phenocrysts in basalt and trachyandesite+trachyte, respectively. As it can be observed in Fig. 6a, 6d and 6f, Cpx1 and Cpx2 show a completely different chemistry with consequence on thermobarometry models.
(ii) cpx3 is the wider group of all, it represents the chemistry of LOA and monotonous zoning outer rims of cpx1 and cpx2, and also the chemistry of homogeneous unzoned phenocrysts in all studied sample. Looking to the diagrams 6d and 6f, it is possible to recognize some differences (Ti, Ca+Mg+Fe, Na) in the Cpx3 group and therefore, to be overprecise, we should further split it in Cpx3a (basalt) and Cpx3b (trachytes and trachyandesites).
(iii) cpx4 represents microlites and micropheocrystals in groundmass of all studied samples. Their composition partly overlaps that of cpx3. However textural characteristics are completely different (phenos vs. microlites).

(iv) cpx5 represents Na-rich pyroxenes as microlites/microphenocrystals in basalts (LH5-2, LH27-1) and thin external rims of major phenocrysts from few trachyandesites (LH15, LH17, LH26-2).
We agree that splitting too much the cpx populations (as for example creating Cpx3a and Cpx3b subgroup) will be unnecessary in the light of a thermobarometric approach.
However, at this stage it is not possible to reduce the subgroup lower than five clusters.
Putting together Cpx1 and Cpx2 should imply that cores are all equal in composition and it is not true. Putting together Cpx4 and Cpx5 would imply that microlites are all equal and the Na-rich signature would be lost.
Concerning the partially overlaps between cpx1 cpx2 and cpx3, it is related to the fact that we are observing LOA oscillatory zoning and normal monotonous zoning. As we reported addressing one of the previous comments: the similarity of compositions between phenocryst and groundmass microlites suggests that there was essentially no major change in the temperature of any of these magmas during ascent (Rutherford, 2008).
However, we agree that original chapter 6.2 was not in shape for publication. We deeply improved it in the revised version.

*The other interesting feature, the presence of microlite and outer rims of aegirine-augite composition, is hard to correlate to specific composition and/or location. Are the cpx microlites in basalts all cpx5?*
As we wrote in the previous comment: "cpx5 represents Na-rich pyroxenes as microlites/microphenocrystals in basalts (LH5-2, LH27-1) and thin external rims of major phenocrysts from few trachyandesites (LH15, LH17, LH26-2)". Na-rich rims are found diffused in basalts and rare in trachyandesites. Na-rich microlites are found just in basalts.
In the revised chapter 6.2, the description of the five cpx-groups is improved and now clear.

*It is well expected that Fo and Mg# decrease from basalts to trachytes, so this is not really a good criterion for discriminating crystal population. How about the behaviour of trace elements such asCr and Ni?*
We agree with reviewer that Fo and Mg# usually decrease from basalts to trachytes (or more felsic rocks) due to the progressive fractionation/extraction of MgO phases. However, concerning our samples, we observed that phenocrysts describe a three-clusters distribution: (i) basalts with Mg# 79-87; (ii) trachyandesites with Mg# 67-80 and (iii) trachytes with Mg#58-63. These three groups represent the ol-compositions that is entered in the chosen Ol-Liq model of Putirka (2008). Additionally, the use of CaO (wt%) allowed us to graphically (Fig. 7a) improve the discrimination between the three populations, with ol-cores in basalt that discriminated well from all the other olivines.
Concerning the use of Cr and Ni. We didn't analyzed Ni at microprobe, because it wasn't allowed in the selected microprobe configuration. We analyzed Cr (as $Cr_2O_3$ wt%, see table S3), it is always below detection limit (0.04 wt%) in all the analyzed olivines in trachytes and trachyandesites.
In basalts most of the olivines show Cr below detection limit and very few spots produce values in the range 0.04-0.06 wt%. Information about Cr-content is too fragile to be used for a possible discrimination between population.
We have no other choice at this stage to work on the classical Mg# index plus CaO, as suggested by Melluso et al. (2014).

*7.3.2 this part is very difficult to follow, it is a list of P-T condition very difficult to visualise. How do the reader will figure out the difference between opx-free and opx-bearing trachytes if they are not distinguishable in Fig. 11? Please rewrite this section.*
We improved the 7.3.2 section. See Lines 627-631 of the revised manuscript.

*Fig. 11 - The difference between basalts, trachyteandesites and trachytes is almost lost in this diagram, particularly for the trachytes. Maybe Fig. 11 can be redrawn and composed of 4 separates diagrams showing the inferred P-T condition for basalts,trachyandesites and trachytes. All the results can be shown together in the last diagram colour coded for composition. It would be easier to see where the different rock types crystallise and this should also be used in the discussion.*

We agree with reviewer's suggestion. As declared earlier, we prepared 4 new diagrams for different rocks. See the new Fig. 11.

*L64-68 - The recent work of Jackson et al Nature 2018 show a numerical modelling for the network of melt and mush.*

We thank the reviewer with this very interesting suggestion. In addition to the suggested work of Jackson et al (2018), we also consider these: Solano et al. (2014) and Cashman et al. (2017 to improve the manuscript.

*L261-262 – It is not clear what do you mean with a lava flows being free of lithics.*

Right, lava flows commonly doesn´t contain lithic clasts. However, we just want to point out that in the LH studied lava samples no fragments from host rocks or from previous magmatic rocks were observed. Thus, we use the terms lithic-free as proposed by many works such as Geshi and Oikawa (2014).

*L366 – It is quite interesting that plagioclase cores are more mafic in trachyandesites than in basalts. Why?*

We thank reviewer for this interesting observation. We consider two possible explanations for the trachyandesite An-rich plagioclase core (An 75-85%). The first scenario is related to $H_2O$ behavior in magma. The second one implies that these An-rich plagioclases tap a more primitive stage of basalt segregation. We believe that both scenarios are compatible with Los Humeros data, and probably concurred together with the crystallization of An-rich phenocrysts in intermediate and felsic rocks.

We explored these two scenarios in discussion at lines 796-811 of the revised manuscript.

We also checked the dataset, and we verified the existence of a limited low quality analyses. We decided to delete them and correct the manuscript coherently.

*L432-441 – No need of this introduction*

Following the reviewer's suggestion, we delete this paragraph.

*L472 - you haven't presented Fig2. 7-10 yet*

We checked carefully the text. All figures are presented in the correct order.

*L482-484 – please revise the English.*

We rephrase the sentence as follow (Lines 515-520 of the revised manuscript):

"Following the textural criteria previous defined, all microprobe analyses related to those rare crystals presenting morphological evidences of disequilibrium, such as patchy zoning (from BSE images), were discarded".

*L485-487 – Strongly oscillatory zoning can be due to different causes mainly magma mixing and/or P-T-H2O fluctuation, it reflects the dynamics of magmatic processes and might be not an indication of equilibrium. On the contrary the low-amplitude oscillatory zoning is kinetically controlled (see Streck 2008 for further details). Therefore, it is important to stress which type of oscillatory zoning the authors refer to. In addition, the fact that the zoning is oscillatory precludes that core and rim formed in evolving liquids with progressively different compositions as argued.*

We agree with the reviewer that a more precise definition of zoning is necessary, since different zoning patterns are indicative of different crystal-melt conditions. Therefore, we made a careful revision of types of zoning and textures.
As declared at Lines 228-229 of the revised manuscript, now the nomenclature of types of zoning and textures follow Ginibre et al. (2002), Streck (2008) and Renjith (2014).

*L488-489. The use of WR composition instead of glass is problematic especially when dealing with microlite.*
Yes, we agree with the reviewer comments about the problems related to the use of WR composition as nominal liquid in equilibrium with microlites/microphenocrystals.
As written in the original manuscript (lines 451-454), we are aware that such procedure put the focus on the early (at depth) steps of the crystallization history and increase the uncertainties on the final stage of crystallization.
However, since the paucity/absence of glass in the groundmass, we had no other choice than use the WR also for microphenocrysts and microlites.
We applied the same tests and rules for all other minerals. We accepted (very few) results only from those minerals in equilibrium with the WR composition (nominal liquid).

*L502-503 – On which basis this additional criterion has been chosen?  The test for equilibrium according to Putirka 2008 is satisfied when KD(Fe-Mg)opx-liq = 0.29±0.06.*
We concur with the Reviewer that the additional test for orthopyroxene is redundant. We deleted it from both text and Opx supplementary table.

*L504-514 – This part is quite confusing.  The Rhodes diagram (Fig. 8) are for olv and opx, eq. 17 of Putirka is for olivine, how the cpx-liq fits with this?*
We agree that the paragraph is quite confusing. Then, we rephrased it (Lines 532-538 revised manuscript).

*Appendix – what is the error on EPMA data? How Actlabs and XRF data compare?*
We now add the analytical error to the methodologies chapter.
(XRF) analyses for major elements compositions: the analytical uncertainties were <1% for $SiO_2$ and $Al_2O_3$, <5% for the other major elements.
(ICP-OES) analyses for major elements compositions: the precision is estimated better than 2% for values higher than 5 wt% and better than 5% in the range 0.1–5 wt%. Data are strongly comparable.
(EMPA) the major-elements analyses of mineral phases show an error of < 2 % for $SiO_2$ and always < 1 % for the other major elements.

Federico Lucci
(on behalf of the coauthors)

---

## Author Comment (AC2) · 9 Oct 2019

REVIEW by Referee#2

We thank Referee#2 for the critical and constructive comments. In the new version of the manuscript we have taken into accounts all the received comments and advice. We detail below how each comment was addressed and considered in the revised version (the Referee's comments typed in italics and our responses typed in red).

*The authors did a great analytical job and their proposal about the multi-reservoir model is in accordance with recent investigations.*
We thank Referee#2 for the appreciation of our job and the constructive comments that have been considered in the revision of the manuscript.

*However, I think data could be used in more detail to explore other scenarios. The main idea will remain, the polybaric model seems unrefutable, but the processes involved during the evolution of magmas could be an important factor for the geothermal system. The model proposed by the authors rely in only one differentiation process; crystallization. Whereas mixing, and perhaps assimilation, seems to be important processes.*
We thank R#2 to confirm that a polybaric model is the only possible explanation for the Los Humeros magmatic activity during the Holocene post-caldera stage.
We agree that many processes such as fractionation, mixing/recharge and assimilation could generally control the magma evolution in feeding systems. However, in the case of Los Humeros, LHPCS products do not show any evidence of mixing and assimilation processes.
Following the comments of Referee#1, we produced major elements mass balance modelling and demonstrated that all studied samples can be produced trough Rayleigh fractional crystallization (FC) of the same fixed mineral assemblage of Pl+Cpx+Ol+Sp (see new sectiuon 8.1). Therefore, all LHPCS studied rocks belongs to a single line of descent and can be generated without the necessity to invoke assimilation or mixing phenomena.
Furthermore, fabric and textures observed in all samples clearly exclude the possibility of assimilation and mixing/recharge processes.
As we wrote in the response to Referee#1, all Los Humeros samples (from basalts to trachytes) are characterized by unzoned homogenous or normally growth/zoned phenocrystals.
We explained already in the reply to reviewer # 1 the conditions for the zoned phenocrysts, and the complete lack of magma mixing for the LHPCS.
However, we agree with R#2 that this point must be better discussed in the manuscript as we do in the new version.

*Something should be explained about the origin of these magmas. Once explained, you can go further and propose how do you think they evolved.*
As we answered to Referee#1, we consider that the origin and the evolution of the Los Humeros post caldera stage (LHPCS) magmatism, requests a deeper petrological approach, which is far beyond the scope of this manuscript. Instead, this paper is focused on the reconstruction of the anatomy of its magmatic plumbing system using the crystal-melt pairs (as in Stroncick et al., 2009; Aulinas et al., 2010; Dahren et al., 2012; Keiding and Sigmarsson, 2012; Coombs and Garner, 2014; Barker et al., 2015).

*I think mixing is evident, for example:*
*1) TAS diagram looks very linear. Find published liquid lines of descent for basalts evolving to trachytes just by crystallization. I think they do not look like the trend dis-played by your samples.*
We already addressed this point in answer to Referee#1 (see also the answer above)
However, a liquid line of descent where basalts evolve to trachytes could be found in White et al. (2009) (in the reference list of the original manuscript). Linear trends could be found also in i Moghadam et al. (2016) and in Lucci et al. (2016) where FC models were used to demonstrate

lines of descent of mafic parents evolve to felsic daughters. Here, we applied the same mathematic model. References added in the revised manuscript.

*2) Plagioclase microlites and phenocryst rims show a very wide compositional range An20-63. Is evident this are not in equilibrium.*
Phenocrysts and microlites show a wide compositional range as normally expected for progressive crystallization from the reservoir to the surface during the ascent of the magma.
We applied to all phenocrysts and to all microlites, the same Min-Liq test for equilibrium, and we selected for thermobarometry models, only those in equilibrium with the nominal melt. The other were discarded.

*3) The presence of Cr-rich titanomagnetites and Cr-poor titanomagnetites in the same sample. In fact, I suggest to remove the ulvospinel data, is of bad quality. Either, the crystals were very small and you excite the surrounding matrix or something failed with the standardization. Although data is not publishable, the relative abundance of Cr is evident and you should explain it.*
We disagree with this comment. As fig 4 (BSE images) shows, analysed specimens were big enough to shot: Cr-spinels phenocrysts are up to 200 µm in diameter (4a). Ilmenite crystals in the groundmass are usually 20-30 µm width and up to 100 µm length (4c, 4d). Concerning Ti-magnetites we always shooted on phenocrysts with up to 50-100 µm in diameter (4g) or on crystals in groundmass with ca 20 µm in diameter (4f).
We reported dimension of spinel in the revised section 6.5, entitled "Spinel and Opaque Minerals".
The used spot sizes were 1-10 µm, depending on the phases analysed (see appendix A methods of the submitted mansucript).
Concerning the possibility to excite the surrounding matrix, spinels are in a groundmass of Cpx+Ol+Pl (and they are Cr-free, see the EMPA Supplementary tables) or they are included in Cpx phenos and Ol-phenos (with $Cr_2O_3$ <0.06 wt%). Therefore, the only source for Cr is the spinel itself.
Concerning the hypothesis of failed standardization, in such case all elements should be wrong not only Cr. Our analyses close very nice (Total 90-97 wt%) and recalculated formulas are coherent with those expected for the Spinel group. Therefore, there is no evidence to say that they are bad analyses.
We are confident to have demonstrated that our analyses are good, and we will keep them as it is.
In the light of thermobarometry, and since no spinel-liquid model was used, discussing the chemistry of spinel is out of scope of the manuscript.

*Thermobarometric results should be used with caution. Its hard to match the results reported on the text with the supplementary data.*
We do not understand what the Referee#2 means for "thermobarometric results should be used with caution". In our work we produced a dataset of ca. 2400 EMPA spots and we obtained, through the application of opportune tests, more than 1200 mineral-liquid equilibrium pairs for the application of thermobarometric models.
We discussed the results following a statistical approach (for every population the internal error and the MSWD were calculated). The results were then graphically presented in a constrained P-T space.
We think we have been overcautious.
We disagree with the statement that it is hard to match the results with supplementarty tables. Every table is divided for phenocrysts and groundmass (as for olivine, orthopyroxene and spinel) or by clustered population (as for Pl and Cpx). For every rock it is used a conventional abbreviation AB (basalt), TA (Trachyandesite) and TR (Trachyte).

In text (chapter 7.3) P-T results are discussed for rock and for mineral, therefore the reader just needs to enter the supplementary material and search for the right mineral phases and rocks.
It is important to understand that we are handling a voluminous dataset of 2400 EMPA analyses, so it requires time to fully explore it. Therefore, we appreciate a lot that R#2 went through the dataset.

*But for example, plagioclase data of almost the same composition found in different WR samples is used to calculate temperature and pressure. These models almost always will yield a number, the idea is to generate a good interpretation for the results, the best possible approximation.*
We analysed the mineral chemistry of a specific sample and then we compared it to the hosting magmatic rock. We used opportune test (as recommended) and then we applied the mineral-liquid thermobarometry model and we discussed statistically the results. As declared in the previous point we have been overcautious, we made our best to propose a model that is strictly based on data without any unconstrained speculation.

*Even if is not evident on their tests, the authors should incorporate explanations on how mixing could affect their thermobarometry results. Protracted heating-mixing could be the driving force for convection-conduction in a geohermal reservoir.*
As we already explained in the response to Referee#1's comments, there is no evidence of mixing in the studied samples.
However, we would like to stress here that Mineral-Liquid thermobarometry models, as defined in all the existing literature, are based on the compositional equilibrium existing between a mineral and its hosting liquid, no matter the origin of the liquid.

Figure captions should be more descriptive and informative. The manuscript is overall well written. Captions should be improved.
We thank Referee#2 for appreciating our work. When necessary, figure captions were modified

**Detailed comments**
*L351-L353. Here you highlight the dissolution on pyroxene. You should do the same for plagioclase.*
Growth mantle is not only due to dissolution of pre-existing grains. It can be achieved because pre-existing grains are more energetically favourable than nucleating new crystals, as explained in the existing literature (e.g. Streck, 2008; Coombs and Gardner, 2004).

*L368-L374. Aegirine is an index mineral on peralkaline rocks. The same for anorthoclase. You reported non peralkaline rocks occur in the studied area. Mixing processes should be explored. The wide compositional range on "microlites" reported above could suggest the same mixing process.*
*L389. Fayalite is present, same comment as above. Fayalite occur in peralkaline rhyolites.*
We know that aegirine and fayalite are considered a clear indication of peralkaline magmatism when they are found together also with sodic amphiboles (i.e., Afverdsonite) in felsic melts (rhyolite and trachytes).
However, we have found aegirine and fayalite compositions only in the groundmass of cpx-bearing ol-basalts, not in felsic melts.
Concerning aegirine, they are outer rims on major cpx phenocrysts and microlites in groundmass. Concerning olivine, basalts show invariably suites of olivines with maximum forsterite (Fo) contents in equilibrium with the respective whole rocks, and vertical trends consistent with closed-system melt differentiation (Roeder and Emslie, 1970; Rhodes et al., 1979; Putirka, 2008; Melluso et al., 2014).
No need to invoke at this stage a peralkaline felsic melt. The evolving fractional crystallization process in an alkali-basaltic closed system could produce both aegirine and fayalitic-olivine. By the

fact, at lines 245-247 of the original manuscript we wrote: "on the total alkali versus silica (TAS) diagram (Le Maitre et al., 2002) LHPCS lavas span from basalts to basaltic trachyandesites, trachyandesites and trachytes (Fig. 3b). Los Humeros mafic rocks fall in the "Basalt" field and, according to Bellieni et al. (1983), Le Maitre et al. (2002) and Giordano et al. (2012), can be classified as alkali-basalts".

Concerning anorthoclase, it is found as phenocrysts and microlites in little amount in all studied rocks. Anorthoclase in alkali basalt could be generated by crystallization due to local high activity of fluids (e.g., Morten and De Francesco, 1993; Upton et al., 2009).

Since all the LHPCS belong to the same line of descent, and the parental melts are transitional- and alkali-basalt there is no need to claim assimilation for anorthoclase presence. This mineral phase is part of the same line of descent.

*L446. It's correct to use the WR; however, you need first to define why this is valid if mixing could modified some magmas. The whole rock would simply be an integrated result of all magmatic processes (mixing+assimilation) that occured just before theeruption.*

We thank Referee#2 for confirming that use of WR is correct in our approach. Furthermore, thanks to the comments of both Referee#1 and Referee#2, we demonstrated in the revised manuscript, that no mixing nor assimilation processes occurred prior to eruption. We believe that this definitively validate our approach for unravel the anatomy of the plumbing system of the LHPCS magmatic activity.

*L484. This sentence is not clear. A crystal could have a patchy zone at the core and then be in equilibrium with the melt (rim). In your sentence you should clarify or have are ference to one of your figures.*

The sentence is a general postulate. However, we agree with reviewer Referee#2 that the sentence must be rephrased. We correct all the paragraph (lines 493-505 of the revised mansucript).

*L485-L487. Not neccesarily; just changes in temperature would record extreme com-positional changes in crystals, without any mixing involved (mass exchange).*

We want to clarify that at lines 485-487 of the original manuscript, we were not talking about mixing or mass exchange. We are just declaring that as general rule for the application of mineral-liquid thermobarometry euhedral crystals should be selected, whereas those showing morphological evidence of disequilibrium should be discarded. Hereafter the original sentence:

The predominant euhedral to subhedral habit of crystals is generally considered an evidence of equilibrium with the surrounding melt (e.g., Keiding and Sigmarsson, 2012). Accordingly, in the first step of our analysis we discarded from the analyzed dataset the minor cluster of crystals presenting morphological evidence of disequilibrium such as patchy chemical zoning (from BSE images).All the microprobe analyses related to the very rare crystals presenting resorbed rims and patchy zoning, were discarded.

As we declared in the previous comments, all the paragraph has been improved in the revised manuscript (see new section 7.1).

*L488-489. First you need to clarify and yield some confidence to the reader about the origin of the melts and the magmatic processes that modified each magma. If mixing occurred, then the WR is a mixture of xenocrysts and phenocrysts. How does this affect the equilibrium between the phenocrysts and the WR? You should clarify the phenocrysts you used, is not clear and not proven the criteria to choose them. "........pristine liquids in equilibrium with early crystallized phenocrysts and microlites"? This is confusing; What dou you mean as microlites??, decompression induced crystals grown during decompression-cooling? Or these are microphenocrysts?? Do you mean all melts where tapped as they were formed?, I mean, how is posible that a liquid is in equilibrium with an early crystallized phenocryst and a microlite??*

As already written earlier, the aim of this work is not to discuss the origin and the petrogenesis of Los Humeros magma. Aim of this work is reconstruct the anatomy of the plumbing system through a thermobarometric approach. We have also demonstrated that all LHPCS studied rocks belong to the same line of descent and derives from the same basaltic parental melt. Textural observations together with FC modelling discard any possibilities of mixing processes.

Regarding the comment "you should clarify the phenocrysts you used and not proven the criteria to choose them", we want to make some comments: Chapters 7 and section 7.1 and 7.2 are dedicated to the presentation of all the tests required for the correct identifications of Mineral-Liquid pairs and how to apply them. All the existing literatures relative to the tests and to the parameters are fully presented in the manuscript.

Phenocrysts and microlites are related to textural characterization, now improved in the revised version of the manuscript. Tests for equilibrium were applied to all minerals. Only those passing the opportune test were then considered for thermobarometry.

Concerning the last point, not all microlites are in equilibrium with the nominal liquid. Those that crystallised during the rapid ascent (see Rutherford, 2008; reference added in the revised manuscript) are not in equilibrium and were not selected for thermobarometry modelling.

*L502-503. You need to explain why this value was chosen.*
This second Opx-test is redundant. We delete it.

*L550. You should delete the negative values. Have no petrological meaning, no matter other authors have interpreted them as anhydrous. All models will yield data, the job of the petrologist is to evaluate which are usable.*
Following interpretations of the existing literature (Keiding and Sigmarsson, 2012; Putirka, 2008), which suggest that the negative values represent a general anhydrous melt condition for the basalts (2010 Eyjafjallajökull eruption), we are just considering our negative values obtained from the same hygrometer in the same way. An anhydrous character corresponds to $H_2O$ < 1wt% content (e.g. Webster et al., 1999). We agree with Referee#2, the job of the petrologist is to present all the obtained data, evaluate them and discuss them, also in the light of the existing literature.

*L551-563. Does these water contents match the pressure calculated with other meth-ods used in this investigation? 1.40 wt.% is very shallow. If these are subduction related magmas how is possible they are anhydrous? I know is dictated by the model, but what do you think, the model approach well the problem?? Moreover, if basalt are required to be anhydrous, then what is the origin of the water required for the evolved LHPCS?? Have you try other hygrometer, different to Lhur and Housh 1991?*
As we have stated earlier and, in the manuscript, we are not dealing with the origin of the LHPCS magmas, instead our aim is to rebuild the anatomy plumbing system of a volcano using a thermobarometry-based approach.

Concerning anhydrous basalts, we addressed it in the previous point. Concerning the felsic rocks, we demonstrated that they are daughter melt of the same line of descent. As written in one of the previous comments, the term "anhydrous basalt" indicates a basaltic melt containing $H_2O$ < 1 wt%. The new FC modelling indicates that trachytes are the ca. 25 wt% of residual melt from the basaltic parental melt. Trachytes, following the hygrometer of Putirka (2008) show a water content of 2 wt%.

A water content of 2 wt% in the residual fractionated liquid, corresponds to $H_2O$: 0.5 wt% in the basaltic parental melt.

Here it is demonstrated how to obtain water in felsic rocks, via fractional crystallization starting from a nominal anhydrous melt.

Concerning the hygrometer, we tested that of Putirka (2008) and that of Lange et al. (2009). The choice of the hygrometer of Putirka (2008) is due to the fact that this hygrometer is a parameter of the Pl-Liq barometry model (Putirka, 2008) used in this manuscript.

*L567-569.  You have a great amount of xenocrysts with felsic compositions.  Many of the cores where temperature was calculated have the same composition as other cores measured in TA. The temperature is different because the WR composition in which are suposely in equilibrium varies. So, which one is the system in equilibrium??*
We would like to clarify that we do not have any xenocrysts of felsic composition in our studied samples. Maybe some plagioclase antecryst cores (see fig. 10), and very rare olivine xenocrystals (Fig. 8a).

*L634-637.  Are these megacrysts or resorbed crystals?, fragmented crystals??.  Is there any possibility these are intrusive xenocrystals??. Give a reference for subsolidus equilibration of groundmass after eruption or explore an alternate possibility.*
We presented petrography in section 5. Alkali feldspar are phenocrystals and microcrystals with euhedral to subhedral habit. It exists a wide literature related to feldspar re-equilibration in subsolvus and subsolidus conditions. We improved the manuscript (Lines 643-649 in the revised manuscript) and added reference for feldspar subsolidus re-equilibration (Nekvasil, 1992; Kontonicas-Charos et al., 2017; Plumper and Putnis, 2009, Brown and Parsons, 1994) and for groundmass subsolidus re-equilibration (Latutrie et al., 2017).

*L650-653.  This is confusing, or at least, with not enough information for the reader in order to understand the author's point of view.   Are these basalts the product of differentiation of the former basalts mentioned just before? These evolution is recorded along the crystallization of the Fe-rich olivine, albites and aegirines? Report the textura of the  borders, are  they  in  equilibrium.  Is this alkaline-low  oxygen  fugacity  mineral assemblage the result of mixing with more alkaline melts? Mixing could occurred at depth or at shallow pressures??*
Cpx-bearing basalts derives to the LH18 basalt through Cpx crystallization and crystal accumulation. We demonstrated it with the new FC-Cumulus modelling (see answer to R#1 comments).
We improved the textural description of mineral phases in petrography and mineral chemistry chapters  (see new section 6).
As addressed earlier in this comment, there is no evidence mixing and assimilation processes in the LHPCS studied samples.

*L657-658. Then, what happened to the proposal that basalts should be anhydrous?*
We already addressed the anhydrous character of the LH basalts earlier

*L670-673.  I suggest a brief proposal about the origin of these magmas should beexplained.  If mafic and intermediate are tapped almost straight from deep reservoirs, then what is the origin for felsic magmas?, do they arrived already evolved to shallow levels?  Moreover, the intermediate melts do not have any traces from mixing?  Mixing has been reported as one of the mechanisms to produce andesite-dacite melts.*
As already written earlier in this comment, in the LHPCS studied samples there is no evidences of magma mixing. The new FC model demonstrates that all the studied rocks belong from the same line of descent and derive from a progressive fractionation of the same mineral assemblages from the same parental basalt (see new section 8.1). As already declared here (and in the manuscript), our focus is the geometry of the plumbing system in the light of geothermal system evolution. Petrogenesis is out of scope. We agree that magma-mixing is one of the mechanisms invoked to produce andesite-dacite melts. However, (i) you can produce andesite-dacite melt also just with

Rayleigh FC (see Moghadam et al., 2016; Lucci et al., 2016), and (ii) we do not have no andesite nor dacite at Los Humeros.

*L674-683. I disagree. You need to explain why some TA and AB have a mixure of restitic phases as ulvospinel (titanomagnetite)? What I mean is that some samples contain evolved titanomagnetites and Cr-rich titanomagnetites. This is a strong evidence for mixing.*
As we wrote earlier, mixing processes would lead to particular textures (fine-sieve, resorption surface, glomerocrysts, reverse zoning, reaction rims, dissolution, breakdown mantle and crystal clots). Restitic phases due to mixing processes should be characterized by disequilibrium textures. None of these textures, nor restitic crystals are observed in our sample. Even if this point has been already addressed earlier, we would like to outline again that Cr-spinel is found as big crystal inclusion in major Mg-rich olivine. Other spinels phenocrysts show very low content of Cr, whereas Cr content dramatically fall to 0 in groundmass microcrystals. This is compatible with a progressive fractional crystallization, in which Cr is controlled by spinel. Again, there is no need to invoke mixing processes.

*L682-L683. I do not understand what you try to explain. Rephrase.*
We reworded that sentence

*L694-695. These are not the only evidence for mixing.*
We rephrase the sentence.

*L715. If I remember well this is the first time alkali-basalt is mentioned. Should be pointed before, this would explain the originof aegirine anf fayalite???*
We presented alkali-basalts at Line 244-247 in chapter 4 of the original manuscript, when we presented the WR chemistry in TAS diagram. We are glad that Reviewer R#2 agrees with us, fayalite and aegirine belongs to the normal evolution of an alkali-basalt.

*L717-718. Harker diagrams are not the best option to explain fractional crystallization. In fact, it seems that the trends are very linearn typical of binary mixing-assimilation. Lets think about the mixture of Fe-Ti oxides you have, mixing occured. A very least you should show trace element evidence. Trace elemnts are very sinsitive to mixing and assimilation, so you could adjust your model.*
We agree with reviewer that Harker diagrams are not the best to sustain FC hypothesis. Therefore, following also the comment by Referee#1, we produced major element mass balance model coupled with Rayleigh fractional crystallization model (see section 8.1 of the revised typescript). We demonstrate that pure FC is able to describe the single line of descent of Los Humeros post caldera stage products. The results obtained through FC-models are in agreement with textural observations and petrography of studied rocks. We agree with reviewer that trace and REE elements are very sensitive to FC-AFC-Mixing processes, however they would request a deep approach that should be comprehensive of the discussion of petrogenesis of magmas. This is far beyond the scope of this manuscript, focused on the possibility to unravel the geometry of a plumbing system using the relationship between mineral assemblages and hosting melts.

*L721-724. Then, how does this support crystalization acting alone to form the tra-chytes??*
We agree with the reviewer. Since the petrogenesis of Los Humeros would request a deeper discussion of whole rock geochemistry (major, trace and REE) and Sr-Nd isotope. We delete the sentence. And we reorganized the paragraph coherently.

*L750. So what is the origin of these anhydrous basalts? If these evolved to more felsic melts what is the origin of water in those?*

As said earlier, it is not the aim of this work to discuss the origin (petrogenesis) of Los Humeros magma. Concerning the felsic rocks and their water content, we already addressed this point earlier.

*L806-813 This paragraph is very confusing. Do you mean this alkaline basalt will arrive almost at its liquidus at shallow pressure, and only then aegirine and fayalite would crystallize?? What is the origin of this alkaline basalt? He basalt has ilmenite-ulvospinel?, calculate the oxygen fugacity and discover is is enough reduced to crystallize fayalite-aegirine.*

LH18 basalts (Texcal) made of Ol+Pl+Sp, shows olivines with dendritic, skeletal and spinifex textures, associated with plagioclase with swallow-tailed textures.The only way to achieve these textures is a rapid ascent from the deep reservoir to the surface (see references in the revised manuscript).

Concerning Cpx-bearing basalts (LH5-2; LH27-1), if we see them erupted, it means that they ascent to the surface at their liquidus conditions. It is hard to imagine a solidified basalt ascent to the surface. Concerning aegirine and fayalite in basalts, the thermobarometric results (discussed at line 806-813 of the original manuscript) indicate that these two phases crystallized in the final ascent of the basaltic magma prior to eruption. As we already explained, the petrogenesis of the LHPCS magmas is far beyond the scope of this work. Knowing the origin of the basalts, it will not change the P-T results or the geometry of the feeding system. Also, oxygen fugacity and discovery the thermodynamic conditions for fayalite-aegirine crystallization are far beyond the scope of this manuscript.

*Figure 3. There are multiple attempts to recreate liquid line of descents for melts evolving by fractional crystallization from basalts with these compositions. Search and try to fit your data. To me look very linear and probably related to massaddition processes.*

As we answered earlier, we demonstrated with mass balance modelling and Rayleigh fractional crystallization that all LHPCS studied melts belong to the same line of descent and are all produced from FC processes starting from the same parental basaltic melt. No need to invoke further assimilation or mixing processes.

*Figure 5. Basalts, trachyandesites and trachytes contain anorthoclase as phenocrysts and in the groundmass. I do not remember an explanation about this in the text. Couldbe mixing with alkaline melts??*

As we explained earlier, the parental melt is an alkali basalt. Aegirine, fayalite and anorthoclase are mineral phases compatible with the melt composition.

Federico Lucci
(on behalf of the coauthors)

---

## Author Response (AR2)

Dear Dr. C.J. Lissenberg, Dear Dr. J. Gottsmann,

many thanks for Your editorial handling of the manuscript entitled "Anatomy of the magmatic plumbing system of Los Humeros Caldera (Mexico): implications for geothermal systems".

Please find attached the revised version of the manuscript named above.

This second revision of the manuscript has been edited following Your comments and advice.

Please, see our responses below (typed in red). Changes in the edited typescript are shown in red.

**REVIEW by Executive Editor (Dr. J. Gottsmann)**

The topical editor has come to the decision to recommend publication pending a couple of technical corrections. While I endorse these recommendations, I do not consider the paper publishable in its current form.

Looking at the ms in its entirety, I conclude that a final careful revision of the English must be conducted. There are numerous elements in the text where the writing can be much clearer, less ambiguous and grammatically correct. As an example, I please look at Conclusion #4. The current wording makes no sense, is impossible to understand and grammatically incorrect.

The uptake of the published paper by your peers will be improved by a well-written narrative. I therefore expect to see a revised version of the manuscript including the technical comments and revision of the English before I make my final decision on the ms.

We apologise for the quality of the English text. We carefully revised the text that was checked by the English mother tongue co-Author (J.C. White).

**REVIEW by Topical Editor (Dr. C.J. Lissenberg)**

The first relates to the analytical totals for the EPMA data, which was raised (for oxides) by Reviewer 2 and appears to be a general issue for all phases. The totals for anhydrous phases are generally below 100%, and regularly below 98%. There needs to be a way for the reader to verify data precision and accuracy, but the analytical techniques section on EPMA analyses does not mention methods or results of standard analysis, nor precision and accuracy. Given the importance of mineral chemistry for the thermobarometry, and the generally low totals, I think this should be discussed. I would recommend adding a table summarising the repeat analyses of the external mineral standards used to Appendix A.

We improved Appendix A.3 "Mineral chemistry". It is now specified the set of standards used for the EMP analyses. A table (Supplementary Table 7) with repeated measurement of two mineral standards (natural wollastonite and natural fayalite) is added. Data precision ($1\sigma$) and accuracy ($1\sigma$) are now reported. Existing literature used to compare and validate the mineral chemistry results obtained is now reported for each analysed phase.

We checked the EMPA dataset and we verified the existence of few analyses with low totals. We decided to delete them and correct the manuscript accordingly.

However, concerning the low totals, it is not true that the anhydrous phases are regularly below 98 wt%. Average totals are: (i) Pl-core: 98.35 wt%, (ii) Pl-rim: 98.26 wt%, (iii) Pl in ground mass (gm) : 98.46 wt%, (iv) Afs phenochrysts (ph) 98.74 wt%, (v) Afs-gm 98.71 wt%, (vi) Cpx1 98.48 wt%, (vii) Cpx2: 99.46 wt%, (viii) Cpx3: 99.28 wt%, (ix) Cpx4: 99.61 wt%, (x) Cpx5: 98.47 wt%, (xi) Ol-ph: 99.95%, (xii) Ol-gm: 99.22 wt%, (xiii) Opx-ph: 99.81 wt%, and (xiv) Opx-gm 99.93 wt%.

As a general rule, analyses with totals between 98 and 102 wt%, with ferric iron ($Fe_2O_3$) calculated, can be considered appropriate.

Since most of the thermobarometry models used in this work are based on the ferrous iron, we decided to present the original EMP analyses, without any further $Fe^{2+}$ - $Fe^{3+}$ correction.

Concerning the low total of some clinopyroxene (in particular those found in basalts, e.g. Cpx1 and Cpx5), we must remember that the LHPCS magmatic system is characterized by the important presence of Aeg compound (Na + $Fe^{3+}$).

Concerning low total of some feldspars, we would like to remember that: (i) all iron should be corrected as $Fe_2O_3$, and (ii) we did not measure BaO and SrO since they were not useful for thermobarometry. However, feldspars in volcanic environments, could show up to 1-1.5 wt% of BaO+SrO.

Taking into account the above points, we believe that improving the totals could risk being merely cosmetics without any effective results/influence on the thermobarometry results.

Nevertheless, we agree with the Editor that the Appendix A.3 needed important technical improvements.

Second, in Table S1: please correct 'Fedspar'
Done.

Finally, regarding Table 1: Given that all of the mineral compositional data are in supplementary tables, I wonder whether the whole-rock geochemical data may be better off in a supplementary table as well. That way, any reader can easily download the Excel sheet for both liquids and crystals and so reproduce the thermobarometric calculations.

We would like to keep the Table 1 (Whole Rock geochemical data) in the text.
However, WR data are reported also in Supplementary Table 6 where they are used for FC-modeling.

We hope in the present form the manuscript may fulfill criteria for publication in Solid Earth.

Your Sincerely,

Federico Lucci
(on behalf of the co-authors)

---

## Author Response (AR3)

Dear Dr. C.J. Lissenberg, Dear Dr. J. Gottsmann,

many thanks for Your editorial handling of the manuscript entitled "Anatomy of the magmatic plumbing system of Los Humeros Caldera (Mexico): implications for geothermal systems".

Please find attached the revised version of the manuscript named above.

This third revision of the manuscript has been edited following Your comments and advice.

Please, see our responses below (typed in italics). Changes in the edited typescript are shown in red.

**REVIEW by Topical Editor (Dr. C.J. Lissenberg)**

**RESPONSE:** *We apologize for the presence of the remaining grammar and spelling issues in the manuscript. We fixed all the highlighted issues plus other misprints found along the whole manuscript after a further grammar check.*

*We hope in the present form the manuscript may fulfill criteria for publication in Solid Earth.*

Your Sincerely,

Federico Lucci
(on behalf of the co-authors)